# Impact of social isolation on grey matter structure and cognitive functions: A population-based longitudinal neuroimaging study

Laurenz Lammer[1]*, Frauke Beyer[1,2,3], Melanie Luppa[4], Christian Sanders[5,6], Ronny Baber[6], Christoph Engel[6], Kerstin Wirkner[6,7], Markus Loffler[6,7], Steffi G Riedel-Heller[4], Arno Villringer[1,2,8], A Veronica Witte[1,2,3]*

[1]Department of Neurology, Max Planck Institute for Human Cognitive and Brain Sciences, Leipzig, Germany; [2]Clinic for Cognitive Neurology, University of Leipzig Medical Center, Leipzig, Germany; [3]CRC Obesity Mechanisms, Subproject A1, University of Leipzig, Leipzig, Germany; [4]Institute of Social Medicine, Occupational Health and Public Health, University of Leipzig, Faculty of Medicine, Leipzig, Germany; [5]Department of Psychiatry and Psychotherapy, University of Leipzig Medical Centre, Leipzig, Germany; [6]Leipzig Research Center for Civilization Diseases (LIFE), University of Leipzig, Leipzig, Germany; [7]Institute for Medical Informatics, Statistics and Epidemiology (IMISE), University of Leipzig, Leipzig, Germany; [8]Berlin School of Mind and Brain, Humboldt University of Berlin, Berlin, Germany

*For correspondence:
lammer@cbs.mpg.de (LL);
veronica.witte@medizin.uni-leipzig.de (AVW)

**Competing interest:** The authors declare that no competing interests exist.

## Abstract

**Background:** Social isolation has been suggested to increase the risk to develop cognitive decline. However, our knowledge on causality and neurobiological underpinnings is still limited.

**Methods:** In this preregistered analysis, we tested the impact of social isolation on central features of brain and cognitive ageing using a longitudinal population-based magnetic resonance imaging (MRI) study. We assayed 1992 cognitively healthy participants (50–82 years old, 921 women) at baseline and 1409 participants after ~6y follow-up.

**Results:** We found baseline social isolation and change in social isolation to be associated with smaller volumes of the hippocampus and clusters of reduced cortical thickness. Furthermore, poorer cognitive functions (memory, processing speed, executive functions) were linked to greater social isolation, too.

**Conclusions:** Combining advanced neuroimaging outcomes with prevalent lifestyle characteristics from a well-characterized population of middle- to older aged adults, we provide evidence that social isolation contributes to human brain atrophy and cognitive decline. Within-subject effects of social isolation were similar to between-subject effects, indicating an opportunity to reduce dementia risk by promoting social networks.

**Funding:** European Union, European Regional Development Fund, Free State of Saxony, LIFE-Leipzig Research Center for Civilization Diseases, University of Leipzig, German Research Foundation.

## Editor's evaluation

This is a large-scale study on how social isolation influences brain structure and cognition. Baseline social isolation and change in social isolation were associated with smaller hippocampus volumes,

reduced cortical thickness, and poorer cognitive function. The level of evidence is strong and converging cross-sectionally and longitudinally.

## Introduction

Over 50 million humans suffer from dementia today. In just 20 years, this number will likely double. Already now, dementia's global annual costs exceed one trillion US dollars (*Prince et al., 2015*), and its detrimental effects on the lives of the afflicted make it a major contributor to the world's burden of disease (*Vos et al., 2020*).

Research on pharmacological interventions targeting dementia pathogenesis has not yielded any result with a clear clinical benefit yet (*Knopman et al., 2021*), and available drugs targeting cognitive symptoms offer at most a minor alleviation (*Knight et al., 2018*). Henceforth, prevention is of cardinal importance and potentially modifiable risk factors are our most promising target (*Livingston et al., 2020*).

Systematic reviews and meta-analyses have concluded that social isolation, the objective lack of social contact, is such a risk factor for dementia (*Kuiper et al., 2015*; *Penninkilampi et al., 2018*) and its main feature cognitive decline (*Evans et al., 2019*; *Kuiper et al., 2016*). Assuming causal relationships, Livingston et al. calculated population-attributable fractions for risk factors for dementia and concluded that 3.5% of cases could be attributed to social isolation. This is almost as many as to obesity, hypertension, and diabetes combined (*Livingston et al., 2020*).

Risk factors of later dementia development often affect the structural brain changes dementia is characterized by: vascular degeneration, amyloid plaques, tau fibrillary tangles, neural degeneration, and grey matter loss. Neuroimaging correlates of these brain changes have been observed multiple years prior to symptom onset in autosomal-dominant dementia (*Gordon et al., 2018*) and can already be detected in cognitively healthy persons using neuroimaging (*Ewers et al., 2011*; *Jack et al., 2013*). Thus, brain magnetic resonance imaging (MRI) can be a potent dementia-risk indicator (*Wang et al., 2019*), might offer pivotal guidance to identify patients for intensive dementia prevention (*Ten Kate et al., 2018*), and serve as secondary outcome for intervention trials (*Stephen et al., 2019*). Still, the link between brain structure and social connection, the umbrella term encompassing social isolation, social support, and loneliness, has not received much attention (*Wassenaar et al., 2019*). Some studies have linked low social connection to an elevated 'brain age' gap estimate (*de Lange et al., 2021*), changes in microstructural (*Molesworth et al., 2015*; *Spreng et al., 2020*; *Tian et al., 2014*), and volumetric measures in brain regions including the hippocampus and the prefrontal cortex (*Blumen and Verghese, 2019*; *Cotton et al., 2020*; *Düzel et al., 2019*; *James et al., 2012*; *Schurz et al., 2021*; *Shen et al., 2022*; *Spreng et al., 2020*; *Taebi et al., 2020*); however, these cross-sectional designs render conclusions about causality difficult. In a longitudinal study using a small sample of 70 participants (37 at follow-up) > 80 years old, microstructural deteriorations and a larger total white matter hyperintensity volume correlated with decreases in predominantly social activities (*Köhncke et al., 2016*). Furthermore, it suggested that white matter changes mediated the positive association between social activities and perceptual speed (*Köhncke et al., 2016*). *Mortimer et al., 2012* conducted a small randomized controlled trial (RCT) with older adults and found increased total brain volumes and cognitive function in participants after a social interaction intervention compared to a non-intervention control group.

Taken together, the current evidence suggests social isolation to have an adverse effect on brain health. Still, data from longitudinal studies are required to distinguish between from within-participant effects on brain structure and cognitive function and to gain insights into temporal dynamics and causal relationships. Furthermore, to pointedly leverage the power of such datasets for an improved understanding of the effect of social isolation, conceptual clarity regarding the dimensions of social connection is pivotal but still lacking.

Moreover, no solid evidence on the mechanistic underpinnings of the relationship between social isolation and accelerated brain ageing exists. Several mutually non-exclusive, partly overlapping theories are used to explain the beneficial effects of social interaction (*Hultsch et al., 1999*; *Kawachi and Berkman, 2001*). Amongst them, the stress-buffering hypothesis puts forward the beneficial effects of social support in strenuous times on mental, cognitive, and immunological health (*Kawachi and Berkman, 2001*), yet this mediating effect has not been explored regarding brain measures.

Longitudinal population-based neuroimaging studies now offer reliable sample sizes to gain knowledge on effect sizes and disentangle correlation from causation to better understand the impact of social isolation on brain and cognitive ageing. In this pre-registered analysis, we aimed to determine the relationship between social isolation, measured using the Lubben Social Network Scale (LSNS-6, *Lubben et al., 2006*), and brain structure and cognitive functions, measured using FreeSurfer segmentations on advanced high-resolution MRI at 3 Tesla and neuropsychological testings, in a large well-characterized longitudinal sample of mid- to late-life individuals (n > 1900) from the Health Study of the Leipzig Research Centre for Civilization Diseases (LIFE) (*Engel et al., 2023*).

To this end, we applied linear mixed effects modelling and structural equation modelling to predict volume of the hippocampus, a focal point of age-related atrophy and Alzheimer's disease pathology (*Rodriguez et al., 2020*), by baseline social isolation and change in social isolation over time. Analogously, we modelled memory performance, processing speed, and executive function, as well as whole-brain vertex-wise cortical thickness. Significance was evaluated based on frequentist p-values and Bayes factors, and we adjusted for control variables including age in all models. Details on MRI preprocessing and predefined statistical analyses were preregistered at https://osf.io/8h5v3/.

We hypothesized that both baseline and change in social isolation would correlate with smaller hippocampal volume, cognitive functions (memory, processing speed, executive functions), and cortical thickness. Additionally, we hypothesized interaction effects of baseline social isolation with change in age in the same direction. Moreover, we aimed to test a mediating role of chronic stress as well as hippocampal volume on cognition in these models and explored possible gender differences in stratified analyses.

**Table 1.** Descriptive statistics.

| Variable | BL, **N = 1992** | FU, **N = 1409** |
|---|---|---|
| Gender (female) | 921 (46%) | 656 (47%) |
| Baseline age (years) | 67 (7) \| 50 \| 82 \| 0 | 68 (7) \| 50 \| 84 \| 0 |
| Change in age (years) | 0.00 (0.00) \| 0.00 \| 0.00 \| 0 | 5.89 (1.97) \| 0.00 \| 9.40 \| 15 |
| Baseline LSNS | 14.1 (5.2) \| 0.0 \| 30.0 \| 181 | 13.7 (5.1) \| 0.0 \| 30.0 \| 20 |
| Change in LSNS | 0.00 (0.00) \| 0.00 \| 0.00 \| 0 | 0.39 (4.38) \| –21.00 \| 18.00 \| 115 |
| HCV (mm³) | 3671 (411) \| 2022 \| 4871 \| 83 | 3487 (430) \| 1,913 \| 4579 \| 665 |
| BMI (kg/m²) | 27.9 (4.2) \| 16.8 \| 46.8 \| 0 | 27.8 (3.7) \| 18.1 \| 46.5 \| 0 |
| Hypertension | 1219 (61%) | 830 (59%) |
| Diabetes | 367 (18%) | 239 (17%) |
| education | 255 (13%) | 153 (11%) |
| CESD | 10 (6) \| 0 \| 48 \| 104 | 10 (6) \| 0 \| 48 \| 62 |
| Memory (SD) | 0.03 (0.97) \| –8.79 \| 1.70 \| 84 | –0.06 (1.04) \| –5.84 \| 1.64 \| 314 |
| Processing speed (SD) | 0.09 (0.92) \| –7.80 \| 1.73 \| 12 | –0.14 (1.10) \| –7.80 \| 1.61 \| 214 |
| Executive functions (SD) | 0.12 (0.95) \| –4.59 \| 3.26 \| 11 | –0.21 (1.04) \| –4.43 \| 3.29 \| 210 |
| TICS | 58 (27) \| 0 \| 166 \| 1480 | 57 (27) \| 0 \| 146 \| 938 |
| Pandemic | 0 (0%) | 412 (31%) |

Values for categorical variables: n (%) yes; values for continuous variables: mean (SD) \| minimum \| maximum \| n missing.

HCV = right-left average hippocampal volume. BMI = body mass index. LSNS = Lubben Social Network Scale, calculated as 30 – LSNS to make larger values indicate greater social isolation. TICS = Trierer Inventar zum chroischen Stress. CESD = Center for Epidemiological Studies Depression Scale. SD = standard deviation. education = no tertiary education.

## Results

We included all individuals equal to or over the age of 50 with available neuroimaging of LIFE (*Engel et al., 2023*) due to the accelerated volume shrinkage starting at about 50 y of age in the hippocampus (*Fjell et al., 2013*). To avoid reverse causation, we further excluded cognitive impairment or prior brain pathology such as history of stroke, neurodegenerative disease, or brain tumours. In total, we analysed 1335 participants at baseline and 912 participants at follow-up with a mean age of 67 and 73 y, respectively, thereof 51% women and an ~6 y mean change in age at follow-up. For various sensitivity analyses, we reincluded participants that did not meet our preregistered inclusion criteria from the entire sample of 1992 participants at baseline and 1409 at follow-up. The sample displayed a high prevalence of cardiovascular risk factors, with 60% hypertension and <20% diabetes, and 11–13% had no tertiary education (*Table 1*).

Individuals exhibited LSNS scores ranging across the whole spectrum, with an average score of 16 and 19.7% scoring below the accepted threshold of 12, indicating elevated risk of social isolation, similar to other populations (*Lubben et al., 2006*). Individual trends in social isolation are depicted in *Appendix 1—figure 1*. Note that for further analyses, LSNS values were calculated as 30 – LSNS to make larger values indicate greater social isolation and coefficients should thus be interpreted accordingly. Hippocampus volumes derived from T1-weighted high-resolution anatomical MRI scans at 3T (*Reuter et al., 2012*) showed shrinkage with higher age of about –0.75% per year (*Figure 1*, left panel), similar to previous estimates (*Fjell et al., 2013*). To test the effects of social isolation on hippocampal volume, we conducted hierarchical linear mixed effects models adjusting for confounding effects of age, gender, and random effects of the individual in a first model (model 1), and additionally for cardiovascular risk factors in a second model (model 2). We differentiated within- and between-subject effects (*van de Pol and Wright, 2009*) of social isolation and investigated the interaction effect of baseline LSNS and change in age to test whether participants that are socially more isolated at baseline experienced more pronounced age-related changes. Please see osf.io/8h5v3/and 'Methods' for details.

In our sample, social isolation was positively correlated with not living alone, being married, the number of persons living in the participants' dwelling, being gainfully employed, younger baseline age, and less change in age but no to gender or having a migration background. See *Appendix 1—tables 1 and 2* for descriptive statistics and details of the associations. To contextualize the observed link to SES, a comparison of SES category frequencies in LIFE-Adult and a fully representative sample (*Lampert et al., 2013b*) is provided in *Appendix 1—table 3*.

### Social isolation and hippocampal volume

We found that both stronger baseline social isolation (values for models 1/2: β = −5.6/–5.7 mm$^3$/point on the LSNS (pt), FDR-corrected q-value (q) = 0.0034/0.0078) and increases in social isolation (β = −4.7/–4.5 mm$^3$/pt, q = 0.0035/0.0066) significantly predict smaller hippocampal volumes independent of confounders (*Table 2*, *Figures 1–3*). Significance of these findings is further underlined by Bayes factors of 15–19 for baseline social isolation and of 2–3 for change in social isolation. The effect size of one point on the LSNS is equivalent to a 2.5-month difference in baseline age.

### Social isolation and cognitive functions

In analogous linear mixed effects models, we tested the effects of social isolation on cognition, measured using domain-specific composite scores based on z-scored results of the trail-making test (TMT A and B) and the CERAD-plus test battery (CERAD – Consortium to Establish a Registry for Alzheimer's Disease, RRID:SCR_003016) assessed under standardized conditions (*Beyer et al., 2017*). Overall, stronger baseline social isolation and to a lesser extent increases in social isolation were linked to worse cognitive performance (*Table 3*, *Figure 1*). Specifically, stronger social isolation at baseline significantly predicted lower executive functions (β = −0.028/–0.017 SD/pt, q = 9.6e-09/0.0014) and lower processing speed (β = −0.018/–0.017 SD/pt, q = 1.2e-05/3e-04). The link to lower memory (β = −0.014/–0.008 SD/pt, q = 0.0016/0.0914) was strong in model 1 but did not survive FDR-correction when controlling for additional covariates. Increases in social isolation over time significantly predicted lower memory in models 1 and 2 (β = −0.019/–0.0018 SD/pt, q = 0.0034/0.0142) but not processing speed (β = −0.007/–0.008 SD/pt, q = 0.238/0.198) and executive functions (β = –0.003/0.001 SD/pt, q = 0.41/0.69). Very high Bayes factors corroborate and substantiate the evidence for the negative

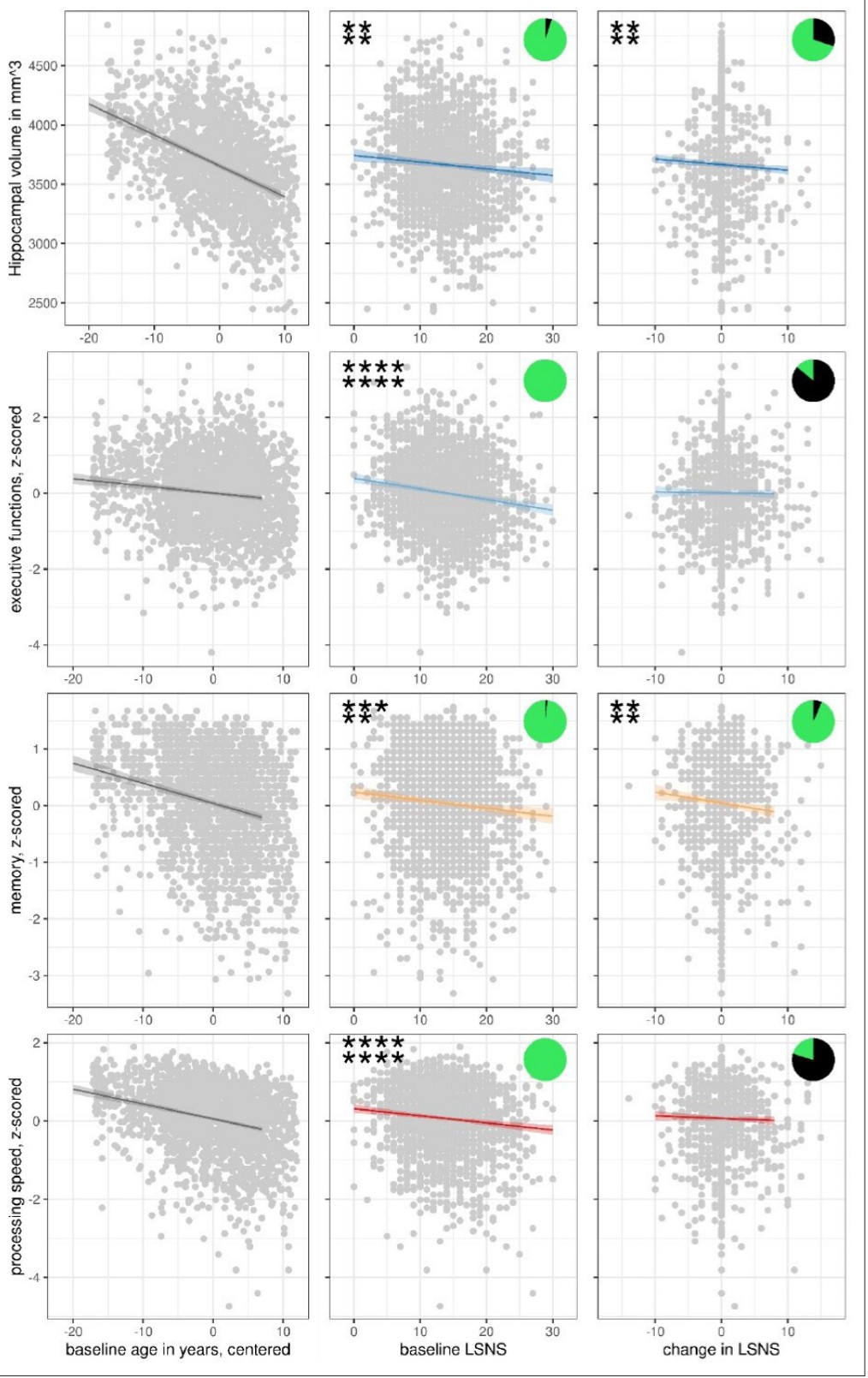

**Figure 1.** Scatterplots with regression lines and 95% confidence intervals for model 1. Asterisks show frequentist levels of significance. The first and second lines show values before and after FDR, respectively. ****p<0.0001, ***p<0.001, **p<0.01, *p<0.05. LSNS, Lubben Social NEltwork ScaPie charts show Bayesian relative evidences. The green and black arc lengths represent the evidence in favour of the alternative and the null hypothesis, respectively.

**Table 2.** Adjusted regression coefficients and measures of significance.

| dv | Model | Predictor | Estimate | 95% CI | p-value | FDR | BF |
|---|---|---|---|---|---|---|---|
| Hippo campal volume | 1 | LSNS_base | −5.6 | −9.3,−2 | 0.0013** | 0.0034** | 18.95** |
| | | LSNS_change | −4.7 | −8,−1.3 | 0.0035** | 0.0069** | 2.36 |
| | | age_base | -26.1 | -28.9, -23.2 | | | |
| | | age_change | −26.8 | −29,−24.7 | | | |
| | 2 | LSNS_base | −5.7 | −9.6,−1.8 | 0.0019** | 0.0078** | 14.93** |
| | | LSNS_change | −4.5 | −8.1,−1 | 0.0066** | 0.0158* | 2.47 |
| | | age_base | −24.2 | −27.3,−21.2 | | | |
| | | age_change | −26.8 | −29.2,−24.5 | | | |

| * p<0.05, BF >3 | ** p<0.01, BF >10 | *** p<0.001, BF >30 | **** p<0.0001, BF >100 |
|---|---|---|---|

The unit of effect sizes is mm³/point on the LSNS.

full model1: dv~LSNS_base + LSNS_change + age_base + age_change + gender.

full model 2: model 1 + hypertension + diabetes + education + BMI + CESD.

dv = dependent variable. CI = confidence interval. FDR = q-values after FDR-correction. BF = Bayes factor in favour of alternative hypothesis. LSNS_base = baseline Lubben Social Network Score. LSNS_change = change in Lubben Social Network Score. CESD = Center for Epidemiological Studies-Depression.

effect of baseline social isolation on cognitive functions. *Figures 2 and 3* allow comparisons of these effects with other predictors for the different dependent variables.

We did not observe interaction effects of social isolation on hippocampal volume or cognitive performance with age. *Appendix 1—tables 4–6* provide a comprehensive summary of all LMEs and predictors including covariates.

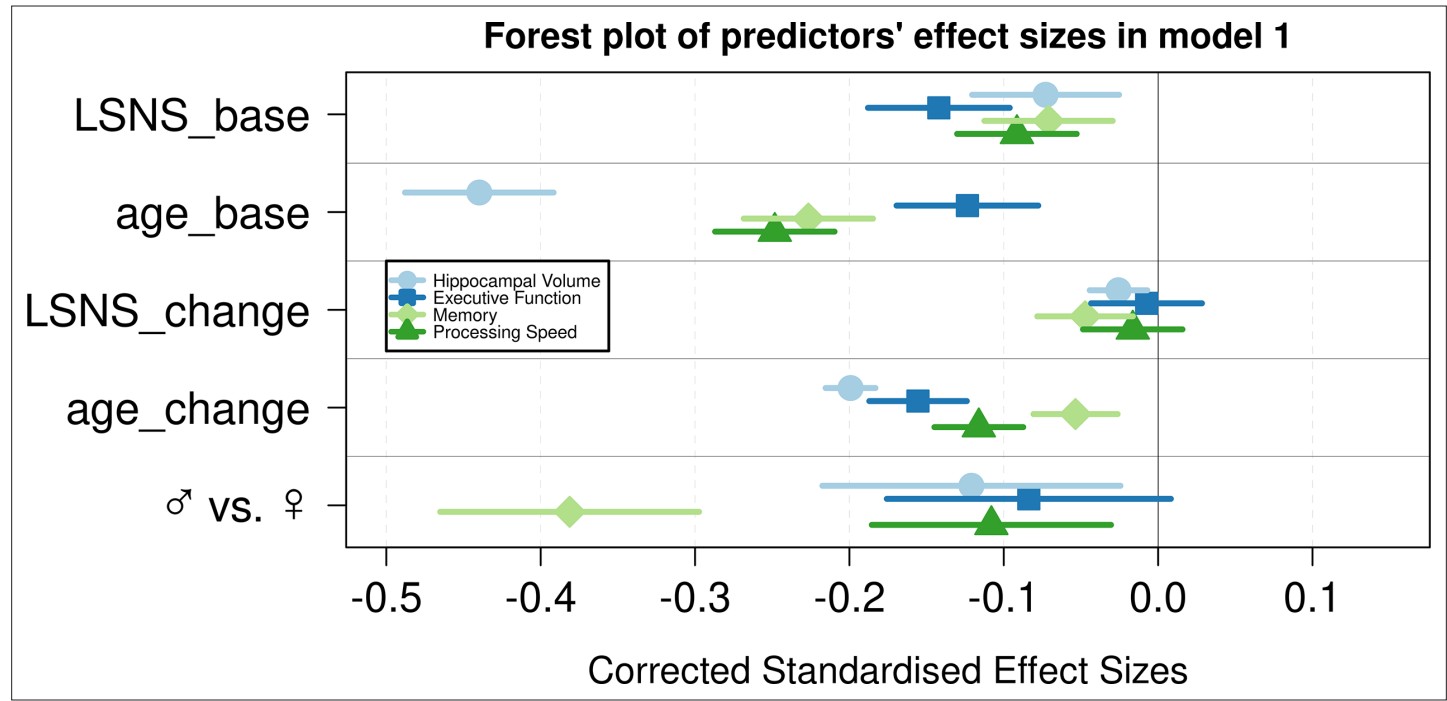

**Figure 2.** Forest plot of predictors' effect sizes in model 1. For the gender variable and the education variable being women and having at least a tertiary degree were coded as 0, respectively. Betas were standardized by the standard deviations of the dependent and independent variable. LSNS_base, baseline Lubben Social Network Scale; age_base, baseline age; LSNS_change, change in Lubben Social Network Scale; age_change, change in age.

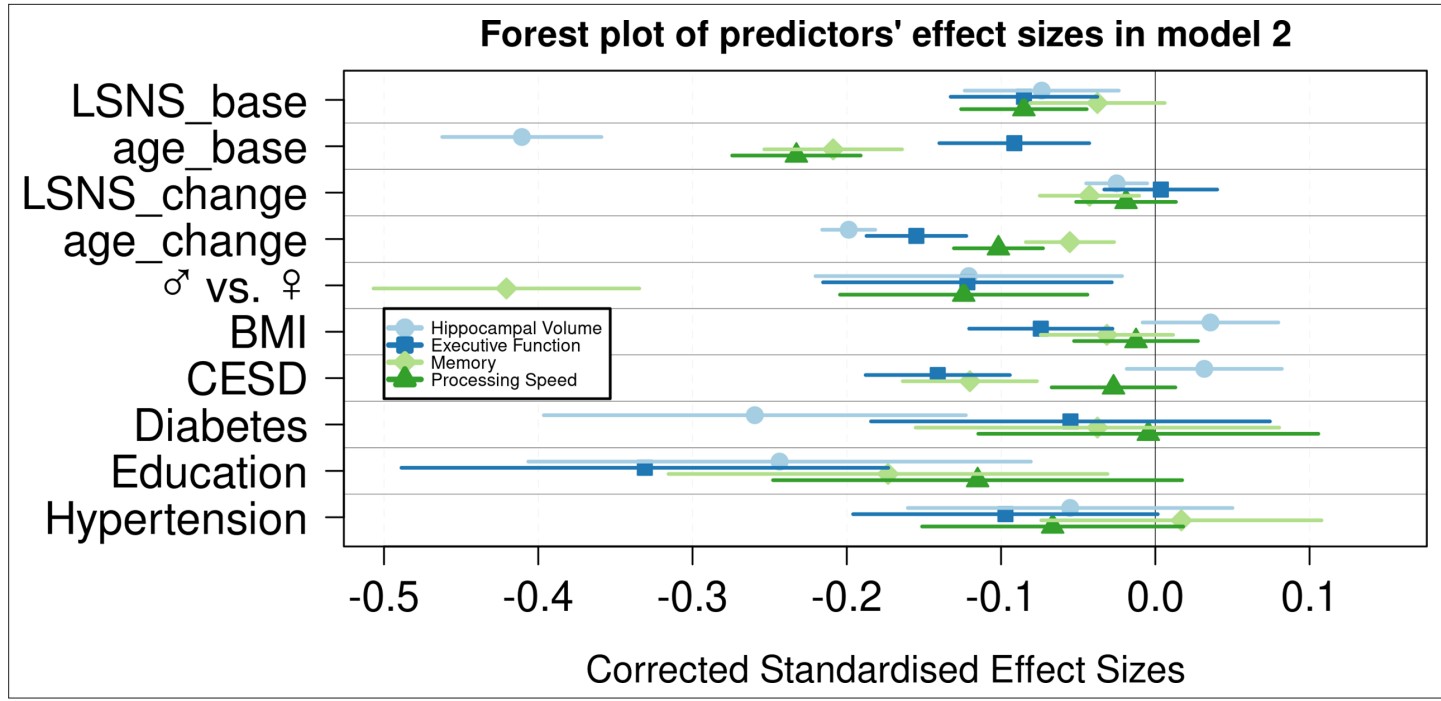

**Figure 3.** Forest plot of predictors' effect sizes in model 2. For the gender variable and the education variable being women and having at least a tertiary degree were coded as 0, respectively. Betas were standardized by the standard deviations of the dependent and independent variable. LSNS_base, baseline Lubben Social Network Scale; age_base, baseline age; LSNS_change, change in Lubben Social Network Scale; age_change, change in age.

## Social isolation and cortical thickness

To explore whether social isolation affects regional cortical thickness, we conducted whole-brain vertex-wise linear mixed effects analyses on FreeSurfer-derived 3D cortical maps (*Reuter et al., 2012*). In model 1, we found a total of eight clusters of significantly decreased cortical thickness associated with stronger baseline social isolation after FDR-correction with an α-level of 5% (*Figure 4*). The clusters were located in the left precuneus, cuneus, precentral gyrus and posterior cingulate gyrus, and right supramarginal gyrus and cuneus. Increases in social isolation over time were linked to decreased cortical thickness in one cluster in the right superior frontal gyrus (*Figure 5*). When additionally controlling for cardiovascular covariates (model 2), no significant clusters were detected. *Table 4* lists these clusters, their locations, and sizes.

## Mediation analyses

Turning to the stress-buffering hypothesis, we investigated whether perceived stress, measured using the Trierer Inventar zum chronischen Stress (TICS) (*Schulz and Schlotz, 1999*), mediated the relationship of social isolation and hippocampal volume. Moreover, we investigated whether hippocampal volume mediated the association between social isolation and cognitive functions. Specifically, we investigated the indirect path resulting from the regressions of follow-up mediator on baseline LSNS and follow-up dependent variable on baseline mediator.

Neither the mediation analyses with chronic stress as a mediator (n = 51 complete observations) nor the mediation analyses with hippocampal volume as a mediator (n = 341–360) yielded significant results. Due to the requirements of the model design and over 50% missingness in the stress questionnaire, the sample sizes of the mediation analyses were gravely diminished. Details on the mediation analyses are provided in *Appendix 1—table 7*.

## Sensitivity analyses

In addition to these pre-registered analyses, we conducted sensitivity analyses to test the robustness of our results on hippocampal volume and cognitive functions. These included possible effects of

**Table 3.** Adjusted regression coefficients and measures of significance.

| dv | Model | Predictor | Estimate | 95% CI | p-value | FDR | BF |
|---|---|---|---|---|---|---|---|
| Executive functions | 1 | LSNS_base | -0.028 | −0.037,−0.019 | 8.0e-10**** | 9.6e-09**** | 1.4e+07**** |
| | | LSNS_change | −0.003 | −0.017, 0.011 | 0.342 | 0.4104 | 0.16 |
| | | age_base | −0.019 | −0.026,−0.012 | | | |
| | | age_change | −0.050 | −0.061,−0.04 | | | |
| | 2 | LSNS_base | −0.017 | −0.026,−0.007 | 2e-04**** | 0.0014** | 128.4**** |
| | | LSNS_change | 0.001 | -0.013, 0.016 | 0.5762 | 0.6914 | 0.13 |
| | | age_base | −0.014 | −0.021,−0.007 | | | |
| | | age_change | -0.051 | −0.061,−0.04 | | | |
| Memory | 1 | LSNS_base | -0.014 | −0.022,−0.006 | 4e-04**** | 0.0016** | 58.91*** |
| | | LSNS_change | -0.019 | −0.032,−0.007 | 0.0014** | 0.0034** | 14.68** |
| | | age_base | −0.035 | −0.042,−0.029 | | | |
| | | age_change | -0.018 | -0.026, -0.009 | | | |
| | 2 | LSNS_base | −0.008 | -0.016, 0.001 | 0.0457* | 0.0914 | 1.23 |
| | | LSNS_change | −0.018 | −0.031,−0.004 | 0.0047** | 0.0142* | 6.67* |
| | | age_base | −0.033 | −0.04,−0.026 | | | |
| | | age_change | -0.018 | -0.028, -0.009 | | | |
| Processing speed | 1 | LSNS_base | −0.018 | −0.026,−0.01 | 1.9e-06**** | 1.2e-05**** | 8.2e+03**** |
| | | LSNS_change | −0.007 | −0.019, 0.006 | 0.1585 | 0.2378 | 0.26 |
| | | age_base | −0.038 | −0.044,−0.032 | | | |
| | | age_change | -0.038 | −0.047,−0.028 | | | |
| | 2 | LSNS_base | −0.017 | −0.025,−0.009 | 2.1e-05**** | 3e-04**** | 1.0e+03**** |
| | | LSNS_change | −0.008 | −0.021, 0.005 | 0.1253 | 0.1981 | 0.53 |
| | | age_base | −0.036 | −0.042,−0.029 | | | |
| | | age_change | -0.033 | −0.043,−0.024 | | | |
| * p<0.05, BF >3 | | ** p<0.01, BF >10 | | *** p<0.001, BF >30 | | **** p<0.0001, BF >100 | |

The unit of effect sizes is standard deviation/point on the LSNS.
full model 1: dv~LSNS_base + LSNS_change + age_base + age_change + gender.
full model 2: model 1 + hypertension + diabetes + education + BMI + CESD.
dv = dependent variable. CI = confidence interval. FDR = q-values after FDR-correction. BF = Bayes factor in favour of alternative hypothesis. LSNS_base = baseline Lubben Social Network Score. LSNS_change = change in Lubben Social Network Score. CESD = Center for Epidemiological Studies-Depression.

the Covid-19 pandemic, effects related to the definition of exclusion criteria or confounder specificities. Analyses accounting for (a) potential effects of measurements before compared to during the Covid-19 pandemic, (b) reducing the exclusion criteria (i.e. not excluding cognitively impaired participants, participants taking centrally active medication, and participants with recent cancer treatment), (c) only including participants with two timepoints and using mean and within scores, (d) using a hypertension cut-off of 140 mmHg, (e) using an MMSE cut-off of <27, (f) additionally controlling for physical activity, (g) additionally controlling for sleep quality, and (h) standardizing cognitive functions using the baseline mean rather than the grand mean confirmed the regression coefficients of our models in terms of direction and size (*Appendix 1—tables 8–15*).

Moreover, we found that treating social isolation as a dichotomous variable, using the standard LSNS cut-off of 12 points, led to results very similar to those of our analyses with continuous LSNS

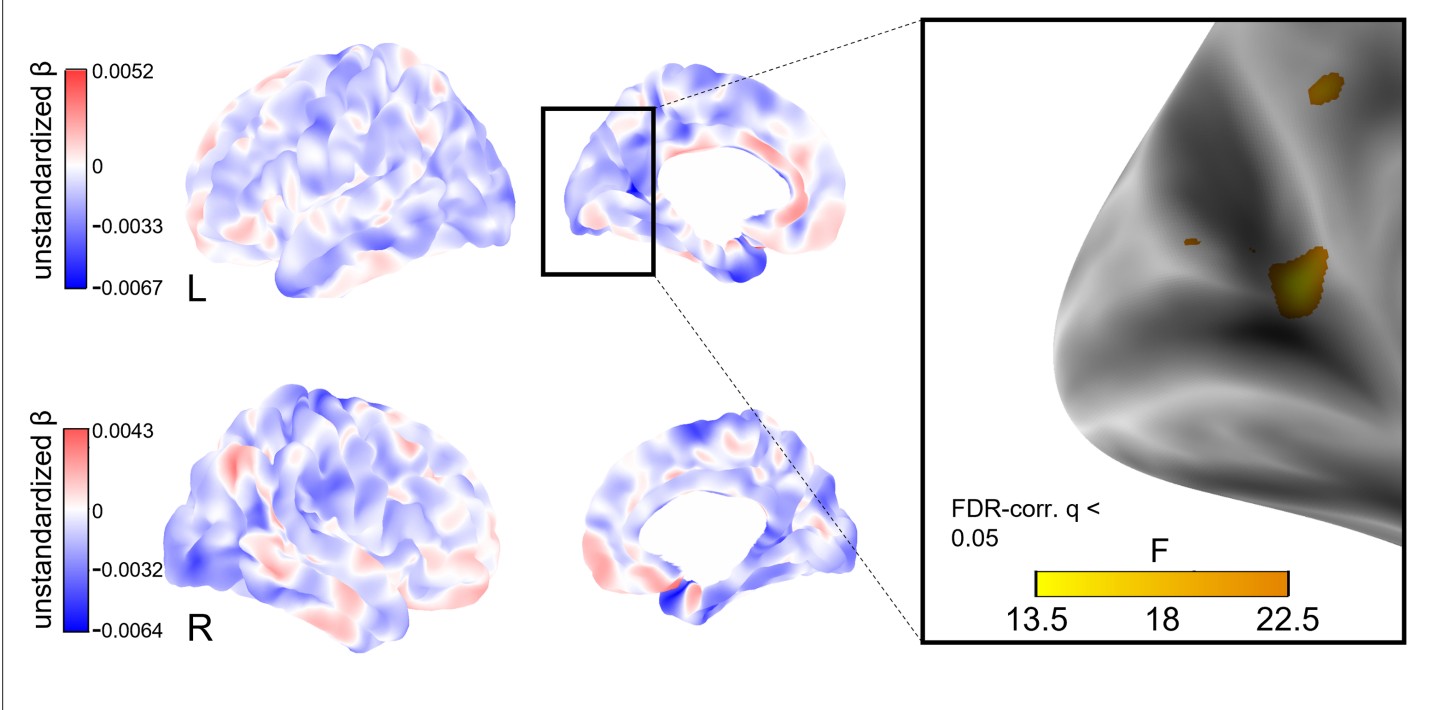

**Figure 4.** Whole-brain analysis of the effect of baseline social isolation on cortical thickness. Unstandardized betas are the vertex-wise effect sizes of baseline social isolation in mm/point on the Lubben Social Network Scale corrected for baseline age, change in age, change in social isolation and gender. The first row shows the left hemisphere. Areas in which stronger isolation links to reduced thickness are marked in blue, the inverse in red. The right hemisphere is shown below. First and second columns show the lateral and medial views, respectively. The box on the right shows three clusters of lower cortical thickness associated with social isolation in the left hemisphere that remained significantly associated after FDR-correction and the F-value of each significant vertex. Significantly associated FDR-corrected clusters in the supramarginal gyrus and cuneus in the right hemisphere and further clusters in the left hemisphere are not highlighted.

scores (*Appendix 1—tables 16–18*). However, we found no evidence for an interaction effect of continuous and categorical LSNS variables (*Appendix 1—tables 19 and 20*).

Of note, neuroscience has historically neglected sex and gender differences, predominantly resulting in increased misdiagnoses of and relatively worse treatments for women (*Shansky and Murphy, 2021*). Therefore, we recalculated analyses in gender-stratified samples (n women = 1110 observations, n male = 1137 observations) to test for differences in the effects of social isolation (*Appendix 1—table 21*). No clear pattern of difference emerged between women and men. A minor observable difference was that the effect of social isolation on hippocampal volume was mostly driven by baseline social isolation amongst women and by change in social isolation amongst men. This pattern was reversed for other outcomes, though.

In order to further investigate the nature of the correlations, we calculated bivariate latent change score (BLCS) models. In these models we simultaneously tested for an effect of baseline social isolation on change in cognitive functions or hippocampal volume and vice versa (see *Appendix 1— figure 2* for a visualization). The BLCS models did not produce solid evidence regarding directionality (*Appendix 1—table 22*). As in the mediation analyses, the design requirements of the BLCS resulted in smaller sample sizes (n = 362–585 complete observations).

## Discussion

In this pre-registered study, we investigated the associations of social isolation with brain structure and cognition in a large cognitively healthy mid- to late-life longitudinal sample. In line with our pre-specified hypotheses, we showed a significant link between stronger baseline social isolation and increases in social isolation over the course of ~6 y and smaller hippocampal volumes. Both predictors had an effect size per point on the LSNS comparable to a 2.5 months difference in baseline

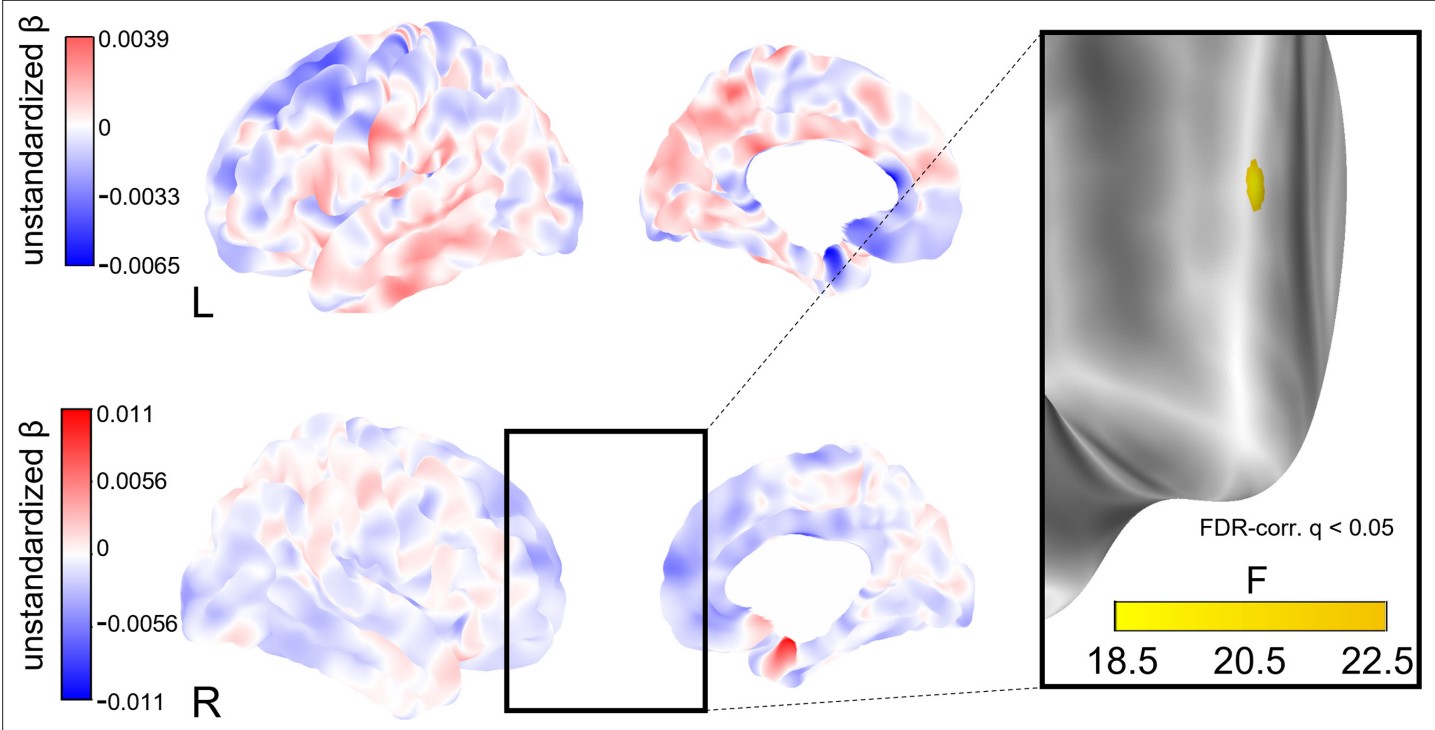

**Figure 5.** Whole-brain analysis of the effect of change in social isolation on cortical thickness. Unstandardized betas are the vertex-wise effect sizes of change in social isolation in mm/point on the Lubben Social Network Scale corrected for baseline age, change in age, and baseline social isolation and gender. The first row shows the left hemisphere. Areas in which stronger isolation links to reduced thickness are marked in blue, the inverse in red. The right hemisphere is shown below. First and second columns show the lateral and medial views, respectively. The box on the right shows a cluster of lower cortical thickness associated with social isolation in the right middle frontal gyrus that was significant after FDR-correction and the F-value of each significant vertex.

age in this age range. Simply put, assuming that if everything else remained stable, the difference between having 1 or 3–4 close and supportive friends is comparable to a 1-year difference in hippocampal ageing. Furthermore, we found significant associations of stronger baseline social isolation with lower executive functions, memory, and processing speed. The link to executive functions was particularly strong with an effect size larger than a 1-year difference in baseline age. For increases in

**Table 4.** FDR-corrected clusters of reduced cortical thickness significantly associated with social isolation.

| POI | Hemisphere | Cortical region | Maximum p-value | Size (mm²) | NVtxs |
|---|---|---|---|---|---|
| LSNS_base | rh | Supramarginal | 1.3e-05 | 43.12 | 114 |
| LSNS_base | rh | Cuneus | 2.8e-05 | 28.04 | 42 |
| LSNS_change | rh | Superior frontal | 1.4e-06 | 43.84 | 65 |
| LSNS_base | lh | Precuneus | 1.1e-06 | 224.34 | 504 |
| LSNS_base | lh | Precuneus | 1.2e-05 | 41.71 | 77 |
| LSNS_base | lh | Cuneus | 9.8e-05 | 10.21 | 15 |
| LSNS_base | lh | Precuneus | 1.1e-04 | 2.74 | 6 |
| LSNS_base | lh | Precentral | 1.1e-04 | 2.32 | 5 |
| LSNS_base | lh | Posterior cingulate | 1.2e-04 | 0.66 | 2 |

full model 1: cortical thickness ~ LSNS_base +LSNS_change +age_base +age_change +sex.

POI = predictor of interest. NVtxs = Number of vertices constituting the cluster. LSNS_base = baseline social isolation. LSNS_change = change in social isolation. rh = right hemisphere. lh = left hemisphere.

social isolation, confidence intervals were wider but effect sizes, except for executive functions, were similar in magnitude to that of baseline social isolation. In multiple sensitivity analyses, we showed the robustness of these findings. Neither applying less exclusion criteria, only including participants with two timepoints, nor controlling for the impact of the ongoing pandemic changed our results substantially. Moreover, we found clusters of decreased cortical thickness in the cuneus, precuneus, precentral, posterior cingulate, supramarginal, and middle frontal gyrus associated with social isolation cross-sectionally or longitudinally. Mediation analysis in smaller sample sizes testing potential effects of social isolation through lowering adverse effects of stress revealed no significant effects.

## Hippocampal volume

Our findings indicate that social isolation contributes to grey matter loss in the hippocampus, a focal point of atrophy in mild cognitive impairment (*Devanand et al., 2007*), and Alzheimer's dementia (*Fox et al., 1996*).

Notably, not only baseline social isolation (a between-subject effect) but also change in social isolation (a within-subject effect) significantly predicted hippocampal volume. Through the employment of statistical LMEs, we were able to distinguish and study effects at these different levels (*van de Pol and Wright, 2009*) and the design helped us to avoid fallacious inferences from single-level data (*Robinson, 1950*) to which simple linear regressions would have been susceptible. Specifically for the study of social isolation as a risk factor for dementia, it is crucial to disentangle between- and within-subject effects. Social isolation has both been described as a trait (*Noonan et al., 2021*), implying it to be an invariant between-subject characteristic and as a potential target for interventions (*Hussenoeder and Riedel-Heller, 2018*), implying it to be a modifiable within-subject effect. The finding of a significant within-subject effect of change in social isolation therefore offers hope for modifiability as it implies that the observed associations are not (exclusively) the effect of an invariant trait. Thus, our data point towards that reducing social isolation could help to maintain hippocampus integrity in ageing.

However, this assumes a causal effect of social isolation. As associations with social isolation could also have resulted from reverse causation through health selection, that is, that participants with accelerated brain ageing are more likely to become socially isolated, this assumption needs careful consideration. Bayes factors imply the absence of an interaction effect of baseline social isolation with change in age and the bivariate latent change score models did not provide evidence in favour of causality in the hypothesized direction either. However, neither did they provide evidence for reverse causality. This inconclusiveness might result from our reduced follow-up sample size and thus related lower power, especially in the latent score models. For example, data from the English Longitudinal Study of Aging from >6000 older adults measured at up to 6 two-year intervals supports the assumed causality of social isolation with regards to memory performance (*Read et al., 2020*). In our study, presence of considerable effect sizes and the high statistical confidence in these estimates on multiple outcomes in our healthy sample without cognitive impairment speaks against the competing hypothesis of reverse causality through health selection and in favour of a causal role of social isolation. Furthermore, the lack of any strong increase in effect size when including health-impaired participants or decrease when applying more stringent exclusion criteria for cognitive health corroborate this interpretation. Still, overall these results only add a modicum of corroboration to the case for a causal role of social isolation.

## Cognitive functions

Baseline social isolation, and to a lesser extent, change in social isolation, were significantly associated with cognitive performance, that is, executive functions, processing speed, and memory, all of which undergo decline in (pathological) ageing (*Blazer et al., 2015*). Again, our results thus imply a detrimental role of social isolation on cognitive functions. We could however not observe that social isolation lowered memory performance through reductions in hippocampal volume, a hypothesis raised by considerations of the central role of the hippocampus in memory (*Buzsáki and Moser, 2013*). Similarly, we could not find evidence that social isolation affected hippocampal volume through higher chronic stress measured with questionnaires, a hypothesis put forward by the stress buffering theory (*Kawachi and Berkman, 2001*). These latter analyses suffered from small sample sizes and a limited number of timepoints. Nonetheless, the lack of any significant link between chronic stress and social

isolation (see *Appendix 1—table 2*) is hard to align with the stress-buffering hypothesis in spite of the missingness in the TICS.

## Cortical thickness

Overall, comparing our brain morphometric results with those of existing cross-sectional studies on social isolation, detected brain regions coincide. A rather small-sampled study did not find a link between social isolation and grey matter volumes (*Lin et al., 2020*) but *James et al., 2012* (occipital lobe), *Blumen and Verghese, 2019* (hippocampus, precuneus, medial frontal gyrus) and *Shen et al., 2022* (hippocampus, right supramarginal gyrus) found decreased volumes in regions we detected, too.

Several of the cortical regions identified in our study (precuneus) belong to the pattern of exacerbated regional atrophy found in Alzheimer's disease. Furthermore, we detected regions known for increased cortical thinning in the healthy process of ageing (cuneus) and both in healthy and pathological ageing (supramarginal gyrus) (*Bakkour et al., 2013*; *Pini et al., 2016*). This indicates an aggravating role of social isolation in cortical thinning that may contribute to normal and accelerated brain ageing processes. However, the findings of lower cortical thickness must be interpreted cautiously due to the limited consistency between cross-sectional and longitudinal effects and the exploratory approach of whole-brain analyses.

## Limitations

A limitation of this study is its uncertain generalizability to the general population because the sample was probably affected by selection and attrition bias common to longitudinal studies (*Chatfield et al., 2005*). Attrition bias might have mostly affected the mediation and BLCS models that thus offered reduced interpretability, despite the comparatively large neuroimaging cohort. However, the LMEs were mostly unscathed by this problem due to their ability to make use of datapoints of participants with only one full observation. In addition, our population represents a WEIRD sample (i.e. western, educated, industrialized, rich, democratic) which might skew our understanding of how social isolation affects brain health (*Laird, 2021*). As we found higher SES to be associated with lower LSNS scores, this relatively high SES sample might have led to underestimation of the detrimental effects of social isolation and increases in social isolation in the ageing process. Considering hippocampus segmentations, it has been argued that FreeSurfer systematically overestimates volumes compared to manual volumetry; however, this difference did barely emerge in participants over the age of 50 (*Wenger et al., 2014*). A further limitation are ceiling effects in the CERAD word list memory task in healthy adults, potentially limiting the sensitivity to detect subtle differences. In addition, time of day during testing might have affected cognitive performance (*Schmidt et al., 2007*), yet we did not control for this. Almost all cognitive tests were performed between 9 a.m. and 1 p.m., though. Covariance of social isolation with other variables such as hypertension or diabetes could have influenced the results. However, note that all variance inflation factors (VIFs) were acceptable, indicating low reason for concern regarding multicollinearity. Lastly, inferences from our results regarding dementia aetiology must be made with caution as we did not investigate clinically diagnosed dementia patients.

In quantitative studies, despite its importance in shaping the research process and conclusions, for example, in functional MRI analysis (*Botvinik-Nezer et al., 2020*), researchers' influence is often disregarded. In the supplementary text, we offer a brief reflexivity section to make relevant influences on this study transparent and to shortly discuss the value of reflexivity for quantitative science.

## Implications for public health and future work

This pre-registered large-scale population neuroimaging analysis adds robust support to the view that social isolation is associated with accelerated brain ageing and cognitive decline in non-demented adults in mid- to late-life. Our findings further imply that social contact protects from detrimental processes and thereby preserves brain structure and function. Henceforth, targeting social isolation through tailored strategies might contribute to maintaining brain health into old age.

We showed that the established LSNS cut-off can be employed by clinicians to identify subjects likely to suffer adverse effects due to social isolation. However, the absence of evidence for more pronounced negative effects of less social contact amongst those that are deemed socially isolated by the cut-off renders a public health strategy focused on high-risk individuals questionable.

While we could not observe significant contributions of physical activity or sleep quality measured using questionnaires in a smaller subsample on brain and cognitive outcomes, previous studies suggested that physical activity (*Musich et al., 2022*) and sleepiness (*Holding et al., 2020*) interact with social isolation and could protect against negative health effects of social isolation, and should therefore be explored in future studies that incorporated these outcomes more systematically.

While we see evidence converging on social isolation as a causal risk factor for dementia and cognitive decline, future neuroimaging studies should pay particular attention to questions of temporality in their design to clear up remaining uncertainties. Studies with more numerous timepoints will be of importance to this end and will furthermore allow us to model important aspects like slopes for individual participants (*van Doorn et al., 2021*). Intervention studies will be the gold standard to provide evidence with regards to the causal role and effect size of social isolation. Multidomain interventions for dementia prevention justifiably become the norm (*Stephen et al., 2019*), so that effects of reduced social isolation must be investigated as a likely contribution to an aggregate effect.

Illuminating the mechanistic underpinnings of the association should be another focus for future research. Studies might prioritize obtaining reliable proxies for the hypothesized mediators. As elevated cortisol levels, in line with the stress-buffering hypothesis, may exert detrimental effects on cognition and contribute to AD pathology (*Ouanes and Popp, 2019*), using hair cortisol, a reliable measure of chronic stress (*Staufenbiel et al., 2013*), could be a promising choice to further investigate this proposed mechanism. In light of the lack of evidence for the stress-buffering hypothesis in our data, alternative mechanistic theories should be pursued, too. The main-effect theory postulates that social relationships foster beneficial health behaviours, affective states and neuroendocrine responses, ultimately protecting neuronal tissue (*Kawachi and Berkman, 2001*). Others point out that socializing is cognitively demanding and requires engagement with complex environments. In the 'use-it-or-lose-it' theory, this is crucial for the maintenance of cognitive function (*Hultsch et al., 1999*). Promising approaches to answer this research question could be interventions specifically targeting one of the hypothesized detrimental processes in isolated individuals and mediation analyses of multi-wave studies with larger sample sizes. Lastly, reverse causality or simultaneity cannot be completely ruled out yet. However, the observed solid correlations in our healthy sample and the lack of an increase in effect sizes when including participants with dementia or low MMSE scores renders this alternative hypothesis to a causal role of social isolation unlikely.

Moreover, studies investigating social isolation due to lockdown measures and its impact on cognitive and brain health will be of great significance.

In light of the relevance of social isolation for cognitive and general health and well-being (*National Academies of Sciences, 2020*), its pervasiveness in the elderly population of the global north (*Livingston et al., 2020*) is alarming. Physical distancing measures have caused an unprecedented rise in the attention to the impact of social isolation but social isolation has been a grave problem before Covid-19 and it will remain a central public health concern thereafter. Existing and future research on reasons for and the role of social isolation in health and disease should provide guidance for the urgently needed development and evaluation of tailored strategies against social isolation and its detrimental effects. These should address social isolation both through intervention strategies on the individual but also societal level, leveraging values like solidarity and communality.

## Materials and methods
### Study design and preregistration
We followed the Strengthening the Reporting of Observational Studies in Epidemiology (STROBE) and Committee on Best Practices in Data Analysis and Sharing (COBIDAS) on MRI guidelines in our reporting wherever appropriately applicable.

The study's preregistration can be found at https://osf.io/8h5v3/. Please refer to it for information on the authors' previous knowledge of the data and a comprehensive overview of our pre-specified hypotheses and models.

### Study population
We used longitudinal data from the 'Health Study of the Leipzig Research Centre for Civilization Diseases' (LIFE). The study was approved by the institutional ethics board of the Medical Faculty of the

University of Leipzig and conducted according to the Declaration of Helsinki. The LIFE-Adult-Study is a population-based panel study of around 10,000 randomly selected participants from Leipzig, a major city with 550,000 inhabitants in Germany. A subgroup of around 2600 participants underwent MRI testing at baseline. The baseline examination was conducted from August 2011 to November 2014. Follow-up assessments were performed around 6–7 y after the respective first examinations (*Engel et al., 2023*). Around 1000 participants of the MRI-subsample returned for follow-up testing.

We included all participants over 50 with MRI data that did not fulfil any of the following exclusion criteria:

- Anamnestic history of stroke
- Any medical condition (i.e. epilepsy, multiple sclerosis, Parkinson's disease)/chronic medication use that would compromise cognitive testing (i.e. cancer treatment in the past 12 mo or drugs affecting the central nervous system)
- Diagnosed dementia or Mini-Mental State Examination (MMSE)-score < 24
- A trained radiologist considered the MRI scans unusable due to brain tumours, or acute ischaemic, haemorrhagic or traumatic lesions

If no MMSE data were available, the participants were excluded if their overall performance in cognitive tests negatively deviated from the wave's mean by 2 standard deviations (SDs) which is a stricter criterion excluding ~2.6% of the sample compared to ~0.8% excluded based on the MMSE. The exclusion criteria were chosen to reduce the potential of reverse causality, that is, dementia symptoms leading to a loss of social connections, as correlations observed in this cognitively intact sample should not stem from dementia symptoms.

## MRI data acquisition, processing, and quality control

We obtained T1-weighted images on a 3 Tesla Siemens Verio MRI scanner (Siemens Healthcare, Erlangen, Germany) with a 3D MPRAGE protocol and the following parameters: inversion time, 900 ms; repetition time, 2300 ms; echo time, 2.98 ms; flip angle, 9°; field of view, 256 × 240 × 176 mm$^3$; voxel size, 1 × 1 × 1 mm$^3$, shimming: tune-up shim, no fat suppression, whole-brain coverage. We processed the scans with FreeSurfer (FreeSurfer, V5.3.0, RRID:SCR_001847) and the standard cross-sectional pipeline recon-all. FreeSurfer automatically measures hippocampal volume, vertex-wise cortical thickness, and intracranial volume. To ensure high within-subject reliability, we employed Free-Surfer's longitudinal pipeline on all scans, including those of participants without a follow-up scan. Please see *Reuter et al., 2012* for details. Moreover, we smoothed the cortical thickness surfaces with a 10 mm kernel to improve reliability and power (*Liem et al., 2015*). Different Linux kernels and Ubuntu versions constituted the computational infrastructure during the data acquisition and processing.

Visual quality control was based on the recommendations of *Klapwijk et al., 2019*. After the baseline data were acquired, our team visually controlled all results of the cross-sectional recon-all pipeline. Additionally, we controlled the outputs of the longitudinal stream of all participants with follow-up data and those whose cross-sectional runs required editing. If we detected errors in the processed scans, we manually edited them (N = 262). We excluded participants from analyses using MRI measures if we deemed the processed scans to be unusable (n = 98).

## Variable construction

### Social isolation

We used the standard Lubben Social Network Scale (LSNS) –6 (*Lubben et al., 2006*) to measure the participants' social isolation. The questionnaire is a suitable tool to measure social isolation (*Valtorta et al., 2016*) has a high internal consistency (Cronbach's $\alpha$ = 0.83), a stable factor structure of the family and non-kin subscale (rotated factor loading comparisons = 0.99) and good convergent validity (correlations with caregiver /emotional support availability and group activity all 0.2–0.46 across multiple sites) (*Lubben et al., 2006*). In order to make larger scores imply more isolation, we subtracted the actual score from the maximum score of 30.

To quantify changes in social isolation, we subtracted the baseline from the follow-up score. For all baseline observation change in LSNS = 0.

In exploratory analyses testing the standard threshold of 12 points, we converted the continuous scores into a dichotomous categorical variable. Change in LSNS scores for these analyses corresponds to positive or negative category shifts.

## Grey matter measures

We used the hippocampal volume derived from FreeSurfer's segmentation and averaged it over both hemispheres. Furthermore, we adjusted it for intracranial volume according to the following formula:

$$HCV_{adjusted, i} = HCV_{raw, i} - \beta * (ICV_{raw, i} - ICV_{mean})$$

where $\beta$ is the unstandardized regression coefficient of hippocampal volume (HCV) on intracranial volume (ICV) from a linear mixed-effects model (LME) (*Jack et al., 1998*).

For whole-brain analyses we used the FreeSurfer fsaverage template and cortical thickness as a vertex-wise outcome.

## Cognitive functions

We calculated domain-specific composite scores and calculated them as follows (*Beyer et al., 2017*):

Executive functions consisted of phonemic and semantic fluency, combined with TMT B/A: executive functions = (z_phonemic fluency + z_semantic fluency + z((TMT B – TMT A)/TMT A))/3.

For the memory score, we defined learning as the sum of three consecutive learning trials of the CERAD word list (10 words), recall as the sum of correctly recalled words after a delay, in which participants performed a nonverbal task, and recognition as the number of correctly recognized words out of a list of 20 presented afterwards: memory = (z_learning + z_recall + z_recognition)/3

Processing speed was defined as the negated z-scored TMT part A score.

Sum-score = z_phonemic fluency + z_semantic fluency + z_sum_learning + z_recall + z_recognition + z((TMT B – TMT A)/TMT A)

Most participants were cognitively tested between 9 a.m. and 1 p.m.

## Stress

Trierer Inventar zum chronischen Stress (TICS) is a German questionnaire assessing perceived stress (57 items, six sub-scales, 0–4 points per item). Its sum score is our measure of participants' chronic stress. The subscales have acceptable to excellent internal consistency (Cronbach's $\alpha$ = 0.76–0.091) and criterion validity of the work overload sub-scale has been shown by demonstrating a significant correlation with cortisol levels over the course of a work days and its ability to differentiate tinnitus patients from healthy controls (*Schulz and Schlotz, 1999*).

## Control and further variables of interest

Month and year of birth of the participants and the date of the MRIs were recorded and used to calculate the age to one decimal point. Age = YOM.MOM – YOB.MOB (YOM/MOM = year/month of MRI, YOB/MOB = year/month of birth). If no MRI was available, we used the date of the LSNS.

For follow-up observations, we calculated: change in age = age at follow-up - baseline age. For all baseline observations change in age = 0.

Data on the following variables was only available for the baseline. Henceforth, we used the baseline values of these control variables for both timepoints.

We calculated the body-mass-index (BMI) according to the standard formula: BMI = weight [kg]/ (height [m])2

In order to control for hypertension and diabetes, we used dichotomized variables. Participants were categorized as hypertensive if they had a previous diagnosis of hypertension, took antihypertensive medication or had an average systolic blood pressure over 160 mmHg. The systolic blood pressure was measured three times. The first measurement was performed after 5 min of rest and three additional minutes of rest passed between each of the following measurements. Participants were categorized as diabetic if they had a previous diagnosis of diabetes, took antidiabetic medication, or HbA1C measured by turbidimetry was $\geq$ 6%.

The participants' education was assessed using an extensive questionnaire (*Lampert et al., 2013a*) and dichotomously categorized based on prior research on education as a protective factor against dementia (*Then et al., 2016*). Please see the supplementary text for details.

Participants had to choose their gender in a binary female/male question. Note that the German 'Geschlecht' does not differentiate between sex and gender. The lack of a clarification and other options is lamented by the authors.

We used the sum-score of the Center for Epidemiological Studies Depression Scale (CES-D) to measure depressive symptoms (*Radloff, 1977*).

For a sensitivity analysis, we created a dichotomous variable coded as 1 if participants answered the LSNS questionnaire after March 22, 2020 (first SARS-CoV-2 lockdown in Germany).

For further sensitivity analyses, we used the global Pittsburg Sleep Quality Index (PSQI) score calculated based on the method proposed by the PSQI authors to measure quality of sleep (*Buysse et al., 1989*) and total physical activity MET-minutes/week as a continuous variable calculated using the International Physical Activity Questionnaire (IPAQ) and its guidelines to obtain a measurement of physical activity (*Hagströmer et al., 2006*).

To explore general participant characteristics of potential relevance to social isolation, we used data on employment, socioeconomic status, marital status, migration background, and number of persons in the participants' dwelling. We categorized participants as non-working if they declared not to be gainfully employed due to other reasons than studying, military, or alternative service. We only considered participants to be married if they also lived with their spouses to avoid including separated but not yet divorced persons as this is more appropriate for the topic at hand. Beyond marital status, not discriminating between legally married couples, cohabitees, and other forms of joined living, we used the number of fellow persons living in the dwelling as a continuous variable and also constructed a categorical variable distinguishing those participants that live alone from those living with others. Participants were considered to have a migration background if they stated that they or at least one of their parents was not born in Germany, thus approximating the definition of the Federal Statistical Office of Germany. Socioeconomic status was at baseline calculated as a metric variable according to the guidelines developed at the Robert Koch Institute (*Lampert et al., 2013b*).

To improve the interpretability of our results, we z-transformed the variables BMI, CESD, TICS, executive function, memory performance, and processing speed by the grand mean and centred the variable baseline age. Additionally, we also centred cognitive performance scores by the baseline mean for a sensitivity analysis.

## Outliers and Imputation

We excluded outliers for our core variables based on a cut-off of 3 SDs (LSNS-score, adjusted hippocampal volume, cognitive functions). Please see *Figure 6* (flowchart) for the limited effect of outlier exclusion on sample sizes of the different models. For further details on outlier detection and handling regarding covariates, please see the supplementary text.

To avoid an excessive reduction in sample size due to missing data, we performed imputations for missing predictor variables using the sample mean, distributions based on existing data, or the participant's mean. Please see the supplementary text for information on our procedures of the respective measures.

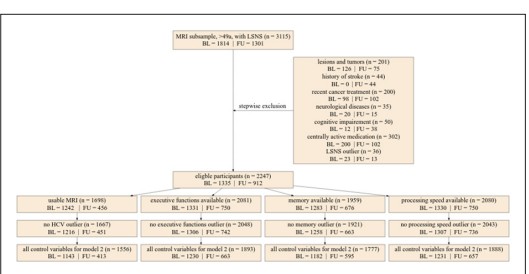

**Figure 6.** Flowchart of stepwise application of exclusion criteria. Small rectangles show the number of participants fulfilling the respective criteria in total and for baseline and follow-up. The large box shows how many participants were excluded due to various exclusion criteria in total for baseline and follow-up. Missing control variables in model 2 were the Center for Epidemiological Studies Depression scores. LSNS, Lubben Social Network Scale; HCV, hippocampal volume; BL, baseline; FU, follow-up.

Furthermore, we used FIML for analyses using structural equation modelling.

## Statistical analyses

All code can be found at https://github.com/LaurenzLammer/socialisolation, (copy archived at *Lammer, 2023*). Please see the supplementary text for information on the software used for the analyses.

## Statistical modelling

### Linear mixed effects models

To investigate the link between social isolation and our outcomes of interest, we employed LMEs with individual as a random effect.

The general structure of the models in the lme4 syntax was:

Dependent variable ~baseline LSNS + change in LSNS + baseline age + change in age + further control variables + (1|participant).

Please see the supplementary text for explicit formulations of all models. We calculated two models for each hypothesis. In model 1, we included age and gender as control variables. Model 2 additionally included education, hypertension, diabetes, depressive symptoms, and BMI. In model 1, the other risk factors are assumed to mediate the effect of social isolation. In model 2, they are assumed to be confounders (see *Appendix 1—figure 3* for a visualization). To measure the effect of ageing, we controlled for baseline age and change in age. Analogously, we differentiated within- and between-subject effects (*van de Pol and Wright, 2009*) of social isolation. Likewise, we calculated the interaction effect of baseline LSNS and change in LSNS. With this methodology we regressed hippocampal volume, the three cognitive functions, and cortical thickness on baseline LSNS, change in LSNS, and the interaction terms. To measure the overall effect of our predictors of interest, we performed a full-null-model comparison (*Bolker et al., 2009*). In addition to standard p-values, we calculated Bayes factors (BFs). The relative evidence was measured by dividing the BF for the full model by the BF of the null model (*Rouder et al., 2016*). This allows us to evaluate the evidence in favour of the full hypothesis compared to the null hypothesis and thus also provide evidence for the absence of an effect (*Keysers et al., 2020*). We report both measures of significance to offer our readers a comprehensive insight into the data, combining the familiarity of classical frequentist inference with the additional implications of BFs (*Keysers et al., 2020*).

## Sensitivity analyses

For the first analysis we added whether participants were tested after the start of lockdown measures to all LMEs. In the second analysis we did not exclude participants due to the intake of centrally active or cancer medication and cognitive impairment. To probe the reliability of the coefficients for LSNS_change, we ran an analysis excluding all participants with only one timepoint and used standard mean and within score calculation. Furthermore, we ran two sensitivity analysis testing whether using a hypertension cut-off of 140 mmHg or an MMSE cut-off of <27 as an exclusion criterion would affect our results. Additionally, we checked whether results would differ if cognitive test scores were standardized by the baseline rather than grand mean and whether the inclusion of physical activity or sleep quality as an additional control variable would affect the results. Furthermore, to test for potential differences in the effect of social isolation between women and men, we divided our dataset by gender and recalculated the frequentist LMEs with both resulting datasets. Moreover, we investigated whether the standard LSNS cut-off would be a sensitive measurement to indicate adverse effects of social isolation on our outcomes and thus be helpful for clinical practice. To this end, we ran our models treating social isolation as a dichotomous categorical variable. Additionally, we ran them with an interaction term of the usual variables with a social isolation category variable to explore if we would find evidence for stronger adverse effects of less social contact amongst the participants deemed socially isolated by the standard cut-off.

To explore links of social isolation to general participant characteristics, we ran LMEs with random intercepts and LSNS sum score as dependent variable. We calculated separate models with socioeconomic status, living alone, number of persons sharing the participant's dwelling, age (differentiated into the two variables baseline age and change in age in one model), employment, gender, chronic stress, migration background, and marital status as independent variables in the full dataset.

## Statistical inference

We report one-sided p-values based on the direction of the predictor/path of interest's regression coefficient and the direction of our pre-defined hypotheses. To obtain one-sided BFs we sampled 10,000 times from the posterior distribution of our predictor of interest's effect. Then we multiplied the BF by 2 and the percentage of sampled effects in the direction of our pre-defined hypotheses.

## **Multiplicity control**

Our threshold for significance for all tests was p<0.05. To control for multiple hypothesis testing we FDR-corrected families of tests and each individual whole-brain analysis (see the supplementary text for definition of families).

BFs of 3–10 and BFs of 10–30 are commonly considered to be moderate or strong evidence in favour of a hypothesis. To evaluate these thresholds in light of multiplicity, we conducted two simulation studies described in the supplementary text that revealed that using a BF threshold of 10.75 rather than 3 would keep α below 5% and that this would not substantially decrease power.

## Model assumptions

To ensure that our continuous predictors are normally distributed, we plotted their histograms. We had to log-transform the CES-D, IPAQ, and PSQI scores to obtain a normal distribution.

To rule out major collinearity, we calculated VIFs. The VIFs did not surpass the threshold of 10 (*Myers, 1990*) in any model.

Furthermore, we tested the stability of our LMEs in R by comparing the estimates obtained from the model based on all data with those obtained from models with the levels of the random effects excluded one at a time. This revealed the models to be fairly stable. Moreover, we visually controlled them for heteroskedasticity with both a histogram and a qq-plot. The qq-plots show a heavy-tailed distribution of the residuals in some models. This is only a minor deficit as the models are not intended to make accurate predictions at specific points (*Gelman and Hill, 2006*).

Fit indices providing further information on the quality of a model fit using structural equation modelling can be found in *Appendix 1—tables 23 and 24* (*Schermelleh-Engel et al., 2003*). Fit index thresholds were surpassed by multiple mediation models. As the BLCS models are saturated, fit indices are uninformative.

## Acknowledgements

We would like to thank all participants and staff of the LIFE-Adult study. This work was supported by grants of the European Union, the European Regional Development Fund, the Free State of Saxony within the framework of the excellence initiative, the LIFE-Leipzig Research Center for Civilization Diseases, University of Leipzig (project numbers: 713-241202, 14505/2470, 14575/2470), and grants of the German Research Foundation, contract grant numbers 209933838 CRC1052-03 A1 (VW) and WI 3342/3-1 (VW).

## Additional information

### Funding

| Funder | Grant reference number | Author |
|---|---|---|
| Deutsche Forschungsgemeinschaft | 209933838 CRC1052-03 A1 | A Veronica Witte |
| Deutsche Forschungsgemeinschaft | WI 3342/3-1 | A Veronica Witte |

 The funders had no role in study design, data collection and interpretation, or the decision to submit the work for publication.

### Author contributions

Laurenz Lammer, Conceptualization, Data curation, Formal analysis, Visualization, Methodology, Writing - original draft, Writing – review and editing; Frauke Beyer, Conceptualization, Data curation, Methodology, Writing – review and editing; Melanie Luppa, Ronny Baber, Resources, Data curation; Christian Sanders, Christoph Engel, Resources, Data curation, Project administration; Kerstin Wirkner, Resources, Project administration; Markus Loffler, Resources, Funding acquisition, Project administration, Conceptualization; Steffi G Riedel-Heller, Resources, Data curation, Conceptualization; Arno Villringer, Resources, Conceptualization, Funding acquisition; A Veronica Witte, Conceptualization, Resources, Data curation, Supervision, Funding acquisition, Methodology, Writing – review and editing

### Author ORCIDs

Laurenz Lammer  http://orcid.org/0000-0001-5612-4924

A Veronica Witte [iD] http://orcid.org/0000-0001-9054-6688

## Ethics

Human subjects: The study was approved by the institutional ethics board of the Medical Faculty of the University of Leipzig (approval numbers 263-2009-14122009, 263/09-ff, 201/17-ek) and conducted according to the declaration of Helsinki. Informed consent to all measurements and consent to publish were obtained from all participants .

## Decision letter and Author response

Decision letter https://doi.org/10.7554/eLife.83660.sa1
Author response https://doi.org/10.7554/eLife.83660.sa2

## Additional files

### Supplementary files

• MDAR checklist

### Data availability

This study obtained access to the data from LIFE (Leipziger Forschungszentrum für Zivilisationser-krankungen) under project agreement PV-573. All data will be exclusively shared by LIFE (https://www.uniklinikum-leipzig.de/einrichtungen/life) based on individual's project proposal under data protection rules according to the University of Leipzig and can thus not be shared by the authors directly. All code used for the study is available at https://github.com/LaurenzLammer/socialisolation (copy archived at *Lammer, 2023*).

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

## Appendix 1

### Outliers

We excluded the datapoints (all measures of the timepoint) of all participants with measures deviating from the mean by 3 SD for our core variables (LSNS-score, adjusted hippocampal volume, cognitive functions). In case of TICS-score deviations by 3 SD we replaced the values with 'NA' and hence did not include them in mediation analyses.

Considering confounders, highly implausible values (±4 SD) for CES-D-score or BMI were treated as missing datapoints and we replaced them with values imputed according to our imputation plans listed below in order not to overly reduce the sample size.

All outlier analyses were conducted separately for baseline and follow-up measurements.

### Imputation

The data on the control variables education, BMI, diabetes, hypertension, age, and gender were complete or mostly complete. Henceforth, we could impute missing datapoints without inducing severe bias by using the sample mean for continuous variables or values drawn from a distribution determined by the existing data for categorical variables.

However, CES-D-scores were an exception amongst our control variables because the questionnaires often missed a single or a few items. As suggested by *Bono et al., 2007*, we imputed up to four missing items per participant using the person mean. Similarly, we imputed up to one item in the LSNS and up to six items in the TICS using the person mean.

If results from one of the cognitive tests required to calculate a composite score for a cognitive function was missing, we calculated the score based on the average performance in the remainder of available tests contributing to the composite score, if at least two tests were available.

*Appendix 1—figure 4* provides an overview of missingness in relevant variables at different LSNS scores.

### Families of tests for multiple comparison correction

The LMEs with hippocampal volume and the cognitive functions as dependent variables form one large family except for models regressing on the interaction of baseline LSNS and change in LSNS. In each family, we separately corrected model 1 and model 2 analyses resulting in two families of 12 tests. Additionally, we FDR-corrected each individual whole-brain analysis using the sided two-stage adaptive FDR-correction in the FreeSurfer-toolbox (*Bernal-Rusiel et al., 2013b*). All other analyses and the whole-brain analyses were considered to be exploratory and must be evaluated as such.

### Education

The participants' education was assessed using an extensive questionnaire and given a score ranging from 1 (no degree at all) to 7 (A-levels + master's degree [or equivalent] or promotion) according to prior research (*Lampert et al., 2013a*). The effects of education and the significance of different degrees are likely to be culture specific. Fortunately, a recent study examined the effects of education in a population of elderly residents of the city of Leipzig. In this study education operationalized as having a tertiary degree or not was found to be a significant predictor of dementia incidence (*Then et al., 2016*). This is approximated with a cut-off at a score <3.6.

### Simulation studies

Although it is sometimes claimed that Bayesian statistics do not require any multiplicity control (*Gelman et al., 2012*), we do not believe that this is the case in our study. A truly Bayesian approach would require researchers to adjust the priors to all other tests with non-independent hypotheses or datasets (*Sjölander and Vansteelandt, 2019*). This is hardly feasible and hence, in practice, Bayesian statistics are usually employed without taking all dependencies into account and their results are measured against thresholds similar to those of frequentist statistics. *Appendix 1—figure 5* shows how this results in an increasing familywise error rate (FWER) with an increasing number of tests in both Bayesian and frequentist statistics using an example from *Keysers et al., 2020*. De Jong has provided a solution for this problem for ANOVAs that has been implemented in the JASP software (*de Jong, 2019*) but there is still a great lack of available tools for researchers using other statistical methods. Henceforth, we decided to conduct a simulation study to find a Bayes factor threshold adjustment that should control our FWER similar to α-adjustments in frequentist statistics.

To find the expected number of false positives for a given number of tests and threshold, we replaced the variables for baseline social isolation and change in social isolation with random normally distributed values with the same SD and kept the original dataset otherwise untouched. Then we calculated our 24 LMEs belonging to the families of tests with the modified dataset and repeated this process 42 times. At a BF threshold of 3, 14 of the 1008 tests were false positives and 881 were detected as true negatives. *Appendix 1—figure 6* shows a histogram of the resulting Bayes factors. The study suggests that for the family size of 12 tests in our study a threshold of about 10.75 would ensure a FWER below 5%. *Appendix 1—table 25* gives an overview of the false positives and FWERs.

Furthermore, we wanted to see how this threshold adjustment would affect the power of our study. For this simulation study we generated a dataset that closely resembles the actual dataset but has different regression coefficients for baseline social isolation and change in social isolation. Instead of the actual coefficients we set the effect size per point on the LSNS to 0.1, 0.2, or 0.5 y of baseline age. We simulated a dataset and calculated a Bayes factor for each model and each effect size. As we only calculated the LMEs without interaction terms for reasons of simplicity, this resulted in a number of 48 Bayes factors from simulated data for each of our 13 runs totalling 624 tests. While our power for the smallest effect sizes was generally small (<10%), it was 85.6% for baseline social isolation with an effect size of half a year of baseline age. Increasing the threshold to 10.75 would not substantially decrease it (81.7%). *Appendix 1—tables 26 and 27* provide an overview of the percentages of false negatives and true positives using the thresholds 3 and 10.75.

## Deviations from our preregistration

For the most part, we stuck closely to our preregistered plan in this study but departed from it at some points for different reasons.

We used the function q-value instead of p.adjust for the FDR correction for the simple reason that it provides us with a more comprehensive output. As we set the argument pi to 1, q-value is equivalent to the classic procedure (*Storey, 2002*).

We originally intended to first perform a full-null model comparison using an ANOVA and only follow this up with the function drop1 in case of a significant value for the respective predictor of interest. Our intention was to avoid any multiplicity problems due to testing all predictors. Using the scope argument of drop1 solved the problem more parsimoniously.

Our plan to exclude participants with two or more lesions in their MRI was the result of an internal equivocation regarding the meaning of an abbreviation. We excluded participants based on the type of lesions but not based on lesion count.

Furthermore, we used FIML for analyses using structural equation modelling. The similar results obtained using our preregistered approach can be found in the pre-print (*Lammer et al., 2021*). Similarly, results based on R version 3.6.1 can be found there. The final analyses were conducted in R version 4.2.2 because 3.6.1 was discontinued at our institute.

Lastly, we changed from the term sex to gender as it seems more appropriate.

## Software

We performed most analyses using R (R Project for Statistical Computing, V4.2.2, RRID:SCR_001905). For the whole-brain analyses we used MATLAB (MATLAB, V9.13 (2022b) RRID:SCR_001622).

We used the package lme4 (R package: lme4, RRID:SCR_015654) to calculate LMEs in R. To obtain reliable p-values, we used the Satterthwaite option from the lmerTest package (R package: lmerTest, RRID:SCR_015656; *Kuznetsova et al., 2017*). In the whole-brain analyses we employed the MATLAB-toolbox provided by FreeSurfer to calculate vertex-wise LMEs (*Bernal-Rusiel et al., 2013a*). For mediation analyses and BLCS models we used the sem function from the lavaan package (*Rosseel, 2012*).

We calculated BFs for all LMEs in R using the BayesFactor package and the functions posterior and generalTestBF with default priors (*Rouder and Morey, 2012*).

FDR-correction was performed using the q-value function (R package: Qvalue, RRID:SCR_001073) in R and the sided two-stage adaptive FDR-correction in the FreeSurfer-toolbox (*Bernal-Rusiel et al., 2013b*).

VIFs were calculated using the package car (*Fox and Weisberg, 2019*).

## Reflexivity

Reflexivity, a sensitivity to and acknowledgement of the ways in which scientists shape the collected data and research findings, is an established hallmark of scientific rigour in qualitative research (*Mays and Pope, 2000*; *Sandelowski and Barroso, 2002*). The challenges addressed by reflexivity are perhaps more pronounced in but by no means exclusive to qualitative studies. Nevertheless (at least in an openly conducted form), it is largely absent from quantitative studies (*Ryan and Golden, 2006*). Methodological reforms in quantitative research like preregistrations and registered reports (*Nosek et al., 2018*; *Nosek and Lakens, 2014*) are valuable tools to limit the researchers' potential to make data fit their prior assumptions but their scope is limited. They do not address some of the most fundamental issues in epidemiology: Which analogies are used to make sense of the data, which questions are being raised and answered, and which theories are chosen to explain phenomena (*Krieger, 2011*)? Disclosing personal characteristics, researchers' values, and positionality relative to the object of research (*Berger, 2015*) thus helps readers assess a study and its findings more thoroughly. Additionally, an external evaluation of the presence and prevalence of non-empirical decision vectors (*Solomon, 2001*) in a field of research can be greatly facilitated. Furthermore, as Stephen J. Gould has put it: "It is dangerous for a scholar even to imagine that he might attain complete neutrality, for then one stops being vigilant about personal preferences and their influences – and then one truly falls victim to the dictates of prejudice" (*Gould, 1996*).

Henceforth, I, as the first author, want to expand this study by a brief reflection on influences that might have played a role in the formation of this study. I am a medical doctoral student with no prior experience in research and conducted this study as the centrepiece of my planned dissertation. Thus, I entered this project with little prior knowledge. I believe that this both made me more flexible and restricted in my choices. On the one hand I was not dedicated to any specific research programme or topic, but on the other hand my reliance on the advice and support from more senior researchers made me emulate their work and methods in many aspects. Further, my worldview has probably made me tend to epidemiological theories (social epidemiology, eco-social theory) (*Berkman et al., 2015*; *Krieger, 2014*) broader than the study of lifestyle factors and hence made me choose social isolation as my research topic. A further characteristic that might be of interest to readers is that during the course of the research, two of my relatives struggled with dementia. Ultimately, this reflexivity is inherently limited, as the use of secondary data precludes me from reflecting on the pivotal processes of data acquisition and participant recruitment.

## Explicit equations of all LMEs using the lme4 syntax

Variables in bold are dropped in the null model.

**H 1.1** Social isolation is negatively associated with hippocampal volume across individuals.

Model111: HCV ~**LSNS_bl** +LSNS_change +age_bl +age_change +sex + (1|subject)

Model112: HCV ~**LSNS_bl** +LSNS_change +age_bl +age_change +sex + hypertension +diabetes + BMI+CESD + education + (1|subject)

**H 1.3** Social isolation is negatively associated with hippocampal volume within individuals.

Model131: HCV ~LSNS_bl +**LSNS_change** +age_bl +age_change +sex + (1|subject)

Model132: HCV ~LSNS_bl +**LSNS_change** +age_bl +age_change +sex + hypertension +diabetes + BMI+CESD + education + (1|subject)

**H 1.5** Participants that are socially more isolated at baseline will experience aggravated age-related changes in hippocampal volume over the follow-up period.

Model151: HCV ~LSNS_bl +LSNS_change +age_bl +age_change + **LSNS_bl\*age_change** +sex + (1|subject)

Model152: HCV ~LSNS_bl +LSNS_change +age_bl +age_change + **LSNS_bl\*age_change** +sex + hypertension +diabetes + BMI+CES.D + education + (1|subject)

**H 2.1** Social isolation is negatively associated with cognitive functions across individuals.

Model211a: executive function ~**LSNS_bl** +LSNS_change +age_bl +age_change +sex + (1|subject)

Model212a: executive function ~**LSNS_bl** +LSNS_change +age_bl +age_change +sex + hypertension +diabetes + BMI+CES.D+education + (1|subject)

Model211b: memory performance ~**LSNS_bl** +LSNS_change +age_bl +age_change + sex + (1|subject)

Model212b: memory performance ~**LSNS_bl** +LSNS_change +age_bl +age_change +sex + hypertension + diabetes +BMI + CES.D+education + (1|subject)

Model211c: processing speed ~**LSNS_bl** +LSNS_change +age_bl +age_change + sex + (1|subject)

Model212c: processing speed ~**LSNS_bl** +LSNS_change +age_bl +age_change +sex + hypertension +diabetes + BMI+CES.D+education + (1|subject)

**H 2.2** Social isolation is negatively associated with cognitive functions within individuals.

Model221a: executive function ~LSNS_bl +**LSNS_change** +age_bl +age_change +sex + (1|subject)

Model222a: executive function ~LSNS_bl +**LSNS_change** +age_bl +age_change +sex + hypertension +diabetes + BMI+CES.D+education + (1|subject)

Model221b: memory performance ~LSNS_bl +**LSNS_change** +age_bl +age_change +sex + (1|subject)

Model222b: memory performance ~LSNS_bl +**LSNS_change** +age_bl +age_change +sex +hypertension + diabetes +BMI + CES.D+education + (1|subject)

Model221c: processing speed ~LSNS_bl +**LSNS_change** +age_bl +age_change + sex + (1|subject)

Model222c: processing speed ~LSNS_bl +**LSNS_change** +age_bl +age_change +sex + hypertension +diabetes + BMI+CES.D+education + (1|subject)

**H 2.3** Participants that are socially more baseline will experience aggravated age-related changes in cognitive function over the follow-up period.

Model231a: executive function ~LSNS_bl +age_bl +age_change +**LSNS_bl*age_change** +sex + (1|subject)

Model231a: executive function ~LSNS_bl +age_bl +age_change +**LSNS_bl*age_change** +sex + hypertension +diabetes + BMI+CES.D+education + (1|subject)

Model231b: memory performance ~LSNS_bl +age_bl +age_change + **LSNS_bl*age_change** +sex + (1|subject)

Model231b: memory performance ~LSNS_bl +age_bl +age_change + **LSNS_bl*age_change** +sex + hypertension +diabetes + BMI+CES.D+education + (1|subject)

Model231c: processing speed ~LSNS_bl +age_bl +age_change +**LSNS_bl*age_change** +sex + (1|subject)

Model231c: processing speed ~LSNS_bl +age_bl +age_change +**LSNS_bl*age_change** +sex + hypertension +diabetes + BMI+CES.D+education + (1|subject)

**H 5.1** In people who are socially more isolated at baseline, an increase in social isolation from baseline to follow-up will have a stronger negative association with HCV than in people who are less socially isolated at baseline.

Model511: HCV ~LSNS_bl +LSNS_change +**LSNS_bl*LSNS_change** +age_bl + age_change +sex + (1|subject)

Model512: HCV ~LSNS_bl +LSNS_change +**LSNS_bl*LSNS_change** + age_bl +age_change +sex + hypertenison +diabetes + BMI+CES.D + education + (1|subject)

## Explicit equations of all LMEs using the FreeSurfer LME syntax

**H 1.2** Social isolation is negatively associated with vertex-wise cortical thickness across individuals.

For model 1 we built a matrix consisting of six columns: intercept (all ones), age_bl, age_change, sex, LSNS_bl and LSNS_change.

The corresponding contrast matrix was [0 0 0 0 1 0].

For model 2 we built a matrix consisting of eleven columns: intercept (all ones), age_bl, age_change, sex, hypertension, diabetes, education, BMI, CES_D, LSNS_bl and LSNS_change.

The corresponding contrast matrix was [0 0 0 0 0 0 0 0 0 1 0].

**H 1.4** Social isolation is negatively associated with vertex-wise cortical thickness within individuals.

For model 1 we built a matrix consisting of six columns: intercept (all ones), age_bl, age_change, sex, LSNS_bl and LSNS_change.

The corresponding contrast matrix was [0 0 0 0 0 1].

For model 2 we built a matrix consisting of eleven columns: intercept (all ones), age_bl, age_change, sex, hypertension, diabetes, education, BMI, CES_D, LSNS_bl and LSNS_change.

The corresponding contrast matrix was [0 0 0 0 0 0 0 0 0 0 1].

**H 1.6** Participants that are socially more isolated at baseline, will experience aggravated age-related changes in cortical thickness over the follow-up period.

For model 1 we built a matrix consisting of seven columns: intercept (all ones), age_bl, age_change, sex, LSNS_bl, LSNS_change and LSNS_bl*age_change. The last term is an interaction between baseline LSNS and age_change.

The corresponding contrast matrix was [0 0 0 0 0 0 1].

For model 2 we built a matrix consisting of twelve columns: intercept (all ones), age_bl, age_change, sex, hypertension, diabetes, education, BMI, CES_D, LSNS_bl, LSNS_change and LSNS_bl*age_change. The last term is an interaction between baseline LSNS and age_change.

The corresponding contrast matrix was [0 0 0 0 0 0 0 0 0 0 0 1].

**Appendix 1—table 1.** Descriptive statistics for further variables.

| Variable | BL, N = 1992 | FU, N = 1409 |
|---|---|---|
| Married | 1,435 (72%) \| 10 | 768 (70%) \| 306 |
| Not working | 1,434 (73%) \| 20 | 887 (81%) \| 309 |
| Living alone | 426 (22%) \| 13 | 296 (27%) \| 305 |
| n in dwelling | 2 (1) \| 1 \| 11 \| 13 | 2 (1) \| 1 \| 10 \| 305 |
| SES | 12.4 (3.3) \| 4.5 \| 21.0 \| 13 | 12.6 (3.3) \| 4.5 \| 21.0 \| 8 |
| Migration background | 332 (17%) \| 10 | 232 (17%) \| 6 |
| Global PSQI | 6 (3) \| 0 \| 19 \| 558 | 5 (3) \| 0 \| 14 \| 1,110 |
| Total weekly METs | 5,910 (5,829) \| 0 \| 38,880 \| 178 | 4,587 (5,994) \| 0 \| 49,500 \| 1,055 |

Values for categorical variables: n (%) yes, | n missing; values for continuous variables: mean (SD) | minimum | maximum | n missing.

BL = baseline. FU = follow-up. not working = not gainfully employed due to other reasons than studies, military or other service. n in dwelling = number of persons living in the participants' dwelling. SES = socioeconomic status. migration background = participant or at least one of the participants parents was not born in Germany. global PSQI = global score of the Pittsburgh Sleep Quality Index. total weekly METs = total weekly metabolic equivalents based on the International Physical Activity Questionnaire

**Appendix 1—table 2.** Regression coefficients and measures of significance.

**Regression coefficients and measures of significance**

| dv | Predictor | Estimate | 95% CI | p-value | n total | n indiv |
|---|---|---|---|---|---|---|
| LSNS_sum | TICS | 0.008 | –0.005, 0.02 | 0.3652 | 933 | 794 |
| LSNS_sum | gender | 0.204 | -0.238, 0.645 | 1.0e-28**** | 3115 | 1921 |
| LSNS_sum | SES | –0.377 | –0.442,–0.312 | 5.1e-07**** | 3099 | 1910 |
| LSNS_sum | not working | 1.125 | 0.686, 1.565 | 1.2e-19**** | 2812 | 1893 |
| LSNS_sum | n_dwell | -1.605 | -1.95, -1.26 | 0.9616 | 2823 | 1899 |
| LSNS_sum | migrat | –0.015 | –0.605, 0.576 | 2.5e-17**** | 3103 | 1913 |
| LSNS_sum | married | -1.951 | -2.399, -1.502 | 9.3e-22**** | 2825 | 1900 |
| LSNS_sum | live alone | 2.337 | 1.862, 2.812 | 0.2344 | 2823 | 1899 |
| LSNS_sum | age_base | 0.084 | 0.051, 0.117 | 5.9e-07**** | 3115 | 1921 |

*Appendix 1—table 2 Continued on next page*

*Appendix 1—table 2 Continued*

**Regression coefficients and measures of significance**

| dv | Predictor | Estimate | 95% CI | p-value | n total | n indiv |
|----|-----------|----------|--------|---------|---------|---------|
| LSNS_sum | age_change | 0.046 | 0.007, 0.085 | 0.0207* | 3115 | 1921 |

* p<0.05, ** p<0.01, *** p<0.001, **** p<0.0001.

dv, dependent variable; CI, confidence interval; n total, total number of observations; n indiv, number of participants; LSNS_sum, Lubben Social Network Scale sum score; SES, socioeconomic status; not working, not gainfully employed due to other reasons than studies, military or alternative service; n dwell, number of persons living in the participants' dwelling; migrat, migration background; live alone, participants living alone; TICS, sum score of the Trierer Inventar zum chronischen Stress (chronic stress questionnaire); age_base, age at baseline; age_change, change in age from baseline to follow-up.

**Appendix 1—table 3.** Comparison of LIFE-SES data with quintiles based on fully representative sample.

| Category | Quintile | LIFE-Adult (%) |
|----------|----------|----------------|
| Low | 1 | 6.75 |
| | 2 | 15.36 |
| | 3 | 22.01 |
| Middle | 4 | 23.82 |
| High | 5 | 32.07 |

Category, socioeconomic status category; LIFE-Adult (%), percentage of participants falling into the range of the quintile cut-off points based on a representative German sample.

**Appendix 1—table 4.** Adjusted regression coefficients and measures of significance of models without interaction terms.

| dv | Model | Predictor | Estimate | 95% CI | p-value | FDR | BF | Total n | n individuals |
|----|-------|-----------|----------|--------|---------|-----|-----|---------|---------------|
| | | LSNS_base | −5.616 | −9.281,−1.951 | 0.0013** | 0.0034** | 18.95** | 1667 | 1306 |
| | | LSNS_change | −4.659 | −8.027,−1.284 | 0.0035** | 0.0069** | 2.36 | 1667 | 1306 |
| | | age_base | −26.079 | −28.932,−23.226 | | | | 1667 | 1306 |
| | | age_change | −26.836 | −29.01,−24.668 | | | | 1667 | 1306 |
| | 1 | gender | −46.504 | −83.617,−9.392 | | | | 1667 | 1306 |
| | | LSNS_base | −5.697 | −9.56,−1.833 | 0.0019** | 0.0078** | 14.93** | 1556 | 1226 |
| | | LSNS_change | -4.548 | −8.131,−0.954 | 0.0066** | 0.0158* | 2.47 | 1556 | 1226 |
| | | age_base | -24.246 | -27.292, -21.199 | | | | 1556 | 1226 |
| | | age_change | -26.831 | −29.157,−24.513 | | | | 1556 | 1226 |
| | | gender | −46.390 | −84.512,−8.269 | | | | 1556 | 1226 |
| | | BMI | 13.722 | -3.151, 30.599 | | | | 1556 | 1226 |
| | | CESD | 12.184 | -7.102, 31.47 | | | | 1556 | 1226 |
| | | diabetes | −99.613 | −152.086,−47.131 | | | | 1556 | 1226 |
| | | education | −93.450 | −155.942,−30.953 | | | | 1556 | 1226 |
| Hippo campal volume | 2 | hypertension | −21.182 | −61.555, 19.195 | | | | 1556 | 1226 |

*Appendix 1—table 4 Continued on next page*

Appendix 1—table 4 Continued

| dv | Model | Predictor | Estimate | 95% CI | p-value | FDR | BF | Total n | n individuals |
|---|---|---|---|---|---|---|---|---|---|
| | | LSNS_base | –0.028 | –0.037,–0.019 | 8.0e-10**** | 9.6e-09**** | 1.4e+07**** | 2048 | 1469 |
| | | LSNS_change | -0.003 | -0.017, 0.011 | 0.342 | 0.4104 | 0.16 | 2048 | 1469 |
| | | age_base | –0.019 | –0.026,–0.012 | | | | 2048 | 1469 |
| | | age_change | -0.050 | -0.061, -0.04 | | | | 2048 | 1469 |
| | 1 | gender | -0.084 | –0.176, 0.008 | | | | 2048 | 1469 |
| | | LSNS_base | -0.017 | -0.026, -0.007 | 2e-04**** | 0.0014** | 128.4**** | 1893 | 1377 |
| | | LSNS_change | 0.001 | –0.013, 0.016 | 0.5762 | 0.6914 | 0.13 | 1893 | 1377 |
| | | age_base | -0.014 | –0.021,–0.007 | | | | 1893 | 1377 |
| | | age_change | –0.051 | –0.061,–0.04 | | | | 1893 | 1377 |
| | | gender | -0.122 | –0.215,–0.028 | | | | 1893 | 1377 |
| | | BMI | -0.074 | –0.121,–0.028 | | | | 1893 | 1377 |
| | | CESD | -0.141 | –0.188,–0.094 | | | | 1893 | 1377 |
| | | diabetes | -0.055 | -0.184, 0.074 | | | | 1893 | 1377 |
| | | education | -0.331 | –0.489,–0.173 | | | | 1893 | 1377 |
| Executive functions | 2 | hypertension | –0.097 | -0.196, 0.002 | | | | 1893 | 1377 |
| | | LSNS_base | -0.014 | -0.022, -0.006 | 4e-04**** | 0.0016** | 58.91*** | 1921 | 1408 |
| | | LSNS_change | –0.019 | –0.032,–0.007 | 0.0014** | 0.0034** | 14.68** | 1921 | 1408 |
| | | age_base | -0.035 | –0.042,–0.029 | | | | 1921 | 1408 |
| | | age_change | –0.018 | –0.026,–0.009 | | | | 1921 | 1408 |
| | 1 | gender | –0.381 | –0.465,–0.297 | | | | 1921 | 1408 |
| | | LSNS_base | -0.008 | -0.016, 0.001 | 0.0457* | 0.0914 | 1.23 | 1777 | 1315 |
| | | LSNS_change | -0.018 | –0.031,–0.004 | 0.0047** | 0.0142* | 6.67* | 1777 | 1315 |
| | | age_base | -0.033 | –0.04,–0.026 | | | | 1777 | 1315 |
| | | age_change | -0.018 | -0.028, -0.009 | | | | 1777 | 1315 |
| | | gender | -0.421 | –0.507,–0.334 | | | | 1777 | 1315 |
| | | BMI | -0.031 | -0.074, 0.012 | | | | 1777 | 1315 |
| | | CESD | –0.120 | –0.164,–0.077 | | | | 1777 | 1315 |
| | | diabetes | –0.038 | -0.155, 0.08 | | | | 1777 | 1315 |
| | | education | -0.173 | –0.316,–0.031 | | | | 1777 | 1315 |
| Memory | 2 | hypertension | 0.017 | –0.074, 0.108 | | | | 1777 | 1315 |

Appendix 1—table 4 Continued on next page

*Appendix 1—table 4 Continued*

| dv | Model | Predictor | Estimate | 95% CI | p-value | FDR | BF | Total n | n individuals |
|---|---|---|---|---|---|---|---|---|---|
| | | LSNS_base | –0.018 | –0.026,–0.01 | 1.9e-06**** | 1.2e-05**** | 8.2e+03**** | 2043 | 1470 |
| | | LSNS_change | –0.007 | –0.019, 0.006 | 0.1585 | 0.2378 | 0.26 | 2043 | 1470 |
| | | age_base | -0.038 | -0.044, -0.032 | | | | 2043 | 1470 |
| | | age_change | -0.038 | –0.047,–0.028 | | | | 2043 | 1470 |
| | 1 | gender | -0.108 | –0.185,–0.031 | | | | 2043 | 1470 |
| | | LSNS_base | -0.017 | –0.025,–0.009 | 2.1e-05**** | 3e-04**** | 1.0e+03**** | 1888 | 1376 |
| | | LSNS_change | -0.008 | –0.021, 0.005 | 0.1253 | 0.1981 | 0.53 | 1888 | 1376 |
| | | age_base | –0.036 | –0.042,–0.029 | | | | 1888 | 1376 |
| | | age_change | –0.033 | –0.043,–0.024 | | | | 1888 | 1376 |
| | | gender | -0.124 | –0.204,–0.044 | | | | 1888 | 1376 |
| | | BMI | -0.013 | -0.053, 0.028 | | | | 1888 | 1376 |
| | | CESD | -0.027 | –0.067, 0.013 | | | | 1888 | 1376 |
| | | diabetes | –0.005 | –0.115, 0.106 | | | | 1888 | 1376 |
| | | education | –0.115 | –0.248, 0.017 | | | | 1888 | 1376 |
| Processing speed | 2 | hypertension | -0.067 | -0.151, 0.018 | | | | 1888 | 1376 |

* p<0.05, BF >3; ** p<0.01, BF >10; *** p<0.001, BF >30; **** p<0.0001, BF >100.
full model1: dv~LSNS_base+LSNS_change+age_base+age_change+gender.
full model2: model1 + hypertension+diabetes+education+BMI+CESD.
The unit of effect sizes on hippocampal volume and cognitive functions are mm³/point on the LSNS and standard deviation/point on the LSNS, respectively.
dv, dependent variable; CI, confidence interval; FDR, p-values after FDR-correction; BF, Bayes Factor in favour of alternative hypothesis; total n, total number of observations; n individuals, number of participants; LSNS_base, baseline Lubben Social Network Score; LSNS_change, change in Lubben Social Network Score; CESD, Center for Epidemiological Studies-Depression.

**Appendix 1—table 5.** Adjusted regression coefficients and measures of significance of models with interaction term of baseline social isolation with change in age.

| dv | Model | Predictor | Estimate | 95% CI | p-value | FDR | BF | Total n | n individuals |
|---|---|---|---|---|---|---|---|---|---|
| | | LSNS_base*age_change | –0.319 | –0.821, 0.182 | 0.1063 | 0.1821 | 0.16 | 1667 | 1306 |
| | | LSNS_base | –5.331 | –9.022,–1.64 | | | | 1667 | 1306 |
| | | LSNS_change | –5.570 | –9.227,–1.909 | | | | 1667 | 1306 |
| | | age_base | -26.052 | –28.905,–23.199 | | | | 1667 | 1306 |
| | | age_change | –22.528 | –29.639,–15.409 | | | | 1667 | 1306 |
| | 1 | gender | –46.176 | –83.284,–9.069 | | | | 1667 | 1306 |
| | | LSNS_base*age_change | –0.302 | -0.833, 0.228 | 0.132 | 0.1981 | 0.15 | 1556 | 1226 |
| | | LSNS_base | –5.423 | –9.315,–1.531 | | | | 1556 | 1226 |
| | | LSNS_change | –5.369 | –9.23,–1.503 | | | | 1556 | 1226 |
| | | age_base | –24.219 | –27.266,–21.173 | | | | 1556 | 1226 |
| | | age_change | -22.738 | –30.287,–15.179 | | | | 1556 | 1226 |
| | | gender | –46.042 | –84.16,–7.926 | | | | 1556 | 1226 |
| | | BMI | 13.544 | -3.326, 30.417 | | | | 1556 | 1226 |
| | | CESD | 12.301 | -6.982, 31.583 | | | | 1556 | 1226 |
| | | diabetes | –99.378 | –151.839,–46.907 | | | | 1556 | 1226 |
| | | education | -93.471 | –155.947,–30.99 | | | | 1556 | 1226 |
| Hippo campal volume | 2 | hypertension | –21.032 | –61.396, 19.335 | | | | 1556 | 1226 |

*Appendix 1—table 5 Continued on next page*

*Appendix 1—table 5 Continued*

| dv | Model | Predictor | Estimate | 95% CI | p-value | FDR | BF | Total n | n individuals |
|---|---|---|---|---|---|---|---|---|---|
| | | LSNS_base*age_change | 0.001 | −0.002, 0.003 | 0.6739 | 0.6739 | 0.08 | 2048 | 1469 |
| | | LSNS_base | −0.029 | −0.039,−0.019 | | | | 2048 | 1469 |
| | | LSNS_change | −0.002 | −0.017, 0.014 | | | | 2048 | 1469 |
| | | age_base | −0.019 | −0.026,−0.012 | | | | 2048 | 1469 |
| | | age_change | −0.057 | −0.09,−0.025 | | | | 2048 | 1469 |
| | 1 | gender | -0.084 | −0.176, 0.008 | | | | 2048 | 1469 |
| | | LSNS_base*age_change | 0.001 | -0.001, 0.003 | 0.7713 | 0.7713 | 0.08 | 1893 | 1377 |
| | | LSNS_base | −0.018 | −0.028,−0.008 | | | | 1893 | 1377 |
| | | LSNS_change | 0.004 | -0.012, 0.019 | | | | 1893 | 1377 |
| | | age_base | −0.014 | −0.021,−0.007 | | | | 1893 | 1377 |
| | | age_change | −0.063 | −0.097,−0.029 | | | | 1893 | 1377 |
| | | gender | −0.123 | −0.216,−0.029 | | | | 1893 | 1377 |
| | | BMI | -0.074 | -0.12, -0.028 | | | | 1893 | 1377 |
| | | CESD | −0.141 | −0.188,−0.094 | | | | 1893 | 1377 |
| | | diabetes | −0.056 | −0.185, 0.073 | | | | 1893 | 1377 |
| | | education | −0.331 | −0.488,−0.173 | | | | 1893 | 1377 |
| Executive functions | 2 | hypertension | -0.098 | −0.196, 0.001 | | | | 1893 | 1377 |
| | | LSNS_base*age_change | 0.000 | −0.002, 0.002 | 0.6695 | 0.6739 | 0.07 | 1921 | 1408 |
| | | LSNS_base | -0.015 | −0.023,−0.006 | | | | 1921 | 1408 |
| | | LSNS_change | -0.018 | −0.032,−0.005 | | | | 1921 | 1408 |
| | | age_base | -0.035 | −0.042,−0.029 | | | | 1921 | 1408 |
| | | age_change | -0.024 | −0.052, 0.005 | | | | 1921 | 1408 |
| | 1 | gender | −0.382 | −0.466,−0.298 | | | | 1921 | 1408 |
| | | LSNS_base*age_change | 0.001 | −0.001, 0.003 | 0.7271 | 0.7713 | 0.08 | 1777 | 1315 |
| | | LSNS_base | −0.008 | −0.018, 0.001 | | | | 1777 | 1315 |
| | | LSNS_change | −0.016 | −0.031, −0.002 | | | | 1777 | 1315 |
| | | age_base | -0.033 | −0.04, −0.026 | | | | 1777 | 1315 |
| | | age_change | -0.027 | −0.057, 0.003 | | | | 1777 | 1315 |
| | | gender | -0.421 | −0.507, −0.335 | | | | 1777 | 1315 |
| | | BMI | -0.031 | −0.074, 0.011 | | | | 1777 | 1315 |
| | | CESD | −0.120 | −0.164, −0.077 | | | | 1777 | 1315 |
| | | diabetes | −0.038 | −0.156, 0.08 | | | | 1777 | 1315 |
| | | education | −0.173 | −0.316, −0.031 | | | | 1777 | 1315 |
| Memory | 2 | hypertension | 0.017 | -0.074, 0.108 | | | | 1777 | 1315 |

*Appendix 1—table 5 Continued*

| dv | Model | Predictor | Estimate | 95% CI | p-value | FDR | BF | Total n | n individuals |
|---|---|---|---|---|---|---|---|---|---|
| | | LSNS_base*age_change | -0.001 | –0.003, 0.001 | 0.2341 | 0.3122 | 0.19 | 2043 | 1470 |
| | | LSNS_base | -0.017 | –0.025, –0.009 | | | | 2043 | 1470 |
| | | LSNS_change | –0.008 | –0.022, 0.005 | | | | 2043 | 1470 |
| | | age_base | –0.038 | –0.044, –0.032 | | | | 2043 | 1470 |
| | | age_change | –0.027 | –0.057, 0.002 | | | | 2043 | 1470 |
| | 1 | gender | –0.107 | –0.185, –0.03 | | | | 2043 | 1470 |
| | | LSNS_base*age_change | 0.000 | –0.002, 0.002 | 0.4856 | 0.6474 | 0.11 | 1888 | 1376 |
| | | LSNS_base | –0.017 | –0.026, –0.008 | | | | 1888 | 1376 |
| | | LSNS_change | –0.008 | -0.022, 0.006 | | | | 1888 | 1376 |
| | | age_base | –0.036 | –0.042, –0.029 | | | | 1888 | 1376 |
| | | age_change | –0.033 | –0.063, 0.003 | | | | 1888 | 1376 |
| | | gender | –0.124 | –0.204, –0.044 | | | | 1888 | 1376 |
| | | BMI | –0.013 | –0.053, 0.028 | | | | 1888 | 1376 |
| | | CESD | –0.027 | –0.067, 0.013 | | | | 1888 | 1376 |
| | | diabetes | –0.004 | –0.115, 0.106 | | | | 1888 | 1376 |
| | | education | –0.115 | –0.248, 0.017 | | | | 1888 | 1376 |
| Processing speed | 2 | hypertension | –0.067 | –0.151, 0.018 | | | | 1888 | 1376 |

full model1: dv~LSNS_base+LSNS_change+age_base+age_change+gender

full model2: model1 +hypertension + diabetes +education + BMI +CESD

dv = dependent variable. CI = confidence interval. FDR = p-values after FDR-correction. BF = Bayes factor in favour of alternative hypothesis. total n = total number of observations. n individuals = number of participants. LSNS_base = baseline Lubben Social Network Score. LSNS_change = change in Lubben Social Network Score. CESD = Center for Epidemiological Studies-Depression.

**Appendix 1—table 6.** Adjusted regression coefficients and measures of significance of models with interaction term of baseline social isolation with change in social isolation.

| dv | Model | Predictor | Estimate | 95% CI | p-value | BF | Total n | n individuals |
|---|---|---|---|---|---|---|---|---|
| | | LSNS_base*LSNS_change | 0.14 | -0.51, 0.79 | 0.6646 | 0.03 | 1667 | 1306 |
| | | LSNS_base | –5.62 | –9.29,–1.96 | | | 1667 | 1306 |
| | | LSNS_change | –6.51 | –15.71, 2.67 | | | 1667 | 1306 |
| | | age_base | –26.08 | –28.93,–23.23 | | | 1667 | 1306 |
| | | age_change | –26.67 | –28.97,–24.38 | | | 1667 | 1306 |
| | 1 | gender | –46.44 | –83.55,–9.33 | | | 1667 | 1306 |
| | | LSNS_base*LSNS_change | 0.18 | –0.5, 0.86 | 0.6985 | 0.05 | 1556 | 1226 |
| | | LSNS_base | –5.70 | –9.57,–1.84 | | | 1556 | 1226 |
| | | LSNS_change | –6.94 | –16.67, 2.78 | | | 1556 | 1226 |
| | | age_base | –24.25 | –27.29,–21.2 | | | 1556 | 1226 |
| | | age_change | –26.63 | –29.08,–24.19 | | | 1556 | 1226 |
| | | gender | –46.32 | –84.44,–8.21 | | | 1556 | 1226 |
| | | BMI | 13.78 | –3.1, 30.65 | | | 1556 | 1226 |
| | | CESD | 12.14 | –7.14, 31.43 | | | 1556 | 1226 |
| | | diabetes | –99.48 | –151.96,–47 | | | 1556 | 1226 |
| | | education | –93.43 | –155.91,–30.93 | | | 1556 | 1226 |
| Hippocampal volume | 2 | hypertension | -21.19 | –61.56, 19.19 | | | 1556 | 1226 |

*Appendix 1—table 6 Continued on next page*

*Appendix 1—table 6 Continued*

| dv | Model | Predictor | Estimate | 95% CI | p-value | BF | Total n | n individuals |
|---|---|---|---|---|---|---|---|---|

full model1: dv~LSNS_base+LSNS_change+age_base+age_change+gender

full model2: model1 + hypertension+diabetes+education+BMI+CESD

dv = dependent variable. CI = confidence interval. BF = Bayes factor in favour of alternative hypothesis. total n = total number of observations. n individuals = number of participants. LSNS_base = baseline Lubben Social Network Score. LSNS_change = change in Lubben Social Network Score. CESD = Center for Epidemiological Studies-Depression.

**Appendix 1—table 7.** Indirect effects of social isolation on hippocampal volume and cognitive functions.

| Mediator | dv | Model | Estimate | SE | z-value | p-value |
|---|---|---|---|---|---|---|
| TICS | Hippo campal volume | 1 | 0.0291 | 0.7414 | 0.0393 | 0.5157 |
| | | 2 | −0.2059 | 1.0916 | −0.1886 | 0.4252 |
| Hippo campal volume | Executive functions | 1 | −0.0008 | 0.0012 | −0.6561 | 0.2559 |
| | | 2 | −0.0011 | 0.0013 | −0.8066 | 0.2099 |
| | Memory | 1 | −0.0008 | 0.0012 | −0.6945 | 0.2437 |
| | | 2 | −0.0014 | 0.0015 | −0.9245 | 0.1776 |
| | Processing speed | 1 | -0.0001 | 0.0005 | −0.2469 | 0.4025 |
| | | 2 | −0.0001 | 0.0007 | −0.1648 | 0.4345 |

model1: corrected for baseline age, change in age and gender.

model2: model1 + hypertension + diabetes +education + BMI +CESD.

dv = dependent variable. SE = standard error. TICS = Trierer Inventar zum chronischen Stress (stress questionnaire).

**Appendix 1—table 8.** Adjusted regression coefficients and measures of significance of models controlling for Covid-pandemic.

| dv | Model | Predictor | Estimate | 95% CI | p-value | FDR | BF | Total n | n individuals |
|---|---|---|---|---|---|---|---|---|---|
| | | LSNS_base | −5.617 | −9.285,−1.949 | 0.0014** | 0.0027** | 19.38** | 1667 | 1306 |
| | | LSNS_change | −5.347 | −8.694,−1.992 | 9e-04**** | 0.0022** | 7.51* | 1667 | 1306 |
| | | age_base | −26.064 | −28.92,−23.209 | | | | 1667 | 1306 |
| | | age_change | −24.528 | −27.14,−21.93 | | | | 1667 | 1306 |
| | | pandemic | −48.811 | −80.321,−17.156 | | | | 1667 | 1306 |
| | 1 | gender | −45.254 | −82.41,−8.098 | | | | 1667 | 1306 |
| | | LSNS_base | −5.704 | −9.57,−1.837 | 0.0019** | 0.0048** | 16.91** | 1556 | 1226 |
| | | LSNS_change | −5.268 | −8.836,−1.687 | 0.002** | 0.0048** | 7.01* | 1556 | 1226 |
| | | age_base | -24.223 | −27.272,−21.174 | | | | 1556 | 1226 |
| | | age_change | −24.606 | −27.382,−21.847 | | | | 1556 | 1226 |
| | | pandemic | −48.519 | −82.488,−14.388 | | | | 1556 | 1226 |
| | | gender | -44.976 | −83.143,−6.809 | | | | 1556 | 1226 |
| | | BMI | 13.485 | −3.334, 30.308 | | | | 1556 | 1226 |
| | | CESD | 12.444 | −6.859, 31.747 | | | | 1556 | 1226 |
| | | diabetes | −99.503 | −152.012,−46.984 | | | | 1556 | 1226 |
| | | education | −93.726 | −156.267,−31.179 | | | | 1556 | 1226 |
| Hippocampal volume | 2 | hypertension | −21.309 | −61.711, 19.097 | | | | 1556 | 1226 |

*Appendix 1—table 8 Continued on next page*

*Appendix 1—table 8 Continued*

| dv | Model | Predictor | Estimate | 95% CI | p-value | FDR | BF | Total n | n individuals |
|---|---|---|---|---|---|---|---|---|---|
| | | LSNS_base | -0.028 | −0.037,−0.019 | 7.0e-10**** | 8.4e-09**** | 1.6e+07**** | 2048 | 1469 |
| | | LSNS_change | −0.002 | −0.016, 0.012 | 0.395 | 0.474 | 0.16 | 2048 | 1469 |
| | | age_base | −0.019 | −0.026,−0.012 | | | | 2048 | 1469 |
| | | age_change | −0.057 | −0.069,−0.044 | | | | 2048 | 1469 |
| | | pandemic | 0.116 | -0.013, 0.246 | | | | 2048 | 1469 |
| | 1 | gender | −0.086 | -0.178, 0.006 | | | | 2048 | 1469 |
| | | LSNS_base | −0.017 | −0.027,−0.008 | 2e-04**** | 0.0013** | 143.45**** | 1893 | 1377 |
| | | LSNS_change | 0.002 | −0.013, 0.017 | 0.6154 | 0.7385 | 0.14 | 1893 | 1377 |
| | | age_base | −0.014 | −0.021,−0.007 | | | | 1893 | 1377 |
| | | age_change | -0.055 | −0.068,−0.043 | | | | 1893 | 1377 |
| | | pandemic | 0.091 | −0.044, 0.227 | | | | 1893 | 1377 |
| | | gender | −0.124 | −0.217,−0.03 | | | | 1893 | 1377 |
| | | BMI | −0.074 | −0.121,−0.028 | | | | 1893 | 1377 |
| | | CESD | −0.141 | −0.188,−0.095 | | | | 1893 | 1377 |
| | | diabetes | −0.055 | −0.184, 0.074 | | | | 1893 | 1377 |
| | | education | -0.330 | −0.488,−0.172 | | | | 1893 | 1377 |
| Executive functions | 2 | hypertension | −0.096 | −0.195, 0.002 | | | | 1893 | 1377 |
| | | LSNS_base | −0.014 | −0.022,−0.006 | 5e-04**** | 0.0015** | 51.07*** | 1921 | 1408 |
| | | LSNS_change | −0.021 | −0.034,−0.009 | 5e-04**** | 0.0015** | 43.05*** | 1921 | 1408 |
| | | age_base | −0.035 | −0.042,−0.029 | | | | 1921 | 1408 |
| | | age_change | −0.006 | -0.017, 0.004 | | | | 1921 | 1408 |
| | | pandemic | −0.222 | −0.339,−0.103 | | | | 1921 | 1408 |
| | 1 | gender | −0.376 | −0.46,−0.292 | | | | 1921 | 1408 |
| | | LSNS_base | −0.007 | −0.016, 0.001 | 0.0494* | 0.0987 | 1.33 | 1777 | 1315 |
| | | LSNS_change | −0.020 | −0.034,−0.007 | 0.0016** | 0.0048** | 16.7** | 1777 | 1315 |
| | | age_base | −0.033 | −0.04,−0.026 | | | | 1777 | 1315 |
| | | age_change | −0.006 | −0.017, 0.005 | | | | 1777 | 1315 |
| | | pandemic | −0.247 | −0.372,−0.122 | | | | 1777 | 1315 |
| | | gender | −0.415 | −0.501,−0.328 | | | | 1777 | 1315 |
| | | BMI | −0.031 | −0.074, 0.011 | | | | 1777 | 1315 |
| | | CESD | -0.119 | −0.162,−0.075 | | | | 1777 | 1315 |
| | | diabetes | −0.037 | −0.155, 0.081 | | | | 1777 | 1315 |
| | | education | −0.176 | −0.318,−0.034 | | | | 1777 | 1315 |
| Memory | 2 | hypertension | 0.015 | −0.076, 0.106 | | | | 1777 | 1315 |

*Appendix 1—table 8 Continued on next page*

Appendix 1—table 8 Continued

| dv | Model | Predictor | Estimate | 95% CI | p-value | FDR | BF | Total n | n individuals |
|---|---|---|---|---|---|---|---|---|---|
| | | LSNS_base | −0.018 | −0.026,−0.01 | 2.0e-06**** | 1.2e-05**** | 8.6e+03**** | 2043 | 1470 |
| | | LSNS_change | -0.007 | −0.019, 0.006 | 0.1576 | 0.2364 | 0.27 | 2043 | 1470 |
| | | age_base | −0.038 | −0.044,−0.032 | | | | 2043 | 1470 |
| | | age_change | −0.037 | −0.049,−0.026 | | | | 2043 | 1470 |
| | | pandemic | −0.005 | −0.123, 0.112 | | | | 2043 | 1470 |
| | 1 | gender | −0.108 | −0.185,−0.03 | | | | 2043 | 1470 |
| | | LSNS_base | −0.017 | −0.025,−0.009 | 2.1e-05**** | 3e-04**** | 1.1e+03**** | 1888 | 1376 |
| | | LSNS_change | -0.008 | −0.021, 0.006 | 0.1276 | 0.1914 | 0.41 | 1888 | 1376 |
| | | age_base | −0.036 | −0.042,−0.029 | | | | 1888 | 1376 |
| | | age_change | −0.034 | −0.045,−0.022 | | | | 1888 | 1376 |
| | | pandemic | 0.009 | −0.112, 0.13 | | | | 1888 | 1376 |
| | | gender | −0.124 | −0.205,−0.044 | | | | 1888 | 1,376 |
| | | BMI | −0.013 | −0.053, 0.028 | | | | 1888 | 1376 |
| | | CESD | -0.027 | −0.067, 0.013 | | | | 1888 | 1,376 |
| | | diabetes | −0.005 | −0.115, 0.106 | | | | 1888 | 1376 |
| | | education | −0.115 | −0.248, 0.017 | | | | 1888 | 1376 |
| Processing speed | 2 | hypertension | −0.066 | −0.151, 0.018 | | | | 1888 | 1376 |

full model1: dv~LSNS_base +LSNS_change +age_base +age_change +gender.
full model2: model1 +hypertension + diabetes +education + BMI +CESD.
The unit of effect sizes on hippocampal volume and cognitive functions are mm³/point on the LSNS and standard deviation/point on the LSNS, respectively.
* p<0.05, BF >3; ** p<0.01, BF >10; *** p<0.001, BF >30; **** p<0.0001, BF >100.
dv = dependent variable. CI = confidence interval. FDR = p-values after FDR-correction. BF = Bayes factor in favour of alternative hypothesis. total n = total number of observations. n individuals = number of participants. LSNS_base = baseline Lubben Social Network Score. LSNS_change = change in Lubben Social Network Score. pandemic = LSNS measurement taking place after initiation of lockdown measures in Germany. CESD = Center for Epidemiological Studies-Depression.

**Appendix 1—table 9.** Adjusted regression coefficients and measures of significance of models using fewer exclusion criteria.

| dv | Model | Predictor | Estimate | 95% CI | p-value | FDR | BF | Total n | n individuals |
|---|---|---|---|---|---|---|---|---|---|
| | | LSNS_base | −5.362 | −9.037,−1.687 | 0.0021** | 0.0051** | 12.35** | 1667 | 1309 |
| | | LSNS_change | −4.206 | −7.616,−0.788 | 0.008** | 0.0137* | 1.17 | 1667 | 1309 |
| | | age_base | -26.194 | −29.056,−23.333 | | | | 1667 | 1309 |
| | | age_change | −26.655 | −28.851,−24.465 | | | | 1667 | 1309 |
| | 1 | gender | −46.466 | −83.665,−9.267 | | | | 1667 | 1309 |
| | | LSNS_base | −5.428 | −9.315,−1.541 | 0.0031** | 0.0094** | 10.11** | 1556 | 1232 |
| | | LSNS_change | −4.368 | −8.04,−0.686 | 0.01* | 0.0201* | 1.6 | 1556 | 1232 |
| | | age_base | −24.542 | −27.6,−21.486 | | | | 1556 | 1232 |
| | | age_change | −26.819 | −29.198,−24.45 | | | | 1556 | 1232 |
| | | gender | −48.740 | −86.973,−10.506 | | | | 1556 | 1232 |
| | | BMI | 10.301 | -6.446, 27.053 | | | | 1556 | 1232 |
| | | CESD | 12.673 | −6.798, 32.143 | | | | 1556 | 1232 |
| | | diabetes | −96.009 | −148.633,−43.372 | | | | 1556 | 1232 |
| | | education | −90.854 | −153.471,−28.23 | | | | 1556 | 1232 |
| Hippo campal volume | 2 | hypertension | −24.435 | -64.91, 16.046 | | | | 1556 | 1232 |

*Appendix 1—table 9 Continued on next page*

*Appendix 1—table 9 Continued*

| dv | Model | Predictor | Estimate | 95% CI | p-value | FDR | BF | Total n | n individuals |
|---|---|---|---|---|---|---|---|---|---|
| | | LSNS_base | –0.030 | –0.038, -0.022 | 2.4e-13**** | 2.8e-12**** | 3.4e+10**** | 2551 | 1756 |
| | | LSNS_change | –0.009 | –0.021, 0.003 | 0.0776 | 0.1146 | 0.49 | 2551 | 1756 |
| | | age_base | –0.018 | –0.024,–0.012 | | | | 2551 | 1756 |
| | | age_change | -0.051 | –0.059,–0.042 | | | | 2551 | 1756 |
| | 1 | gender | -0.112 | –0.194,–0.03 | | | | 2551 | 1756 |
| | | LSNS_base | –0.019 | –0.027,–0.011 | 5.6e-06**** | 6.7e-05**** | 3.8e+03**** | 2365 | 1649 |
| | | LSNS_change | -0.005 | -0.018, 0.007 | 0.198 | 0.264 | 0.3 | 2365 | 1649 |
| | | age_base | –0.011 | –0.018,–0.005 | | | | 2365 | 1649 |
| | | age_change | –0.052 | –0.061,–0.043 | | | | 2365 | 1649 |
| | | gender | –0.160 | –0.244,–0.077 | | | | 2365 | 1649 |
| | | BMI | –0.052 | –0.093,–0.01 | | | | 2365 | 1649 |
| | | CESD | -0.137 | –0.179,–0.095 | | | | 2365 | 1649 |
| | | diabetes | -0.040 | -0.153, 0.073 | | | | 2365 | 1649 |
| | | education | -0.315 | -0.45, -0.18 | | | | 2365 | 1649 |
| Executive functions | 2 | hypertension | -0.102 | -0.191, -0.014 | | | | 2365 | 1649 |
| | | LSNS_base | –0.014 | –0.021,–0.006 | 2e-04**** | 7e-04**** | 120.88**** | 2418 | 1697 |
| | | LSNS_change | –0.016 | –0.027,–0.005 | 0.0017** | 0.0051** | 10.58** | 2418 | 1697 |
| | | age_base | –0.036 | –0.042,–0.03 | | | | 2418 | 1697 |
| | | age_change | –0.024 | –0.031,–0.016 | | | | 2418 | 1697 |
| | 1 | gender | –0.383 | –0.459,–0.306 | | | | 2418 | 1697 |
| | | LSNS_base | -0.006 | –0.014, 0.002 | 0.0818 | 0.1403 | 0.68 | 2246 | 1590 |
| | | LSNS_change | -0.015 | –0.026,–0.003 | 0.0057** | 0.0136* | 5.02* | 2246 | 1590 |
| | | age_base | –0.033 | –0.04,–0.027 | | | | 2246 | 1590 |
| | | age_change | –0.025 | –0.034,–0.017 | | | | 2246 | 1590 |
| | | gender | –0.429 | –0.508,–0.351 | | | | 2246 | 1590 |
| | | BMI | –0.030 | -0.068, 0.009 | | | | 2246 | 1590 |
| | | CESD | –0.133 | –0.172,–0.093 | | | | 2246 | 1590 |
| | | diabetes | –0.067 | –0.172, 0.038 | | | | 2246 | 1590 |
| | | education | -0.148 | –0.272,–0.023 | | | | 2246 | 1590 |
| Memory | 2 | hypertension | 0.032 | –0.052, 0.115 | | | | 2246 | 1590 |

*Appendix 1—table 9 Continued on next page*

*Appendix 1—table 9 Continued*

| dv | Model | Predictor | Estimate | 95% CI | p-value | FDR | BF | Total n | n individuals |
|---|---|---|---|---|---|---|---|---|---|
| | | LSNS_base | –0.016 | –0.023,–0.009 | 1.9e-06**** | 1.1e-05**** | 7.7e+03**** | 2552 | 1749 |
| | | LSNS_change | -0.014 | –0.025,–0.003 | 0.0047** | 0.0093** | 4.61* | 2552 | 1749 |
| | | age_base | –0.038 | –0.044,–0.033 | | | | 2552 | 1749 |
| | | age_change | –0.037 | –0.045,–0.029 | | | | 2552 | 1749 |
| | 1 | gender | –0.096 | –0.163,–0.028 | | | | 2552 | 1749 |
| | | LSNS_base | –0.013 | –0.02,–0.006 | 2e-04**** | 0.0011** | 162.63**** | 2366 | 1640 |
| | | LSNS_change | –0.016 | –0.027,–0.005 | 0.0023** | 0.0092** | 10.97** | 2366 | 1640 |
| | | age_base | –0.036 | –0.041,–0.03 | | | | 2366 | 1640 |
| | | age_change | –0.036 | –0.044,–0.028 | | | | 2366 | 1640 |
| | | gender | –0.116 | –0.186,–0.045 | | | | 2366 | 1640 |
| | | BMI | –0.021 | –0.056, 0.014 | | | | 2366 | 1640 |
| | | CESD | –0.037 | –0.073,–0.002 | | | | 2366 | 1640 |
| | | diabetes | –0.025 | –0.121, 0.07 | | | | 2366 | 1640 |
| | | education | –0.172 | –0.284,–0.06 | | | | 2366 | 1640 |
| Processing speed | 2 | hypertension | –0.049 | –0.123, 0.026 | | | | 2366 | 1640 |

full model1: dv~LSNS_base +LSNS_change +age_base +age_change +gender.

full model2: model1 +hypertension + diabetes +education + BMI +CESD.

The unit of effect sizes on hippocampal volume and cognitive functions are mm³/point on the LSNS and standard deviation/point on the LSNS, respectively.

* p<0.05, BF >3; ** p<0.01, BF >10; *** p<0.001, BF >30; **** p<0.0001, BF >100.

dv = dependent variable. CI = confidence interval. FDR = p-values after FDR-correction. BF = Bayes factor in favour of alternative hypothesis. total n = total number of observations. n individuals = number of participants. LSNS_base = baseline Lubben Social Network Score. LSNS_change = change in Lubben Social Network Score. CESD = Center for Epidemiological Studies-Depression.

**Appendix 1—table 10.** Adjusted regression coefficients and measures of significance of models including only participants with data for both timepoints.

| dv | Model | Predictor | Estimate | 95% CI | p-value | FDR | BF | Total n | n individuals |
|---|---|---|---|---|---|---|---|---|---|
| | | mean LSNS | –7.344 | –11.754,–2.934 | 6e-04**** | 0.0022** | 40.15*** | 1281 | 920 |
| | | LSNS within | –4.584 | –7.955,–1.207 | 0.0039** | 0.0118* | 2.86 | 1281 | 920 |
| | | mean age | –26.691 | –30.087,–23.295 | | | | 1281 | 920 |
| | | age within | –26.052 | –28.219,–23.877 | | | | 1281 | 920 |
| | 1 | gender | –41.688 | –84.828, 1.452 | | | | 1281 | 920 |
| | | mean LSNS | –7.216 | –11.815,–2.617 | 0.0011** | 0.0064** | 26.85** | 1210 | 880 |
| | | LSNS within | –4.401 | –7.981,–0.807 | 0.0082** | 0.0245* | 2.1 | 1210 | 880 |
| | | mean age | –25.319 | –28.9,–21.739 | | | | 1210 | 880 |
| | | age within | –25.990 | –28.309,–23.662 | | | | 1210 | 880 |
| | | gender | –41.693 | –85.905, 2.519 | | | | 1210 | 880 |
| | | BMI | 11.325 | –7.727, 30.38 | | | | 1210 | 880 |
| | | CESD | 4.825 | –17.285, 26.933 | | | | 1,210 | 880 |
| | | diabetes | –94.690 | –157.099,–32.268 | | | | 1,210 | 880 |
| | | education | –83.335 | –159.602,–7.057 | | | | 1,210 | 880 |
| Hippo campal volume | 2 | hypertension | –23.297 | –69.476, 22.882 | | | | 1,210 | 880 |

*Appendix 1—table 10 Continued on next page*

*Appendix 1—table 10 Continued*

| dv | Model | Predictor | Estimate | 95% CI | p-value | FDR | BF | Total n | n individuals |
|---|---|---|---|---|---|---|---|---|---|
| | | mean LSNS | −0.029 | −0.039,−0.018 | 8.4e-08**** | 1.0e-06**** | 1.8e+05**** | 1,631 | 1,052 |
| | | LSNS within | 0.000 | −0.016, 0.015 | 0.4988 | 0.54 | 0.11 | 1,631 | 1,052 |
| | | mean age | −0.014 | −0.022,−0.006 | | | | 1,631 | 1,052 |
| | | age within | −0.053 | −0.064,−0.043 | | | | 1,631 | 1,052 |
| | 1 | gender | −0.004 | −0.109, 0.102 | | | | 1,631 | 1,052 |
| | | mean LSNS | −0.018 | −0.029,−0.007 | 6e-04**** | 0.0064** | 55.52*** | 1,518 | 1,002 |
| | | LSNS within | 0.002 | −0.014, 0.018 | 0.5779 | 0.6304 | 0.11 | 1,518 | 1,002 |
| | | mean age | −0.007 | −0.016, 0.001 | | | | 1,518 | 1,002 |
| | | age within | −0.052 | −0.063,−0.041 | | | | 1,518 | 1,002 |
| | | gender | −0.051 | −0.158, 0.057 | | | | 1,518 | 1,002 |
| | | BMI | −0.059 | −0.113,−0.005 | | | | 1,518 | 1,002 |
| | | CESD | −0.147 | −0.2,−0.094 | | | | 1,518 | 1,002 |
| | | diabetes | −0.169 | −0.321,−0.018 | | | | 1,518 | 1,002 |
| | | education | −0.290 | −0.477,−0.103 | | | | 1,518 | 1,002 |
| Executive functions | 2 | hypertension | −0.123 | −0.235,−0.011 | | | | 1,518 | 1,002 |
| | | mean LSNS | −0.009 | −0.019, 0 | 0.0292* | 0.0585 | 1.63 | 1,539 | 1,026 |
| | | LSNS within | −0.015 | −0.029,−0.002 | 0.0147* | 0.0353* | 1.89 | 1,539 | 1,026 |
| | | mean age | −0.031 | −0.039,−0.024 | | | | 1,539 | 1,026 |
| | | age within | −0.018 | −0.027,−0.009 | | | | 1,539 | 1,026 |
| | 1 | gender | −0.351 | −0.446,−0.256 | | | | 1,539 | 1,026 |
| | | mean LSNS | −0.005 | −0.015, 0.005 | 0.1807 | 0.3082 | 0.52 | 1,436 | 974 |
| | | LSNS within | −0.013 | −0.027, 0.001 | 0.0365* | 0.0877 | 1.15 | 1,436 | 974 |
| | | mean age | −0.027 | −0.035,−0.02 | | | | 1,436 | 974 |
| | | age within | −0.017 | −0.027,−0.008 | | | | 1,436 | 974 |
| | | gender | −0.397 | −0.493,−0.301 | | | | 1,436 | 974 |
| | | BMI | 0.005 | −0.044, 0.053 | | | | 1,436 | 974 |
| | | CESD | −0.122 | −0.17,−0.075 | | | | 1,436 | 974 |
| | | diabetes | −0.079 | −0.215, 0.056 | | | | 1,436 | 974 |
| | | education | −0.166 | −0.333, 0.001 | | | | 1,436 | 974 |
| Memory | 2 | hypertension | 0.006 | −0.094, 0.107 | | | | 1,436 | 974 |

*Appendix 1—table 10 Continued on next page*

*Appendix 1—table 10 Continued*

| dv | Model | Predictor | Estimate | 95% CI | p-value | FDR | BF | Total n | n individuals |
|---|---|---|---|---|---|---|---|---|---|
| | | mean LSNS | –0.016 | –0.025,–0.007 | 3e-04**** | 0.0017** | 88.66*** | 1,625 | 1,052 |
| | | LSNS within | –0.005 | –0.019, 0.01 | 0.2628 | 0.3505 | 0.17 | 1,625 | 1,052 |
| | | mean age | –0.039 | –0.046,–0.032 | | | | 1,625 | 1,052 |
| | | age within | –0.038 | –0.047,–0.028 | | | | 1,625 | 1,052 |
| | 1 | gender | –0.100 | –0.191,–0.009 | | | | 1,625 | 1,052 |
| | | mean LSNS | –0.014 | –0.024,–0.005 | 0.0017** | 0.0066** | 23.31** | 1,513 | 1,001 |
| | | LSNS within | –0.006 | –0.021, 0.008 | 0.2055 | 0.3082 | 0.35 | 1,513 | 1,001 |
| | | mean age | –0.038 | –0.045,–0.03 | | | | 1,513 | 1,001 |
| | | age within | –0.033 | –0.043,–0.023 | | | | 1,513 | 1,001 |
| | | gender | –0.115 | –0.209,–0.021 | | | | 1,513 | 1,001 |
| | | BMI | –0.016 | –0.064, 0.032 | | | | 1,513 | 1,001 |
| | | CESD | –0.039 | –0.085, 0.007 | | | | 1,513 | 1,001 |
| | | diabetes | 0.041 | –0.091, 0.173 | | | | 1,513 | 1,001 |
| | | education | 0.033 | –0.13, 0.196 | | | | 1,513 | 1,001 |
| Processing speed | 2 | hypertension | –0.070 | –0.168, 0.028 | | | | 1513 | 1,001 |

full model1: dv~LSNS_base+LSNS_change+age_base+age_change+gender
full model2: model1 + hypertension+diabetes+education+BMI+CESD
The unit of effect sizes on hippocampal volume and cognitive functions are mm³/point on the LSNS and standard deviation/point on the LSNS, respectively.
* p<0.05, BF >3; ** p<0.01, BF >10; *** p<0.001, BF >30; **** p<0.0001, BF >100.
dv = dependent variable. CI = confidence interval. FDR = p-values after FDR-correction. BF = Bayes factor in favour of alternative hypothesis. total n = total number of observations. n individuals = number of participants. LSNS_base = baseline Lubben Social Network Score. LSNS_change = change in Lubben Social Network Score. CESD = Center for Epidemiological Studies-Depression.

**Appendix 1—table 11.** Adjusted regression coefficients and measures of significance of models using a systolic blood pressure of 140 as a cut-off for hypertension.

| dv | Model | Predictor | Estimate | 95% CI | p-value | FDR | BF | Total n | n individuals |
|---|---|---|---|---|---|---|---|---|---|
| | | LSNS_base | –5.616 | –9.281,–1.951 | 0.0013** | 0.0034** | 18.34** | 1667 | 1306 |
| | | LSNS_change | –4.659 | –8.027,–1.284 | 0.0035** | 0.0069** | 2.3 | 1667 | 1306 |
| | | age_base | -26.079 | -28.932, -23.226 | | | | 1667 | 1306 |
| | | age_change | -26.836 | -29.01, -24.668 | | | | 1667 | 1306 |
| | 1 | gender | -46.504 | -83.617, -9.392 | | | | 1667 | 1306 |
| | | LSNS_base | –5.814 | –9.679,–1.948 | 0.0016** | 0.0065** | 19.34** | 1556 | 1226 |
| | | LSNS_change | –4.564 | –8.148,–0.971 | 0.0064** | 0.0154* | 2.69 | 1556 | 1226 |
| | | age_base | –24.531 | –27.577,–21.486 | | | | 1556 | 1226 |
| | | age_change | –26.822 | –29.148,–24.504 | | | | 1556 | 1226 |
| | | gender | –46.179 | –84.385,–7.974 | | | | 1556 | 1226 |
| | | BMI | 12.505 | –4.264, 29.277 | | | | 1556 | 1226 |
| | | CESD | 11.135 | –8.105, 30.375 | | | | 1556 | 1226 |
| | | diabetes | –101.939 | –154.404,–49.466 | | | | 1556 | 1226 |
| Hippo campal volume | | education | –83.250 | –145.334,–21.16 | | | | 1556 | 1226 |
| | 2 | hypertension | –9.769 | -51.307, 31.773 | | | | 1556 | 1226 |

*Appendix 1—table 11 Continued on next page*

*Appendix 1—table 11 Continued*

| dv | Model | Predictor | Estimate | 95% CI | p-value | FDR | BF | Total n | n individuals |
|---|---|---|---|---|---|---|---|---|---|
| | | LSNS_base | −0.028 | −0.037,−0.019 | 8.0e-10**** | 9.6e-09**** | 1.4e+07**** | 2048 | 1469 |
| | | LSNS_change | −0.003 | −0.017, 0.011 | 0.342 | 0.4104 | 0.16 | 2048 | 1469 |
| | | age_base | −0.019 | −0.026,−0.012 | | | | 2048 | 1469 |
| | | age_change | -0.050 | −0.061,−0.04 | | | | 2048 | 1469 |
| | 1 | gender | -0.084 | −0.176, 0.008 | | | | 2048 | 1469 |
| | | LSNS_base | -0.017 | −0.027,−0.008 | 2e-04**** | 0.0013** | 132.13**** | 1893 | 1377 |
| | | LSNS_change | 0.001 | -0.013, 0.016 | 0.574 | 0.6888 | 0.12 | 1893 | 1377 |
| | | age_base | −0.013 | −0.021,−0.006 | | | | 1893 | 1377 |
| | | age_change | −0.051 | −0.062,−0.04 | | | | 1893 | 1377 |
| | | gender | −0.119 | −0.212,−0.025 | | | | 1893 | 1377 |
| | | BMI | −0.074 | −0.12,−0.028 | | | | 1893 | 1377 |
| | | CESD | −0.142 | −0.189,−0.095 | | | | 1893 | 1377 |
| | | diabetes | −0.052 | -0.181, 0.077 | | | | 1893 | 1377 |
| | | education | −0.354 | −0.51,−0.198 | | | | 1893 | 1377 |
| Executive functions | 2 | hypertension | -0.127 | −0.229,−0.025 | | | | 1893 | 1377 |
| | | LSNS_base | −0.014 | −0.022,−0.006 | 4e-04**** | 0.0016** | 58.04*** | 1921 | 1408 |
| | | LSNS_change | -0.019 | -0.032, -0.007 | 0.0014** | 0.0034** | 14.76** | 1921 | 1408 |
| | | age_base | −0.035 | −0.042,−0.029 | | | | 1921 | 1408 |
| | | age_change | -0.018 | −0.026,−0.009 | | | | 1921 | 1408 |
| | 1 | gender | −0.381 | −0.465,−0.297 | | | | 1921 | 1408 |
| | | LSNS_base | −0.007 | −0.016, 0.001 | 0.0475* | 0.0951 | 1.18 | 1777 | 1315 |
| | | LSNS_change | -0.018 | -0.031, -0.004 | 0.0047** | 0.0141* | 6.8* | 1777 | 1315 |
| | | age_base | −0.032 | −0.039,−0.025 | | | | 1777 | 1315 |
| | | age_change | -0.018 | −0.028,−0.009 | | | | 1777 | 1315 |
| | | gender | −0.420 | −0.507,−0.334 | | | | 1777 | 1315 |
| | | BMI | −0.029 | −0.071, 0.014 | | | | 1777 | 1315 |
| | | CESD | −0.119 | −0.162,−0.075 | | | | 1777 | 1315 |
| | | diabetes | -0.033 | -0.151, 0.084 | | | | 1777 | 1315 |
| | | education | -0.198 | -0.339, -0.057 | | | | 1777 | 1315 |
| Memory | 2 | hypertension | −0.015 | -0.109, 0.079 | | | | 1777 | 1315 |

*Appendix 1—table 11 Continued on next page*

*Appendix 1—table 11 Continued*

| dv | Model | Predictor | Estimate | 95% CI | p-value | FDR | BF | Total n | n individuals |
|---|---|---|---|---|---|---|---|---|---|
| | | LSNS_base | –0.018 | –0.026,–0.01 | 1.9e-06**** | 1.2e-05**** | 8.1e+03**** | 2043 | 1470 |
| | | LSNS_change | –0.007 | -0.019, 0.006 | 0.1585 | 0.2378 | 0.25 | 2043 | 1470 |
| | | age_base | –0.038 | –0.044,–0.032 | | | | 2043 | 1470 |
| | | age_change | –0.038 | –0.047,–0.028 | | | | 2043 | 1470 |
| | 1 | gender | –0.108 | –0.185,–0.031 | | | | 2043 | 1470 |
| | | LSNS_base | –0.017 | –0.025,–0.009 | 1.9e-05**** | 2e-04**** | 1.1e+03**** | 1888 | 1376 |
| | | LSNS_change | –0.008 | –0.021, 0.005 | 0.1238 | 0.1973 | 0.48 | 1888 | 1376 |
| | | age_base | –0.037 | –0.043,–0.03 | | | | 1888 | 1376 |
| | | age_change | –0.033 | –0.043,–0.024 | | | | 1888 | 1376 |
| | | gender | –0.125 | –0.206,–0.045 | | | | 1888 | 1376 |
| | | BMI | –0.018 | -0.057, 0.022 | | | | 1888 | 1376 |
| | | CESD | –0.030 | -0.07, 0.01 | | | | 1888 | 1376 |
| | | diabetes | –0.011 | -0.121, 0.099 | | | | 1888 | 1376 |
| | | education | –0.105 | -0.237, 0.026 | | | | 1888 | 1376 |
| Processing speed | 2 | hypertension | –0.019 | –0.106, 0.069 | | | | 1888 | 1376 |

\* p<0.05, BF >3; \*\* p<0.01, BF >10; \*\*\* p<0.001, BF >30; \*\*\*\* p<0.0001, BF >100.

full model1: dv~LSNS_base+LSNS_change+age_base+age_change+gender
full model2: model1 + hypertension+diabetes+education+BMI+CESD
The unit of effect sizes on hippocampal volume and cognitive functions are mm³/point on the LSNS and standard deviation/point on the LSNS, respectively.
dv = dependent variable. CI = confidence interval. FDR = p-values after FDR-correction. BF = Bayes factor in favour of alternative hypothesis. total n = total number of observations. n individuals = number of participants. LSNS_base = baseline Lubben Social Network Score. LSNS_change = change in Lubben Social Network Score. CESD = Center for Epidemiological Studies-Depression.

**Appendix 1—table 12.** Adjusted regression coefficients and measures of significance of models using an MMST score of 27 as cut-off for inclusion.

| dv | Model | Predictor | Estimate | 95% CI | p-value | FDR | BF | Total n | n individuals |
|---|---|---|---|---|---|---|---|---|---|
| | | LSNS_base | -6.811 | -10.548, -3.075 | 2e-04**** | 7e-04**** | 123.9**** | 1579 | 1235 |
| | | LSNS_change | –4.755 | –8.191,–1.314 | 0.0034** | 0.0069** | 2.91 | 1579 | 1235 |
| | | age_base | -25.045 | –27.923,–22.167 | | | | 1579 | 1235 |
| | | age_change | –26.913 | –29.146,–24.689 | | | | 1579 | 1235 |
| | 1 | gender | -44.363 | –81.921,–6.805 | | | | 1579 | 1235 |
| | | LSNS_base | –6.992 | –10.919,–3.064 | 2e-04**** | 0.0017** | 121.62**** | 1476 | 1159 |
| | | LSNS_change | –4.563 | –8.184,–0.933 | 0.0069** | 0.0173* | 3.04* | 1476 | 1159 |
| | | age_base | –23.204 | –26.261,–20.147 | | | | 1476 | 1159 |
| | | age_change | –26.882 | –29.251,–24.525 | | | | 1476 | 1159 |
| | | gender | –42.353 | –80.94,–3.767 | | | | 1476 | 1159 |
| | | BMI | 15.149 | –2.009, 32.309 | | | | 1476 | 1159 |
| | | CESD | 13.511 | –6.205, 33.227 | | | | 1476 | 1159 |
| | | diabetes | –101.214 | –154.93,–47.492 | | | | 1476 | 1159 |
| Hippocampal volume | | education | –90.005 | –154.773,–25.23 | | | | 1476 | 1159 |
| | 2 | hypertension | –27.620 | –68.265, 13.025 | | | | 1476 | 1159 |

*Appendix 1—table 12 Continued on next page*

*Appendix 1—table 12 Continued*

| dv | Model | Predictor | Estimate | 95% CI | p-value | FDR | BF | Total n | n individuals |
|---|---|---|---|---|---|---|---|---|---|
| | | LSNS_base | -0.026 | −0.035,−0.016 | 4.3e-08**** | 5.2e-07**** | 3.3e+05**** | 1951 | 1397 |
| | | LSNS_change | −0.002 | -0.016, 0.012 | 0.3867 | 0.4641 | 0.15 | 1951 | 1397 |
| | | age_base | −0.017 | −0.024,−0.01 | | | | 1951 | 1397 |
| | | age_change | −0.055 | −0.065,−0.044 | | | | 1951 | 1397 |
| | 1 | gender | −0.083 | −0.177, 0.011 | | | | 1951 | 1397 |
| | | LSNS_base | −0.016 | −0.026,−0.007 | 5e-04**** | 0.0021** | 64.25*** | 1808 | 1310 |
| | | LSNS_change | 0.002 | -0.012, 0.017 | 0.6208 | 0.711 | 0.12 | 1808 | 1310 |
| | | age_base | −0.013 | −0.021,−0.006 | | | | 1808 | 1310 |
| | | age_change | −0.054 | −0.064,−0.043 | | | | 1808 | 1310 |
| | | gender | −0.113 | −0.209,−0.016 | | | | 1808 | 1310 |
| | | BMI | −0.059 | −0.107,−0.011 | | | | 1808 | 1310 |
| | | CESD | −0.130 | −0.179,−0.082 | | | | 1808 | 1310 |
| | | diabetes | −0.033 | −0.168, 0.101 | | | | 1808 | 1310 |
| | | education | -0.261 | −0.426,−0.096 | | | | 1808 | 1310 |
| Executive functions | 2 | hypertension | −0.109 | −0.21,−0.007 | | | | 1808 | 1310 |
| | | LSNS_base | -0.013 | −0.021,−0.004 | 0.0015** | 0.0046** | 17.75** | 1828 | 1336 |
| | | LSNS_change | -0.019 | -0.032, -0.006 | 0.0022** | 0.0054** | 9.93* | 1828 | 1336 |
| | | age_base | −0.035 | −0.042,−0.028 | | | | 1828 | 1336 |
| | | age_change | −0.023 | −0.032,−0.014 | | | | 1828 | 1336 |
| | 1 | gender | -0.376 | −0.461,−0.291 | | | | 1828 | 1336 |
| | | LSNS_base | -0.007 | -0.016, 0.002 | 0.0638 | 0.1276 | 0.97 | 1694 | 1247 |
| | | LSNS_change | −0.017 | −0.031,−0.003 | 0.0072** | 0.0173* | 6.68* | 1694 | 1247 |
| | | age_base | −0.032 | −0.039,−0.025 | | | | 1694 | 1247 |
| | | age_change | −0.023 | −0.033,−0.014 | | | | 1694 | 1247 |
| | | gender | -0.415 | -0.502, -0.328 | | | | 1694 | 1247 |
| | | BMI | -0.028 | −0.072, 0.016 | | | | 1694 | 1247 |
| | | CESD | −0.120 | −0.164,−0.076 | | | | 1694 | 1247 |
| | | diabetes | −0.007 | −0.127, 0.113 | | | | 1694 | 1247 |
| | | education | -0.116 | −0.262, 0.029 | | | | 1694 | 1247 |
| Memory | 2 | hypertension | 0.002 | −0.089, 0.094 | | | | 1694 | 1247 |

*Appendix 1—table 12 Continued on next page*

*Appendix 1—table 12 Continued*

| dv | Model | Predictor | Estimate | 95% CI | p-value | FDR | BF | Total n | n individuals |
|---|---|---|---|---|---|---|---|---|---|
| | | LSNS_base | –0.016 | –0.024,–0.008 | 3.4e-05**** | 2e-04**** | 561.09**** | 1945 | 1392 |
| | | LSNS_change | –0.006 | –0.019, 0.007 | 0.166 | 0.2295 | 0.25 | 1945 | 1392 |
| | | age_base | –0.038 | –0.044,–0.032 | | | | 1945 | 1392 |
| | | age_change | –0.040 | –0.049,–0.03 | | | | 1945 | 1392 |
| | 1 | gender | –0.106 | –0.186,–0.026 | | | | 1945 | 1392 |
| | | LSNS_base | –0.015 | –0.023,–0.006 | 3e-04**** | 0.0017** | 118.91**** | 1802 | 1303 |
| | | LSNS_change | –0.007 | –0.02, 0.006 | 0.1523 | 0.2571 | 0.39 | 1802 | 1303 |
| | | age_base | –0.036 | –0.043,–0.03 | | | | 1802 | 1303 |
| | | age_change | –0.035 | –0.045,–0.025 | | | | 1802 | 1303 |
| | | gender | –0.122 | –0.205,–0.039 | | | | 1802 | 1303 |
| | | BMI | 0.002 | –0.04, 0.044 | | | | 1802 | 1303 |
| | | CESD | –0.027 | –0.068, 0.015 | | | | 1802 | 1303 |
| | | diabetes | –0.017 | –0.132, 0.097 | | | | 1802 | 1303 |
| | | education | -0.104 | –0.243, 0.035 | | | | 1802 | 1303 |
| Processing speed | 2 | hypertension | –0.042 | –0.129, 0.044 | | | | 1802 | 1303 |

full model1: dv~LSNS_base+LSNS_change+age_base+age_change+gender

full model2: model1 + hypertension+diabetes+education+BMI+CESD

The unit of effect sizes on hippocampal volume and cognitive functions are mm³/point on the LSNS and standard deviation/point on the LSNS, respectively.

* p<0.05, BF >3; ** p<0.01, BF >10; *** p<0.001, BF >30; **** p<0.0001, BF >100.

dv = dependent variable. CI = confidence interval. FDR = p-values after FDR-correction. BF = Bayes factor in favour of alternative hypothesis. total n = total number of observations. n individuals = number of participants. LSNS_base = baseline Lubben Social Network Score. LSNS_change = change in Lubben Social Network Score. CESD = Center for Epidemiological Studies-Depression.

**Appendix 1—table 13.** Adjusted regression coefficients and measures of significance of models controlling for physical activity.

| dv | Model | Predictor | Estimate | 95% CI | p-value | FDR | BF | Total n | n individuals |
|---|---|---|---|---|---|---|---|---|---|
| | 2 | ipaq | -12.253 | –27.485, 2.86 | | | | 1250 | 1141 |
| | 2 | LSNS_base | -5.887 | –9.939,–1.834 | 0.0022** | 0.0067** | 17.64** | 1250 | 1141 |
| | 2 | LSNS_change | -9.057 | -14.719, -3.312 | 0.0011** | 0.0042** | 11.47** | 1250 | 1141 |
| | 2 | age_base | –23.367 | –26.536,–20.199 | | | | 1250 | 1141 |
| | 2 | age_change | –27.978 | –31.859,–24.157 | | | | 1250 | 1141 |
| | 2 | gender | –36.998 | –76.761, 2.76 | | | | 1250 | 1141 |
| | 2 | BMI | 16.826 | –1.735, 35.382 | | | | 1250 | 1141 |
| | 2 | CESD | 11.892 | -8.516, 32.3 | | | | 1250 | 1141 |
| | 2 | diabetes | -113.206 | -168.435, -57.981 | | | | 1250 | 1141 |
| | 2 | education | -97.882 | -164.736, -31.03 | | | | 1250 | 1141 |
| Hippo campal volume | 2 | hypertension | -16.735 | –58.972, 25.495 | | | | 1250 | 1141 |

*Appendix 1—table 13 Continued on next page*

*Appendix 1—table 13 Continued*

| dv | Model | Predictor | Estimate | 95% CI | p-value | FDR | BF | Total n | n individuals |
|---|---|---|---|---|---|---|---|---|---|
| | 2 | ipaq | −0.035 | −0.083, 0.013 | | | | 1392 | 1249 |
| | 2 | LSNS_base | −0.018 | −0.028,−0.007 | 4e-04*** | 0.0024** | 87.51*** | 1392 | 1249 |
| | 2 | LSNS_change | 0.002 | -0.026, 0.029 | 0.5483 | 0.731 | 0.17 | 1392 | 1249 |
| | 2 | age_base | −0.012 | −0.02,−0.004 | | | | 1392 | 1249 |
| | 2 | age_change | −0.050 | −0.068,−0.031 | | | | 1392 | 1249 |
| | 2 | gender | −0.149 | −0.251,−0.047 | | | | 1392 | 1249 |
| | 2 | BMI | -0.068 | −0.119,−0.018 | | | | 1392 | 1249 |
| | 2 | CESD | −0.150 | −0.202,−0.098 | | | | 1392 | 1249 |
| | 2 | diabetes | -0.038 | −0.18, 0.103 | | | | 1392 | 1249 |
| Executive functions | 2 | education | -0.275 | -0.45, -0.099 | | | | 1392 | 1249 |
| | 2 | hypertension | −0.072 | −0.18, 0.037 | | | | 1392 | 1249 |
| | 2 | ipaq | −0.023 | -0.066, 0.021 | | | | 1342 | 1211 |
| | 2 | LSNS_base | -0.007 | -0.016, 0.002 | 0.0735 | 0.1765 | 0.95 | 1342 | 1211 |
| | 2 | LSNS_change | −0.013 | −0.037, 0.01 | 0.1342 | 0.2683 | 0.47 | 1342 | 1211 |
| | 2 | age_base | -0.032 | −0.039,−0.024 | | | | 1342 | 1211 |
| | 2 | age_change | 0.001 | −0.016, 0.017 | | | | 1342 | 1211 |
| | 2 | gender | −0.406 | −0.5,−0.312 | | | | 1342 | 1211 |
| | 2 | BMI | -0.032 | −0.079, 0.014 | | | | 1342 | 1211 |
| | 2 | CESD | -0.119 | -0.167, -0.071 | | | | 1342 | 1211 |
| | 2 | diabetes | -0.066 | −0.194, 0.063 | | | | 1342 | 1211 |
| | 2 | education | -0.154 | -0.312, 0.004 | | | | 1342 | 1211 |
| Memory | 2 | hypertension | 0.012 | −0.087, 0.111 | | | | 1342 | 1211 |
| | 2 | ipaq | -0.041 | −0.081,−0.001 | | | | 1392 | 1252 |
| | 2 | LSNS_base | −0.015 | −0.023,−0.006 | 3e-04*** | 0.0024** | 107.46**** | 1392 | 1252 |
| | 2 | LSNS_change | −0.011 | −0.036, 0.013 | 0.1869 | 0.2804 | 0.48 | 1392 | 1252 |
| | 2 | age_base | −0.033 | −0.04,−0.027 | | | | 1392 | 1252 |
| | 2 | age_change | −0.036 | −0.053,−0.019 | | | | 1392 | 1252 |
| | 2 | gender | -0.117 | -0.202, -0.033 | | | | 1392 | 1252 |
| | 2 | BMI | -0.015 | −0.057, 0.027 | | | | 1392 | 1252 |
| | 2 | CESD | -0.011 | −0.054, 0.032 | | | | 1392 | 1252 |
| | 2 | diabetes | -0.029 | -0.146, 0.087 | | | | 1392 | 1252 |
| | 2 | education | −0.058 | −0.201, 0.084 | | | | 1392 | 1252 |
| Processing speed | 2 | hypertension | −0.057 | −0.147, 0.032 | | | | 1392 | 1252 |

full model1: dv~LSNS_base +LSNS_change +age_base +age_change +gender.

full model2: model1 +hypertension + diabetes +education + BMI +CESD.

The unit of effect sizes on hippocampal volume and cognitive functions are mm³/point on the LSNS and standard deviation/point on the LSNS, respectively.

* p<0.05, BF >3; ** p<0.01, BF >10; *** p<0.001, BF >30; **** p<0.0001, BF >100.

dv = dependent variable. CI = confidence interval. FDR = p-values after FDR-correction. BF = Bayes factor in favour of alternative hypothesis. total n = total number of observations. n individuals = number of participants. ipaq; total METs/week based on the International Physical Activity Questionnaire; LSNS_base = baseline Lubben Social Network Score. LSNS_change = change in Lubben Social Network Score. CESD = Center for Epidemiological Studies-Depression.

**Appendix 1—table 14.** Adjusted regression coefficients and measures of significance of models controlling for sleep quality.

| dv | Model | Predictor | Estimate | 95% CI | p-value | FDR | BF | Total n | n individuals |
|---|---|---|---|---|---|---|---|---|---|
| | 2 | psqi | -11.915 | -30.746, 6.993 | | | | 1019 | 912 |
| | 2 | LSNS_base | -4.588 | −9.086,–0.09 | 0.0228* | 0.0548 | 2.39 | 1019 | 912 |
| | 2 | LSNS_change | -1.302 | -7.72, 5.143 | 0.3451 | 0.4601 | 0.13 | 1019 | 912 |
| | 2 | age_base | -21.787 | −25.361,–18.213 | | | | 1019 | 912 |
| | 2 | age_change | -27.975 | −32.159,–23.815 | | | | 1019 | 912 |
| | 2 | gender | –42.342 | -86.176, 1.482 | | | | 1019 | 912 |
| | 2 | BMI | 10.944 | –9.704, 31.591 | | | | 1019 | 912 |
| | 2 | CESD | 32.295 | 9.313, 55.285 | | | | 1019 | 912 |
| | 2 | diabetes | –105.790 | −166.428,–45.135 | | | | 1019 | 912 |
| | 2 | education | –170.384 | −252.271,–88.503 | | | | 1019 | 912 |
| Hippo campal volume | 2 | hypertension | –15.415 | –61.357, 30.533 | | | | 1019 | 912 |
| | 2 | psqi | 0.024 | –0.034, 0.082 | | | | 1112 | 987 |
| | 2 | LSNS_base | –0.018 | −0.03,–0.007 | 0.0011** | 0.0133* | 38.25*** | 1112 | 987 |
| | 2 | LSNS_change | -0.011 | −0.039, 0.018 | 0.2303 | 0.3455 | 0.4 | 1112 | 987 |
| | 2 | age_base | –0.004 | -0.013, 0.006 | | | | 1112 | 987 |
| | 2 | age_change | –0.051 | −0.07,–0.032 | | | | 1112 | 987 |
| | 2 | gender | –0.199 | −0.314,–0.083 | | | | 1112 | 987 |
| | 2 | BMI | –0.065 | −0.122,–0.009 | | | | 1112 | 987 |
| | 2 | CESD | -0.150 | −0.21,–0.09 | | | | 1112 | 987 |
| | 2 | diabetes | -0.154 | -0.311, 0.003 | | | | 1112 | 987 |
| | 2 | education | -0.273 | −0.49,–0.057 | | | | 1112 | 987 |
| Executive functions | 2 | hypertension | –0.058 | –0.178, 0.062 | | | | 1112 | 987 |
| | 2 | psqi | –0.005 | –0.056, 0.046 | | | | 1066 | 940 |
| | 2 | LSNS_base | -0.008 | -0.018, 0.003 | 0.0692 | 0.1384 | 1.06 | 1066 | 940 |
| | 2 | LSNS_change | –0.031 | −0.056,–0.007 | 0.0059** | 0.0237* | 6.91* | 1066 | 940 |
| | 2 | age_base | –0.031 | −0.04,–0.023 | | | | 1066 | 940 |
| | 2 | age_change | –0.030 | −0.047,–0.014 | | | | 1066 | 940 |
| | 2 | gender | -0.482 | −0.585,–0.379 | | | | 1066 | 940 |
| | 2 | BMI | –0.010 | -0.06, 0.041 | | | | 1066 | 940 |
| | 2 | CESD | –0.111 | −0.165,–0.057 | | | | 1066 | 940 |
| | 2 | diabetes | –0.087 | -0.225, 0.052 | | | | 1066 | 940 |
| | 2 | education | –0.092 | -0.282, 0.097 | | | | 1066 | 940 |
| Memory | 2 | hypertension | 0.054 | –0.053, 0.162 | | | | 1066 | 940 |

*Appendix 1—table 14 Continued on next page*

*Appendix 1—table 14 Continued*

| dv | Model | Predictor | Estimate | 95% CI | p-value | FDR | BF | Total n | n individuals |
|---|---|---|---|---|---|---|---|---|---|
| | 2 | psqi | -0.057 | −0.103,–0.01 | | | | 1114 | 986 |
| | 2 | LSNS_base | –0.013 | −0.023,–0.004 | 0.003** | 0.0179* | 15.38** | 1114 | 986 |
| | 2 | LSNS_change | -0.025 | -0.048, -0.002 | 0.0158* | 0.0475* | 4.08* | 1114 | 986 |
| | 2 | age_base | –0.031 | −0.039,–0.024 | | | | 1114 | 986 |
| | 2 | age_change | -0.024 | −0.04,–0.008 | | | | 1114 | 986 |
| | 2 | gender | –0.163 | −0.256,–0.07 | | | | 1114 | 986 |
| | 2 | BMI | –0.009 | −0.054, 0.037 | | | | 1114 | 986 |
| | 2 | CESD | 0.018 | −0.03, 0.066 | | | | 1114 | 986 |
| | 2 | diabetes | -0.059 | -0.185, 0.067 | | | | 1114 | 986 |
| | 2 | education | 0.003 | -0.168, 0.174 | | | | 1114 | 986 |
| Processing speed | 2 | hypertension | –0.005 | -0.101, 0.092 | | | | 1114 | 986 |

full model1: dv~LSNS_base+LSNS_change+age_base+age_change+gender

full model2: model1 + hypertension+diabetes+education+BMI+CESD

The unit of effect sizes on hippocampal volume and cognitive functions are mm³/point on the LSNS and standard deviation/point on the LSNS, respectively.

* p<0.05, BF >3; ** p<0.01, BF >10; *** p<0.001, BF >30; **** p<0.0001, BF >100.

dv = dependent variable. CI = confidence interval. FDR = p-values after FDR-correction. BF = Bayes factor in favour of alternative hypothesis. total n = total number of observations. n individuals = number of participants. psqi = global Pittsburgh Sleep Quality Index score. LSNS_base = baseline Lubben Social Network Score. LSNS_change = change in Lubben Social Network Score. CESD = Center for Epidemiological Studies-Depression.

**Appendix 1—table 15.** Adjusted regression coefficients and measures of significance of models using cognitive scores standardized using the baseline mean.

| dv | Model | Predictor | Estimate | 95% CI | p-value | FDR | BF | Total n | n individuals |
|---|---|---|---|---|---|---|---|---|---|
| | | LSNS_base | –0.028 | −0.037,–0.019 | 8.4e-10**** | 1.0e-08**** | 1.3e+07**** | 2047 | 1468 |
| | | LSNS_change | –0.002 | −0.017, 0.012 | 0.3681 | 0.4417 | 0.15 | 2047 | 1468 |
| | | age_base | -0.017 | -0.024, -0.01 | | | | 2047 | 1468 |
| | | age_change | -0.050 | −0.06,–0.04 | | | | 2047 | 1468 |
| | 1 | gender | -0.073 | −0.164, 0.018 | | | | 2047 | 1468 |
| | | LSNS_base | -0.017 | −0.026,–0.007 | 2e-04**** | 0.0015** | 111.14**** | 1892 | 1376 |
| | | LSNS_change | 0.002 | −0.013, 0.017 | 0.6112 | 0.7335 | 0.13 | 1892 | 1376 |
| | | age_base | –0.012 | −0.02,–0.005 | | | | 1892 | 1376 |
| | | age_change | -0.050 | −0.061,–0.04 | | | | 1892 | 1376 |
| | | gender | -0.111 | −0.204,–0.018 | | | | 1892 | 1376 |
| | | BMI | –0.072 | −0.119,–0.026 | | | | 1892 | 1376 |
| | | CESD | –0.139 | −0.185,–0.092 | | | | 1892 | 1376 |
| | | diabetes | -0.059 | −0.187, 0.07 | | | | 1892 | 1376 |
| | | education | -0.349 | −0.504,–0.193 | | | | 1892 | 1376 |
| Executive functions | 2 | hypertension | –0.101 | −0.199,–0.003 | | | | 1892 | 1376 |

*Appendix 1—table 15 Continued on next page*

*Appendix 1—table 15 Continued*

| dv | Model | Predictor | Estimate | 95% CI | p-value | FDR | BF | Total n | n individuals |
|---|---|---|---|---|---|---|---|---|---|
| | | LSNS_base | −0.014 | −0.022,−0.006 | 4e-04**** | 0.0016** | 58.49*** | 1920 | 1407 |
| | | LSNS_change | −0.019 | −0.032,−0.006 | 0.0018** | 0.0044** | 11.45** | 1920 | 1407 |
| | | age_base | -0.035 | −0.042,−0.029 | | | | 1920 | 1407 |
| | | age_change | −0.017 | −0.026,−0.008 | | | | 1920 | 1407 |
| | 1 | gender | −0.378 | −0.462,−0.295 | | | | 1920 | 1407 |
| | | LSNS_base | −0.007 | −0.016, 0.001 | 0.0479* | 0.0957 | 1.21 | 1776 | 1314 |
| | | LSNS_change | −0.017 | −0.031,−0.004 | 0.0062** | 0.0148* | 5.51* | 1776 | 1314 |
| | | age_base | −0.032 | −0.039,−0.025 | | | | 1776 | 1314 |
| | | age_change | −0.018 | −0.028,−0.009 | | | | 1776 | 1314 |
| | | gender | −0.419 | −0.505,−0.333 | | | | 1776 | 1314 |
| | | BMI | −0.031 | −0.074, 0.012 | | | | 1776 | 1314 |
| | | CESD | −0.119 | −0.162,−0.076 | | | | 1776 | 1314 |
| | | diabetes | −0.037 | -0.155, 0.08 | | | | 1776 | 1314 |
| | | education | −0.197 | −0.338,−0.055 | | | | 1776 | 1314 |
| Memory | 2 | hypertension | 0.015 | −0.076, 0.106 | | | | 1776 | 1314 |
| | | LSNS_base | −0.018 | −0.026,−0.01 | 1.9e-06**** | 1.2e-05**** | 8.1e+03**** | 2042 | 1469 |
| | | LSNS_change | −0.006 | −0.019, 0.006 | 0.1611 | 0.2416 | 0.26 | 2042 | 1469 |
| | | age_base | −0.038 | −0.044,−0.032 | | | | 2042 | 1469 |
| | | age_change | −0.038 | −0.047,−0.028 | | | | 2042 | 1469 |
| | 1 | gender | −0.108 | −0.185,−0.03 | | | | 2042 | 1469 |
| | | LSNS_base | −0.017 | −0.025,−0.009 | 2.0e-05**** | 2e-04**** | 1.3e+03**** | 1887 | 1375 |
| | | LSNS_change | −0.008 | −0.021, 0.006 | 0.1279 | 0.1987 | 0.44 | 1887 | 1375 |
| | | age_base | −0.036 | −0.042,−0.029 | | | | 1887 | 1375 |
| | | age_change | −0.033 | −0.043,−0.024 | | | | 1887 | 1375 |
| | | gender | −0.124 | −0.204,−0.043 | | | | 1887 | 1375 |
| | | BMI | −0.013 | -0.053, 0.028 | | | | 1887 | 1375 |
| | | CESD | −0.027 | -0.067, 0.013 | | | | 1887 | 1375 |
| | | diabetes | −0.005 | −0.115, 0.105 | | | | 1887 | 1375 |
| | | education | −0.103 | −0.235, 0.029 | | | | 1887 | 1375 |
| Processing speed | 2 | hypertension | −0.067 | −0.152, 0.017 | | | | 1887 | 1375 |

full model1: dv~LSNS_base +LSNS_change +age_base +age_change +gender.

full model2: model1 +hypertension + diabetes +education + BMI +CESD.

The unit of effect sizes on hippocampal volume and cognitive functions are mm³/point on the LSNS and standard deviation/point on the LSNS, respectively.

* P<0.05, BF >3; ** p<0.01, BF >10; *** p<0.001, BF >30; **** p<0.0001, BF >100.

dv = dependent variable. CI = confidence interval. FDR = p-values after FDR-correction. BF = Bayes factor in favour of alternative hypothesis. total n = total number of observations. n individuals = number of participants. LSNS_base = baseline Lubben Social Network Score. LSNS_change = change in Lubben Social Network Score. CESD = Center for Epidemiological Studies-Depression.

**Appendix 1—table 16.** Adjusted regression coefficients and measures of significance of models without interaction terms coding social isolation dichotomously.

| dv | Model | Predictor | Estimate | 95% CI | p-value | FDR | BF | Total n | n individuals |
|---|---|---|---|---|---|---|---|---|---|
| | | LSNS_base | −84.196 | −132.116,−36.27 | 3e-04**** | 0.0012** | 65.9*** | 1667 | 1306 |
| | | LSNS_change | -39.200 | −76.322,−2.129 | 0.0193* | 0.0462* | 0.64 | 1667 | 1306 |
| | | age_base | −26.126 | −28.968,−23.285 | | | | 1667 | 1306 |
| | | age_change | −26.947 | −29.131,−24.77 | | | | 1667 | 1306 |
| | 1 | gender | −44.575 | −81.629,−7.521 | | | | 1667 | 1306 |
| | | LSNS_base | −81.625 | −131.358,−31.884 | 7e-04**** | 0.004** | 43.8*** | 1556 | 1226 |
| | | LSNS_change | −38.922 | −77.936, 0.026 | 0.0252* | 0.101 | 0.71 | 1556 | 1226 |
| | | age_base | −24.310 | −27.348,−21.272 | | | | 1556 | 1226 |
| | | age_change | −26.905 | −29.243,−24.575 | | | | 1556 | 1226 |
| | | gender | −44.492 | −82.58,−6.406 | | | | 1556 | 1226 |
| | | BMI | 14.402 | −2.476, 31.281 | | | | 1556 | 1226 |
| | | CESD | 10.096 | −8.894, 29.085 | | | | 1556 | 1226 |
| | | diabetes | −97.865 | −150.283,−45.437 | | | | 1556 | 1226 |
| | | education | −95.626 | −157.878,−33.368 | | | | 1556 | 1226 |
| Hippo campal volume | 2 | hypertension | -22.042 | −62.348, 18.267 | | | | 1556 | 1226 |
| | | LSNS_base | −0.224 | −0.347,−0.101 | 2e-04**** | 0.0011** | 153.03**** | 2048 | 1469 |
| | | LSNS_change | −0.065 | -0.219, 0.088 | 0.2012 | 0.2415 | 0.27 | 2048 | 1469 |
| | | age_base | −0.020 | −0.027,−0.013 | | | | 2048 | 1469 |
| | | age_change | −0.050 | −0.06,−0.04 | | | | 2048 | 1469 |
| | 1 | gender | −0.077 | −0.17, 0.016 | | | | 2048 | 1469 |
| | | LSNS_base | −0.107 | −0.232, 0.018 | 0.0461* | 0.1383 | 1.32 | 1893 | 1377 |
| | | LSNS_change | 0.023 | -0.137, 0.183 | 0.6125 | 0.6682 | 0.11 | 1893 | 1377 |
| | | age_base | −0.015 | −0.022,−0.007 | | | | 1893 | 1377 |
| | | age_change | −0.051 | −0.061,−0.04 | | | | 1893 | 1377 |
| | | gender | −0.121 | −0.215,−0.027 | | | | 1893 | 1377 |
| | | BMI | −0.076 | −0.123,−0.03 | | | | 1893 | 1377 |
| | | CESD | -0.154 | −0.2,−0.108 | | | | 1893 | 1377 |
| | | diabetes | -0.054 | −0.184, 0.076 | | | | 1893 | 1377 |
| | | education | −0.348 | −0.506,−0.19 | | | | 1893 | 1377 |
| Executive functions | 2 | hypertension | −0.098 | −0.197, 0.001 | | | | 1893 | 1377 |

*Appendix 1—table 16 Continued on next page*

*Appendix 1—table 16 Continued*

| dv | Model | Predictor | Estimate | 95% CI | p-value | FDR | BF | Total n | n individuals |
|---|---|---|---|---|---|---|---|---|---|
| | | LSNS_base | –0.128 | –0.239,–0.017 | 0.012* | 0.036* | 2.79 | 1921 | 1408 |
| | | LSNS_change | –0.117 | -0.255, 0.021 | 0.048* | 0.096 | 0.67 | 1921 | 1408 |
| | | age_base | –0.036 | –0.043,–0.029 | | | | 1921 | 1408 |
| | | age_change | –0.018 | –0.027,–0.009 | | | | 1921 | 1408 |
| | 1 | gender | –0.379 | –0.463,–0.295 | | | | 1921 | 1408 |
| | | LSNS_base | –0.060 | –0.175, 0.055 | 0.1537 | 0.205 | 0.44 | 1777 | 1315 |
| | | LSNS_change | -0.086 | –0.231, 0.059 | 0.1222 | 0.205 | 0.41 | 1777 | 1315 |
| | | age_base | –0.033 | –0.04,–0.026 | | | | 1777 | 1315 |
| | | age_change | –0.019 | –0.028,–0.009 | | | | 1777 | 1315 |
| | | gender | –0.421 | –0.507,–0.335 | | | | 1777 | 1315 |
| | | BMI | –0.031 | –0.074, 0.012 | | | | 1777 | 1315 |
| | | CESD | –0.126 | –0.169,–0.083 | | | | 1777 | 1315 |
| | | diabetes | -0.037 | –0.155, 0.081 | | | | 1777 | 1315 |
| | | education | –0.179 | –0.321,–0.037 | | | | 1777 | 1315 |
| Memory | 2 | hypertension | 0.016 | –0.075, 0.107 | | | | 1777 | 1315 |
| | | LSNS_base | –0.254 | –0.357,–0.151 | 7.0e-07**** | 8.4e-06**** | 2.1e+04**** | 2043 | 1470 |
| | | LSNS_change | –0.089 | –0.23, 0.051 | 0.1061 | 0.1414 | 0.35 | 2043 | 1470 |
| | | age_base | –0.038 | –0.044,–0.032 | | | | 2043 | 1470 |
| | | age_change | –0.038 | –0.047,–0.028 | | | | 2043 | 1470 |
| | 1 | gender | –0.103 | –0.18,–0.025 | | | | 2043 | 1470 |
| | | LSNS_base | –0.238 | –0.344,–0.131 | 6.6e-06**** | 7.9e-05**** | 3.8e+03**** | 1888 | 1376 |
| | | LSNS_change | -0.101 | –0.245, 0.044 | 0.0861 | 0.1722 | 0.62 | 1888 | 1376 |
| | | age_base | –0.036 | –0.042,–0.03 | | | | 1888 | 1376 |
| | | age_change | –0.033 | –0.043,–0.024 | | | | 1888 | 1376 |
| | | gender | –0.119 | –0.199,–0.039 | | | | 1888 | 1376 |
| | | BMI | –0.013 | –0.053, 0.027 | | | | 1888 | 1376 |
| | | CESD | –0.033 | -0.073, 0.006 | | | | 1888 | 1376 |
| | | diabetes | –0.001 | –0.112, 0.109 | | | | 1888 | 1376 |
| | | education | –0.122 | –0.254, 0.01 | | | | 1888 | 1376 |
| Processing speed | 2 | hypertension | –0.067 | –0.151, 0.018 | | | | 1888 | 1376 |

full model1: dv~LSNS_base +LSNS_change +age_base +age_change +gender.

full model2: model1 +hypertension + diabetes +education + BMI +CESD.

The unit of effect sizes on hippocampal volume and cognitive functions are mm³/point on the LSNS and standard deviation/point on the LSNS, respectively.

* p<0.05, BF >3; ** p<0.01, BF >10; *** p<0.001, BF >30; **** p<0.0001, BF >100.

dv = dependent variable. CI = confidence interval. FDR = p-values after FDR-correction. BF = Bayes factor in favour of alternative hypothesis. total n = total number of observations. n individuals = number of participants. LSNS_base = baseline Lubben Social Network Score. LSNS_change = change in Lubben Social Network Score. CESD = Center for Epidemiological Studies-Depression.

**Appendix 1—table 17.** Adjusted regression coefficients and measures of significance of models with interaction term of baseline social isolation with change in age coding social isolation dichotomously.

| dv | Model | Predictor | Estimate | 95% CI | p-value | FDR | BF | Total n | n individuals |
|---|---|---|---|---|---|---|---|---|---|
| | | LSNS_base*age_change | −5.548 | -13.179, 2.057 | 0.0766 | 0.1312 | 0.22 | 1667 | 1306 |
| | | LSNS_base | −80.584 | −128.741,−32.428 | | | | 1667 | 1306 |
| | | LSNS_change | −54.052 | −96.412,−11.813 | | | | 1667 | 1306 |
| | | age_base | −26.085 | −28.926,−23.244 | | | | 1667 | 1306 |
| | | age_change | −26.202 | −28.609,−23.797 | | | | 1667 | 1306 |
| | 1 | gender | −44.416 | −81.456,−7.377 | | | | 1667 | 1306 |
| | | LSNS_base*age_change | −5.693 | -13.731, 2.305 | 0.0817 | 0.1722 | 0.28 | 1556 | 1226 |
| | | LSNS_base | −77.832 | −127.826,−27.839 | | | | 1556 | 1226 |
| | | LSNS_change | −54.295 | −98.952,−9.81 | | | | 1556 | 1226 |
| | | age_base | −24.257 | −27.294,−21.22 | | | | 1556 | 1226 |
| | | age_change | −26.109 | −28.697,−23.525 | | | | 1556 | 1226 |
| | | gender | −44.285 | −82.353,−6.219 | | | | 1556 | 1226 |
| | | BMI | 14.474 | −2.392, 31.343 | | | | 1556 | 1226 |
| | | CESD | 10.297 | −8.685, 29.277 | | | | 1556 | 1226 |
| | | diabetes | −98.022 | −150.411,−45.622 | | | | 1556 | 1226 |
| | | education | −95.589 | −157.807,−33.365 | | | | 1556 | 1226 |
| Hippo campal volume | 2 | hypertension | -22.260 | −62.545, 18.028 | | | | 1556 | 1226 |
| | | LSNS_base*age_change | 0.008 | −0.025, 0.041 | 0.6861 | 0.6861 | 0.08 | 2048 | 1469 |
| | | LSNS_base | −0.234 | −0.363,−0.105 | | | | 2048 | 1469 |
| | | LSNS_change | −0.046 | −0.218, 0.126 | | | | 2048 | 1469 |
| | | age_base | −0.020 | −0.027,−0.013 | | | | 2048 | 1469 |
| | | age_change | −0.051 | −0.063,−0.04 | | | | 2048 | 1469 |
| | 1 | gender | −0.078 | −0.17, 0.015 | | | | 2048 | 1469 |
| | | LSNS_base*age_change | 0.012 | −0.022, 0.046 | 0.7533 | 0.7533 | 0.1 | 1893 | 1377 |
| | | LSNS_base | -0.121 | −0.253, 0.01 | | | | 1893 | 1377 |
| | | LSNS_change | 0.052 | −0.128, 0.232 | | | | 1893 | 1377 |
| | | age_base | −0.015 | −0.022,−0.007 | | | | 1893 | 1377 |
| | | age_change | −0.053 | −0.064,−0.041 | | | | 1893 | 1377 |
| | | gender | -0.122 | −0.216,−0.027 | | | | 1893 | 1377 |
| | | BMI | -0.076 | -0.123, -0.029 | | | | 1893 | 1377 |
| | | CESD | −0.154 | −0.201,−0.108 | | | | 1893 | 1377 |
| | | diabetes | −0.055 | −0.185, 0.075 | | | | 1893 | 1377 |
| | | education | −0.348 | −0.506,−0.19 | | | | 1893 | 1377 |
| Executive functions | 2 | hypertension | −0.097 | -0.197, 0.002 | | | | 1893 | 1377 |

*Appendix 1—table 17 Continued on next page*

*Appendix 1—table 17 Continued*

| dv | Model | Predictor | Estimate | 95% CI | p-value | FDR | BF | Total n | n individuals |
|---|---|---|---|---|---|---|---|---|---|
| | | LSNS_base*age_change | -0.004 | –0.034, 0.026 | 0.3908 | 0.4264 | 0.12 | 1921 | 1408 |
| | | LSNS_base | –0.124 | –0.239,–0.008 | | | | 1921 | 1408 |
| | | LSNS_change | -0.126 | -0.281, 0.027 | | | | 1921 | 1408 |
| | | age_base | –0.036 | –0.042,–0.029 | | | | 1921 | 1408 |
| | | age_change | –0.017 | –0.027,–0.007 | | | | 1921 | 1408 |
| | 1 | gender | –0.379 | –0.463,–0.295 | | | | 1921 | 1408 |
| | | LSNS_base*age_change | 0.002 | –0.029, 0.034 | 0.5602 | 0.6682 | 0.11 | 1777 | 1315 |
| | | LSNS_base | -0.062 | –0.182, 0.057 | | | | 1777 | 1315 |
| | | LSNS_change | –0.080 | -0.244, 0.083 | | | | 1777 | 1315 |
| | | age_base | -0.033 | -0.04, -0.026 | | | | 1777 | 1315 |
| | | age_change | -0.019 | -0.03, -0.008 | | | | 1777 | 1315 |
| | | gender | –0.421 | –0.507,–0.335 | | | | 1777 | 1315 |
| | | BMI | –0.031 | –0.074, 0.012 | | | | 1777 | 1315 |
| | | CESD | –0.126 | –0.169,–0.083 | | | | 1777 | 1315 |
| | | diabetes | –0.037 | –0.155, 0.081 | | | | 1777 | 1315 |
| | | education | –0.179 | –0.321,–0.037 | | | | 1777 | 1315 |
| Memory | 2 | hypertension | 0.016 | –0.075, 0.107 | | | | 1777 | 1315 |
| | | LSNS_base*age_change | –0.020 | -0.05, 0.01 | 0.0932 | 0.1399 | 0.42 | 2043 | 1470 |
| | | LSNS_base | –0.229 | –0.338,–0.12 | | | | 2043 | 1470 |
| | | LSNS_change | –0.136 | –0.293, 0.02 | | | | 2043 | 1470 |
| | | age_base | –0.038 | –0.044,–0.032 | | | | 2043 | 1470 |
| | | age_change | –0.035 | –0.045,–0.024 | | | | 2043 | 1470 |
| | 1 | gender | –0.102 | –0.179,–0.025 | | | | 2043 | 1470 |
| | | LSNS_base*age_change | -0.017 | –0.047, 0.014 | 0.1434 | 0.205 | 0.39 | 1888 | 1376 |
| | | LSNS_base | –0.218 | –0.331,–0.106 | | | | 1888 | 1376 |
| | | LSNS_change | –0.141 | -0.303, 0.022 | | | | 1888 | 1376 |
| | | age_base | –0.036 | –0.042,–0.029 | | | | 1888 | 1376 |
| | | age_change | –0.031 | –0.041,–0.02 | | | | 1888 | 1376 |
| | | gender | –0.118 | –0.198,–0.038 | | | | 1888 | 1376 |
| | | BMI | -0.013 | –0.053, 0.027 | | | | 1888 | 1376 |
| | | CESD | –0.033 | –0.072, 0.007 | | | | 1888 | 1376 |
| | | diabetes | 0.000 | –0.11, 0.11 | | | | 1888 | 1376 |
| | | education | –0.122 | –0.254, 0.01 | | | | 1888 | 1376 |
| Processing speed | 2 | hypertension | –0.068 | –0.152, 0.017 | | | | 1888 | 1376 |

full model1: dv~LSNS_base +LSNS_change +age_base +age_change +gender.

full model2: model1 +hypertension + diabetes +education + BMI +CESD.

dv = dependent variable. CI = confidence interval. FDR = p-values after FDR-correction. BF = Bayes factor in favour of alternative hypothesis. total n = total number of observations. n individuals = number of participants. LSNS_base = baseline Lubben Social Network Score. LSNS_change = change in Lubben Social Network Score. CESD = Center for Epidemiological Studies-Depression.

**Appendix 1—table 18.** Adjusted regression coefficients and measures of significance of models with interaction term of baseline social isolation with change in social isolation coding social isolation dichotomously.

| dv | Model | Predictor | Estimate | 95% CI | p-value | BF | Total n | n individuals |
|---|---|---|---|---|---|---|---|---|
| | | LSNS_base*LSNS_change | 88.30 | 6.68, 170.14 | 0.9828 | 0.01 | 1667 | 1306 |
| | | LSNS_base | −81.55 | −129.51,−33.59 | | | 1667 | 1306 |
| | | LSNS_change | −73.96 | −123.04,−25.02 | | | 1667 | 1306 |
| | | age_base | -26.07 | -28.91, -23.23 | | | 1667 | 1306 |
| | | age_change | -26.01 | -28.35, -23.67 | | | 1667 | 1306 |
| | 1 | gender | -44.60 | -81.64, -7.57 | | | 1667 | 1306 |
| | | LSNS_base*LSNS_change | 80.69 | −5.28, 166.93 | 0.9669 | 0.03 | 1556 | 1226 |
| | | LSNS_base | −79.16 | −128.94,−29.38 | | | 1556 | 1226 |
| | | LSNS_change | −70.92 | −122.76,−19.26 | | | 1556 | 1226 |
| | | age_base | −24.26 | −27.3,−21.23 | | | 1556 | 1226 |
| | | age_change | −26.01 | −28.53,−23.5 | | | 1556 | 1226 |
| | | gender | −44.46 | −82.53,−6.39 | | | 1556 | 1226 |
| | | BMI | 14.26 | −2.6, 31.12 | | | 1556 | 1226 |
| | | CESD | 10.14 | −8.84, 29.12 | | | 1556 | 1226 |
| | | diabetes | -96.81 | -149.22, -44.4 | | | 1556 | 1226 |
| | | education | −95.85 | −158.07,−33.62 | | | 1556 | 1226 |
| Hippo campal volume | 2 | hypertension | -22.07 | −62.35, 18.22 | | | 1556 | 1226 |

full model1: dv~LSNS_base +LSNS_change +age_base +age_change +gender.

full model2: model1 +hypertension + diabetes +education + BMI +CESD

dv = dependent variable. CI = confidence interval. BF = Bayes factor in favour of alternative hypothesis. total n = total number of observations. n individuals = number of participants. LSNS_base = baseline Lubben Social Network Score. LSNS_change = change in Lubben Social Network Score. CESD = Center for Epidemiological Studies-Depression

**Appendix 1—table 19.** Adjusted regression coefficients and measures of significance of models with interaction of baseline social isolation with dichotomously coded social isolation.

| dv | Model | Predictor | Estimate | 95% CI | p-value | FDR | BF | Total n | n individuals |
|---|---|---|---|---|---|---|---|---|---|
| | | LSNS_base:LSNS_cut | 2.250 | –6.458, 10.963 | 0.6938 | 0.8905 | 0.16 | 1667 | 1306 |
| | | LSNS_cut | –71.797 | -236.505, 92.595 | | | | 1667 | 1306 |
| | | LSNS_base | –4.417 | -8.954, 0.121 | | | | 1667 | 1306 |
| | | LSNS_change | –3.100 | -6.926, 0.746 | | | | 1667 | 1306 |
| | | age_base | –26.018 | –28.869,–23.167 | | | | 1667 | 1306 |
| | | age_change | -26.635 | –28.876,–24.4 | | | | 1667 | 1306 |
| | 1 | gender | –46.137 | –83.249,–9.026 | | | | 1667 | 1306 |
| | | LSNS_base:LSNS_cut | 2.285 | –6.996, 11.574 | 0.6854 | 0.8905 | 0.07 | 1556 | 1226 |
| | | LSNS_cut | –72.191 | -248.895, 104.119 | | | | 1556 | 1226 |
| | | LSNS_base | -4.504 | –9.251, 0.242 | | | | 1556 | 1226 |
| | | LSNS_change | –2.989 | -7.062, 1.109 | | | | 1556 | 1226 |
| | | age_base | -24.193 | –27.238,–21.149 | | | | 1556 | 1226 |
| | | age_change | -26.615 | -29.017, -24.22 | | | | 1556 | 1226 |
| | | gender | –45.963 | –84.1,–7.829 | | | | 1556 | 1226 |
| | | BMI | 13.651 | –3.22, 30.525 | | | | 1556 | 1226 |
| | | CESD | 11.931 | –7.334, 31.198 | | | | 1556 | 1226 |
| | | diabetes | –98.801 | –151.224,–46.371 | | | | 1556 | 1226 |
| | | education | –94.113 | –156.541,–31.678 | | | | 1556 | 1226 |
| Hippo campal volume | 2 | hypertension | –21.159 | -61.488, 19.173 | | | | 1556 | 1226 |
| | | LSNS_base:LSNS_cut | 0.017 | -0.01, 0.045 | 0.8905 | 0.8905 | 0.05 | 2048 | 1469 |
| | | LSNS_cut | –0.298 | -0.845, 0.249 | | | | 2048 | 1469 |
| | | LSNS_base | -0.033 | -0.045, -0.02 | | | | 2048 | 1469 |
| | | LSNS_change | –0.002 | -0.019, 0.014 | | | | 2048 | 1469 |
| | | age_base | –0.019 | –0.026,–0.012 | | | | 2048 | 1469 |
| | | age_change | –0.049 | –0.059,–0.038 | | | | 2048 | 1469 |
| | 1 | gender | –0.088 | –0.18, 0.005 | | | | 2048 | 1469 |
| | | LSNS_base:LSNS_cut | 0.016 | –0.013, 0.046 | 0.8577 | 0.8905 | 0.1 | 1893 | 1377 |
| | | LSNS_cut | -0.234 | -0.819, 0.351 | | | | 1893 | 1377 |
| | | LSNS_base | -0.023 | -0.036, -0.01 | | | | 1893 | 1377 |
| | | LSNS_change | 0.000 | -0.017, 0.016 | | | | 1893 | 1377 |
| | | age_base | –0.014 | –0.021,–0.006 | | | | 1893 | 1377 |
| | | age_change | –0.049 | –0.06,–0.039 | | | | 1893 | 1377 |
| | | gender | -0.127 | –0.221,–0.034 | | | | 1893 | 1377 |
| | | BMI | –0.075 | –0.122,–0.029 | | | | 1893 | 1377 |
| | | CESD | –0.141 | –0.187,–0.094 | | | | 1893 | 1377 |
| | | diabetes | –0.055 | –0.184, 0.074 | | | | 1893 | 1377 |
| | | education | –0.332 | –0.489,–0.174 | | | | 1893 | 1377 |
| Executive functions | 2 | hypertension | –0.097 | –0.195, 0.002 | | | | 1893 | 1377 |

*Appendix 1—table 19 Continued on next page*

*Appendix 1—table 19 Continued*

| dv | Model | Predictor | Estimate | 95% CI | p-value | FDR | BF | Total n | n individuals |
|---|---|---|---|---|---|---|---|---|---|
| | | LSNS_base:LSNS_cut | 0.003 | −0.022, 0.029 | 0.5968 | 0.8905 | 0.1 | 1921 | 1408 |
| | | LSNS_cut | −0.044 | −0.545, 0.457 | | | | 1921 | 1408 |
| | | LSNS_base | −0.016 | −0.027,−0.004 | | | | 1921 | 1408 |
| | | LSNS_change | −0.020 | −0.035,−0.005 | | | | 1921 | 1408 |
| | | age_base | -0.035 | −0.042,−0.029 | | | | 1921 | 1408 |
| | | age_change | -0.017 | −0.026,−0.008 | | | | 1921 | 1408 |
| | 1 | gender | -0.382 | −0.466,−0.298 | | | | 1921 | 1408 |
| | | LSNS_base:LSNS_cut | −0.001 | −0.028, 0.027 | 0.4788 | 0.8905 | 0.16 | 1777 | 1315 |
| | | LSNS_cut | 0.043 | −0.501, 0.587 | | | | 1777 | 1315 |
| | | LSNS_base | −0.009 | −0.021, 0.003 | | | | 1777 | 1315 |
| | | LSNS_change | −0.020 | −0.035,−0.004 | | | | 1777 | 1315 |
| | | age_base | -0.033 | -0.04, -0.026 | | | | 1777 | 1315 |
| | | age_change | −0.018 | −0.028,−0.009 | | | | 1777 | 1315 |
| | | gender | −0.421 | −0.507,−0.335 | | | | 1777 | 1315 |
| | | BMI | −0.031 | -0.074, 0.012 | | | | 1777 | 1315 |
| | | CESD | −0.120 | −0.164,−0.077 | | | | 1777 | 1315 |
| | | diabetes | −0.038 | -0.156, 0.08 | | | | 1777 | 1315 |
| | | education | −0.173 | −0.315,−0.031 | | | | 1777 | 1315 |
| Memory | 2 | hypertension | 0.017 | −0.074, 0.108 | | | | 1777 | 1315 |
| | | LSNS_base:LSNS_cut | −0.005 | −0.03, 0.02 | 0.3485 | 0.8905 | 0.17 | 2043 | 1470 |
| | | LSNS_cut | −0.028 | −0.521, 0.465 | | | | 2043 | 1470 |
| | | LSNS_base | −0.011 | −0.022, 0 | | | | 2043 | 1470 |
| | | LSNS_change | −0.001 | −0.016, 0.014 | | | | 2043 | 1470 |
| | | age_base | −0.038 | −0.044,−0.032 | | | | 2043 | 1470 |
| | | age_change | −0.038 | −0.047,−0.028 | | | | 2043 | 1470 |
| | 1 | gender | −0.104 | −0.182,−0.027 | | | | 2043 | 1470 |
| | | LSNS_base:LSNS_cut | −0.005 | −0.032, 0.021 | 0.3406 | 0.8905 | 0.2 | 1888 | 1376 |
| | | LSNS_cut | −0.017 | -0.538, 0.503 | | | | 1888 | 1376 |
| | | LSNS_base | −0.010 | −0.021, 0.002 | | | | 1888 | 1376 |
| | | LSNS_change | -0.002 | −0.017, 0.013 | | | | 1888 | 1376 |
| | | age_base | −0.036 | −0.042,−0.029 | | | | 1888 | 1376 |
| | | age_change | −0.034 | −0.043,−0.024 | | | | 1888 | 1376 |
| | | gender | −0.119 | −0.2,−0.039 | | | | 1888 | 1376 |
| | | BMI | −0.012 | −0.052, 0.028 | | | | 1888 | 1376 |
| | | CESD | −0.028 | −0.068, 0.012 | | | | 1888 | 1376 |
| | | diabetes | −0.004 | -0.114, 0.106 | | | | 1888 | 1376 |
| | | education | −0.115 | −0.248, 0.017 | | | | 1888 | 1376 |
| Processing speed | 2 | hypertension | −0.066 | −0.151, 0.018 | | | | 1888 | 1376 |

\* p<0.05, BF >3; \*\* p<0.01, BF >10; \*\*\* p<0.001, BF >30; \*\*\*\* p<0.0001, BF >100.

full model1: dv~LSNS_base+LSNS_change+age_base+age_change+gender
full model2: model1 + hypertension+diabetes+education+BMI+CESD
The unit of effect sizes on hippocampal volume and cognitive functions are mm³/point on the LSNS and standard deviation/point on the LSNS, respectively.

dv = dependent variable. CI = confidence interval. FDR = p-values after FDR-correction. BF = Bayes factor in favour of alternative hypothesis. total n = total number of observations. n individuals = number of participants. LSNS_base = baseline Lubben Social Network Score. LSNS_change = change in Lubben Social Network Score. CESD = Center for Epidemiological Studies-Depression

**Appendix 1—table 20.** Adjusted regression coefficients and measures of significance of models with interaction of change in social isolation with dichotomously coded social isolation.

| dv | Model | Predictor | Estimate | 95% CI | p-value | FDR | BF | Total n | n individuals |
|---|---|---|---|---|---|---|---|---|---|
| | | LSNS_change:LSNS_cut | –0.930 | –9.602, 7.76 | 0.4167 | 0.4547 | 0.01 | 1667 | 1306 |
| | | LSNS_cut | –28.481 | -70.048, 12.893 | | | | 1667 | 1306 |
| | | LSNS_base | –4.055 | -8.326, 0.216 | | | | 1667 | 1306 |
| | | LSNS_change | –3.107 | –7.124, 0.926 | | | | 1667 | 1306 |
| | | age_base | –26.031 | –28.882,–23.18 | | | | 1667 | 1306 |
| | | age_change | -26.718 | -28.955, -24.489 | | | | 1667 | 1306 |
| | 1 | gender | –45.786 | –82.874,–8.701 | | | | 1667 | 1306 |
| | | LSNS_change:LSNS_cut | –0.531 | –9.699, 8.652 | 0.4547 | 0.4547 | 0.08 | 1556 | 1226 |
| | | LSNS_cut | -28.795 | –71.993, 14.198 | | | | 1556 | 1226 |
| | | LSNS_base | -4.120 | -8.611, 0.37 | | | | 1556 | 1226 |
| | | LSNS_change | –3.056 | -7.347, 1.258 | | | | 1556 | 1226 |
| | | age_base | –24.213 | –27.257,–21.169 | | | | 1556 | 1226 |
| | | age_change | –26.727 | –29.127,–24.336 | | | | 1556 | 1226 |
| | | gender | –45.537 | –83.637,–7.44 | | | | 1556 | 1226 |
| | | BMI | 13.763 | -3.103, 30.633 | | | | 1556 | 1226 |
| | | CESD | 11.997 | -7.275, 31.269 | | | | 1556 | 1226 |
| | | diabetes | –98.822 | –151.25,–46.385 | | | | 1556 | 1226 |
| Hippo | | education | -93.862 | -156.289, -31.429 | | | | 1556 | 1226 |
| campal volume | 2 | hypertension | –21.270 | -61.601, 19.064 | | | | 1556 | 1226 |
| | | LSNS_change:LSNS_cut | –0.035 | –0.067,–0.003 | 0.165 | 0.4547 | 2.32 | 2048 | 1469 |
| | | LSNS_cut | 0.088 | –0.06, 0.236 | | | | 2048 | 1469 |
| | | LSNS_base | –0.032 | –0.044,–0.02 | | | | 2048 | 1469 |
| | | LSNS_change | 0.004 | –0.014, 0.021 | | | | 2048 | 1469 |
| | | age_base | –0.019 | –0.026,–0.012 | | | | 2048 | 1469 |
| | | age_change | –0.047 | –0.058,–0.037 | | | | 2048 | 1469 |
| | 1 | gender | –0.086 | –0.178, 0.006 | | | | 2048 | 1469 |
| | | LSNS_change:LSNS_cut | –0.023 | –0.057, 0.012 | 0.0992 | 0.4547 | 0.63 | 1893 | 1377 |
| | | LSNS_cut | 0.108 | –0.043, 0.259 | | | | 1893 | 1377 |
| | | LSNS_base | –0.022 | –0.035,–0.01 | | | | 1893 | 1377 |
| | | LSNS_change | 0.003 | –0.016, 0.021 | | | | 1893 | 1377 |
| | | age_base | –0.014 | –0.021,–0.006 | | | | 1893 | 1377 |
| | | age_change | –0.049 | –0.06,–0.038 | | | | 1893 | 1377 |
| | | gender | –0.125 | –0.219,–0.031 | | | | 1893 | 1377 |
| | | BMI | –0.075 | –0.121,–0.029 | | | | 1893 | 1377 |
| | | CESD | –0.140 | –0.187,–0.093 | | | | 1893 | 1377 |
| | | diabetes | –0.056 | –0.185, 0.073 | | | | 1893 | 1377 |
| | | education | -0.331 | -0.489, -0.174 | | | | 1893 | 1377 |
| Executive functions | 2 | hypertension | -0.097 | –0.196, 0.001 | | | | 1893 | 1377 |

*Appendix 1—table 20 Continued on next page*

*Appendix 1—table 20 Continued*

| dv | Model | Predictor | Estimate | 95% CI | p-value | FDR | BF | Total n | n individuals |
|---|---|---|---|---|---|---|---|---|---|
| | | LSNS_change:LSNS_cut | –0.015 | –0.044, 0.015 | 0.3674 | 0.4547 | 0.28 | 1921 | 1408 |
| | | LSNS_cut | 0.040 | –0.094, 0.173 | | | | 1921 | 1408 |
| | | LSNS_base | –0.016 | –0.027,–0.005 | | | | 1921 | 1408 |
| | | LSNS_change | -0.017 | -0.033, -0.001 | | | | 1921 | 1408 |
| | | age_base | –0.035 | –0.042,–0.029 | | | | 1921 | 1408 |
| | | age_change | –0.016 | –0.026,–0.007 | | | | 1921 | 1408 |
| | 1 | gender | –0.382 | –0.466,–0.298 | | | | 1921 | 1408 |
| | | LSNS_change:LSNS_cut | -0.002 | -0.034, 0.03 | 0.4505 | 0.4547 | 0.16 | 1777 | 1315 |
| | | LSNS_cut | 0.032 | -0.106, 0.169 | | | | 1777 | 1315 |
| | | LSNS_base | -0.009 | –0.021, 0.002 | | | | 1777 | 1315 |
| | | LSNS_change | -0.019 | -0.036, -0.002 | | | | 1777 | 1315 |
| | | age_base | -0.033 | -0.04, -0.026 | | | | 1777 | 1315 |
| | | age_change | –0.018 | –0.028,–0.008 | | | | 1777 | 1315 |
| | | gender | –0.421 | –0.508,–0.335 | | | | 1777 | 1315 |
| | | BMI | –0.032 | -0.074, 0.011 | | | | 1777 | 1315 |
| | | CESD | -0.120 | –0.164,–0.077 | | | | 1777 | 1315 |
| | | diabetes | –0.038 | -0.156, 0.08 | | | | 1777 | 1315 |
| | | education | –0.173 | –0.315,–0.031 | | | | 1777 | 1315 |
| Memory | 2 | hypertension | 0.017 | -0.074, 0.108 | | | | 1777 | 1315 |
| | | LSNS_change:LSNS_cut | -0.005 | -0.034, 0.024 | 0.3146 | 0.4547 | 0.15 | 2043 | 1470 |
| | | LSNS_cut | –0.116 | –0.246, 0.014 | | | | 2043 | 1470 |
| | | LSNS_base | –0.012 | –0.022,–0.002 | | | | 2043 | 1470 |
| | | LSNS_change | 0.001 | –0.015, 0.017 | | | | 2043 | 1470 |
| | | age_base | –0.038 | –0.044,–0.032 | | | | 2043 | 1470 |
| | | age_change | -0.037 | –0.047,–0.027 | | | | 2043 | 1470 |
| | 1 | gender | -0.105 | –0.183,–0.028 | | | | 2043 | 1470 |
| | | LSNS_change:LSNS_cut | -0.002 | –0.033, 0.029 | 0.4507 | 0.4547 | 0.15 | 1888 | 1376 |
| | | LSNS_cut | –0.120 | -0.252, 0.012 | | | | 1888 | 1376 |
| | | LSNS_base | –0.010 | –0.021, 0 | | | | 1888 | 1376 |
| | | LSNS_change | –0.001 | –0.017, 0.015 | | | | 1888 | 1376 |
| | | age_base | –0.036 | –0.042,–0.029 | | | | 1888 | 1376 |
| | | age_change | –0.033 | –0.043,–0.023 | | | | 1888 | 1376 |
| | | gender | –0.121 | –0.201,–0.04 | | | | 1888 | 1376 |
| | | BMI | –0.012 | –0.052, 0.028 | | | | 1888 | 1376 |
| | | CESD | –0.028 | –0.068, 0.012 | | | | 1888 | 1376 |
| | | diabetes | –0.004 | –0.114, 0.106 | | | | 1888 | 1376 |
| | | education | –0.116 | –0.249, 0.016 | | | | 1888 | 1376 |
| Processing speed | 2 | hypertension | –0.066 | –0.151, 0.018 | | | | 1888 | 1376 |

full model1: dv~LSNS_base+LSNS_change+age_base+age_change+gender

full model2: model1 + hypertension+diabetes+education+BMI+CESD

The unit of effect sizes on hippocampal volume and cognitive functions are mm³/point on the LSNS and standard deviation/point on the LSNS, respectively.

* p<0.05, BF >3; ** p<0.01, BF >10; *** p<0.001, BF >30; **** p<0.0001, BF >100.

dv = dependent variable. CI = confidence interval. FDR = p-values after FDR-correction. BF = Bayes factor in favour of alternative hypothesis. total n = total number of observations. n individuals = number of participants. LSNS_base = baseline Lubben Social Network Score. LSNS_change = change in Lubben Social Network Score. CESD = Center for Epidemiological Studies-Depression.

**Appendix 1—table 21.** Adjusted regression coefficients and measures of significance of models stratified by gender.

| dv | Model | Gender | Predictor | Estimate | 95% CI | p-value | FDR | Total n | n individuals |
|---|---|---|---|---|---|---|---|---|---|
| | | | LSNS_base | −7.904 | −13.243,−2.565 | 0.0019** | 0.01136203571 | 821 | |
| | | | LSNS_change | −1.774 | −5.926, 2.391 | 0.2008 | 0.30123866746 | 821 | |
| | | | LSNS_base*age_change | −0.379 | -0.997, 0.238 | 0.1142 | 0.22831958503 | 821 | |
| | 1 | Female | LSNS_base*LSNS_change | -0.245 | -1.039, 0.549 | 0.2722 | 821.00000000000 | 641 | |
| | | | LSNS_base | −4.188 | -9.231, 0.856 | 0.0518 | 0.10361941628 | 846 | |
| | | | LSNS_change | −6.954 | −12.13,−1.763 | 0.0044** | 0.01330355028 | 846 | |
| | | | LSNS_base*age_change | −0.182 | −0.96, 0.594 | 0.3227 | 0.43024443281 | 846 | |
| | 1 | Male | LSNS_base*LSNS_change | 0.653 | -0.367, 1.68 | 0.895 | 846.00000000000 | 665 | |
| | | | LSNS_base | -10.046 | −15.757,−4.334 | 3e-04**** | 0.00356326264 | 757 | |
| | | | LSNS_change | −0.904 | −5.303, 3.515 | 0.3433 | 0.51492077295 | 757 | |
| | | | LSNS_base*age_change | -0.349 | -1.004, 0.305 | 0.1476 | 0.29523685430 | 757 | |
| | 2 | Female | LSNS_base*LSNS_change | −0.108 | -0.94, 0.723 | 0.3988 | 757.00000000000 | 595 | |
| | | | LSNS_base | -2.798 | −8.072, 2.477 | 0.1491 | 0.29813938291 | 799 | |
| | | | LSNS_change | -7.481 | −12.964,−1.976 | 0.004** | 0.02377229139 | 799 | |
| | | | LSNS_base*age_change | −0.203 | −1.019, 0.608 | 0.3114 | 0.47285051084 | 799 | |
| Hippo campal volume | 2 | Male | LSNS_base*LSNS_change | 0.683 | −0.385, 1.755 | 0.8949 | 799.00000000000 | 631 | |
| | | | LSNS_base | −0.033 | −0.046,−0.019 | 1.1e-06**** | 0.00001324504 | 994 | |
| | | | LSNS_change | −0.012 | -0.032, 0.009 | 0.1362 | 0.23354595086 | 994 | |
| | 1 | Female | LSNS_base*age_change | 0.001 | −0.003, 0.004 | 0.6347 | 0.63469552193 | 994 | |
| | | | LSNS_base | −0.024 | −0.037,−0.012 | 5.9e-05**** | 0.00035124409 | 1,054 | |
| | | | LSNS_change | 0.004 | −0.015, 0.024 | 0.6745 | 0.73579050782 | 1,054 | |
| | 1 | Male | LSNS_base*age_change | 0.001 | -0.002, 0.004 | 0.6629 | 0.73579050782 | 1,054 | |
| | | | LSNS_base | −0.020 | −0.035,−0.006 | 0.0025** | 0.01480365447 | 904 | |
| | | | LSNS_change | −0.001 | -0.023, 0.02 | 0.4542 | 0.54501865266 | 904 | |
| | 2 | Female | LSNS_base*age_change | 0.001 | -0.002, 0.005 | 0.7667 | 0.76666321855 | 904 | |
| | | | LSNS_base | -0.015 | −0.028,−0.002 | 0.0122* | 0.04879051790 | 989 | |
| | | | LSNS_change | 0.004 | −0.016, 0.025 | 0.6613 | 0.72350282272 | 989 | |
| Executive functions | 2 | Male | LSNS_base*age_change | 0.001 | -0.002, 0.004 | 0.6632 | 0.72350282272 | 989 | |
| | | | LSNS_base | -0.010 | -0.022, 0.001 | 0.0428* | 0.10263106694 | 926 | |
| | | | LSNS_change | −0.024 | −0.041,−0.006 | 0.004** | 0.01584355921 | 926 | |
| | 1 | Female | LSNS_base*age_change | 0.000 | −0.003, 0.002 | 0.4135 | 0.45108055271 | 926 | |
| | | | LSNS_base | −0.017 | −0.028,−0.005 | 0.0026** | 0.01044641225 | 995 | |
| | | | LSNS_change | −0.015 | -0.034, 0.003 | 0.0502 | 0.10361941628 | 995 | |
| | 1 | Male | LSNS_base*age_change | 0.001 | -0.002, 0.004 | 0.8058 | 0.80583563356 | 995 | |
| | | | LSNS_base | -0.005 | −0.017, 0.008 | 0.2349 | 0.40271104563 | 843 | |
| | | | LSNS_change | −0.023 | −0.041,−0.005 | 0.0072** | 0.02874128134 | 843 | |
| | 2 | Female | LSNS_base*age_change | 0.000 | -0.003, 0.003 | 0.4073 | 0.54304918017 | 843 | |
| | | | LSNS_base | -0.009 | −0.021, 0.003 | 0.0758 | 0.21377460911 | 934 | |
| | | | LSNS_change | −0.014 | −0.033, 0.006 | 0.0891 | 0.21377460911 | 934 | |
| Memory | 2 | Male | LSNS_base*age_change | 0.002 | -0.001, 0.005 | 0.8409 | 0.84087211792 | 934 | |

*Appendix 1—table 21 Continued on next page*

*Appendix 1—table 21 Continued*

| dv | Model | Gender | Predictor | Estimate | 95% CI | p-value | FDR | Total n | n individuals |
|---|---|---|---|---|---|---|---|---|---|
| | | | LSNS_base | −0.014 | −0.026,−0.002 | 0.0112* | | 0.03374550604 | 991 |
| | | | LSNS_change | -0.007 | −0.025, 0.012 | 0.2434 | | 0.32450988520 | 991 |
| | 1 | Female | LSNS_base*age_change | −0.001 | -0.004, 0.002 | 0.3495 | | 0.41941795284 | 991 |
| | | | LSNS_base | −0.022 | −0.032,−0.012 | 1.2e-05**** | | 0.00014476332 | 1,052 |
| | | | LSNS_change | −0.005 | -0.023, 0.012 | 0.27 | | 0.40493916979 | 1,052 |
| | 1 | Male | LSNS_base*age_change | −0.001 | -0.004, 0.002 | 0.2665 | | 0.40493916979 | 1,052 |
| | | | LSNS_base | −0.012 | −0.025, 0.001 | 0.0346* | | 0.10389307138 | 903 |
| | | | LSNS_change | −0.016 | −0.035, 0.003 | 0.0528 | | 0.12678399573 | 903 |
| | 2 | Female | LSNS_base*age_change | 0.001 | −0.002, 0.004 | 0.6653 | | 0.72576997994 | 903 |
| | | | LSNS_base | −0.020 | −0.031,−0.01 | 8.1e-05**** | | 0.00097014826 | 985 |
| | | | LSNS_change | 0.000 | −0.018, 0.019 | 0.5119 | | 0.68246828507 | 985 |
| Processing speed | 2 | Male | LSNS_base*age_change | −0.001 | -0.004, 0.002 | 0.3152 | | 0.47285051084 | 985 |

full model1: dv~LSNS_base+LSNS_change+age_base+age_change.

full model2: model1 + hypertension+diabetes+education+BMI+CESD.

The unit of effect sizes on hippocampal volume and cognitive functions for non-interaction models are mm³/point on the LSNS and standard deviation/point on the LSNS, respectively. For interaction models the unit in the denominator is multiplied by year or point on the LSNS.

* p<0.05; ** p<0.01; *** p<0.001; **** p<0.0001.

dv = dependent variable. CI = confidence interval. FDR = p-values after FDR-correction. LSNS_base = baseline Lubben Social Network Score. LSNS_change = change in Lubben Social Network Score. CESD = Center for Epidemiological Studies-Depression full model1: dv~LSNS_base +LSNS_change +age_base +age_change full model2: model1 +hypertension + diabetes +education + BMI +CESD

**Appendix 1—table 22.** Relevant regressions of bivariate latent change score models.

| dv | Predictor | Estimate | SE | p-value | q-value | n |
|---|---|---|---|---|---|---|
| ΔHCV | LSNS_base | 0.001 | 0.004 | 0.571 | 0.590 | 1536 |
| ΔLSNS | HCV_base | −0.199 | 0.168 | 0.118 | 0.123 | 1536 |
| ΔEF | LSNS_base | −0.014 | 0.007 | 0.024* | 0.094 | 1536 |
| ΔLSNS | EF_base | −0.188 | 0.161 | 0.121 | 0.123 | 1536 |
| ΔMemo | LSNS_base | 0.001 | 0.006 | 0.59 | 0.590 | 1536 |
| ΔLSNS | Memo_base | -0.303 | 0.165 | 0.033* | 0.123 | 1536 |
| ΔPS | LSNS_base | -0.007 | 0.007 | 0.177 | 0.354 | 1536 |
| ΔLSNS | PS_base | −0.202 | 0.174 | 0.123 | 0.123 | 1536 |

* p<0.05.

dv = dependent variable. SE = standard error. n = number of participants. _base = baseline score of. Δ = change in. LSNS = Lubben Social Network Score. HCV = hippocampal volume. EF = executive functions. Memo = memory performance. PS = processing speed.

**Appendix 1—table 23.** Fit indices of mediation analyses of model 1.

| Fit index | 311 | ok? | 411a | ok? | 411b | ok? | 411c | ok? |
|---|---|---|---|---|---|---|---|---|
| chisq | 0.733 | | 4.675 | | 0.607 | | 0.215 | |
| df | 1.000 | | 1.000 | | 1.000 | | 1.000 | |
| p-value | 0.392 | Good fit | 0.031 | Acceptable fit | 0.436 | Good fit | 0.643 | Good fit |
| chisq/df | 0.733 | Good fit | 4.675 | Unacceptable fit | 0.607 | Good fit | 0.215 | Good fit |
| rmsea | 0.000 | Good fit | 0.049 | Good fit | 0.000 | Good fit | 0.000 | Good fit |
| rmsea_lower | 0.000 | | 0.012 | | 0.000 | | 0.000 | |
| rmsea_upper | 0.064 | | 0.097 | | 0.062 | | 0.052 | |
| srmr | 0.009 | Good fit | 0.006 | Good fit | 0.002 | Good fit | 0.001 | Good fit |
| nnfi | 1.005 | Unacceptable fit | 0.946 | Unacceptable fit | 1.006 | Unacceptable fit | 1.018 | Unacceptable fit |

*Appendix 1—table 23 Continued on next page*

*Appendix 1—table 23 Continued*

| Fit index | 311 | ok? | 411a | ok? | 411b | ok? | 411c | ok? |
|---|---|---|---|---|---|---|---|---|
| cfi | 1.000 | Good fit | 0.996 | Good fit | 1.000 | Good fit | 1.000 | Good fit |

311: indirect effect of social isolation on hippocampal volume via chronic stress.
411a: indirect effect of social isolation on executive functions via hippocampal volume.
411b: indirect effect of social isolation on memory via hippocampal volume.
411c: indirect effect of social isolation on processing speed via hippocampal volume.
chisq = chi squared. df = degrees of freedom

**Appendix 1—table 24.** Fit indices of mediation analyses of model 2.

| Fit index | 312 | ok? | 412a | ok? | 412b | ok? | 412c | ok? |
|---|---|---|---|---|---|---|---|---|
| chisq | 1.243 | | 0.211 | | 5.238 | | 0.395 | |
| df | 5.000 | | 1.000 | | 1.000 | | 1.000 | |
| p-value | 0.941 | Good fit | 0.646 | Good fit | 0.022 | Acceptable fit | 0.530 | Good fit |
| chisq/df | 0.249 | Good fit | 0.211 | Good fit | 5.238 | Unacceptable fit | 0.395 | Good fit |
| rmsea | 0.000 | Good fit | 0.000 | Good fit | 0.053 | Acceptable fit | 0.000 | Good fit |
| rmsea_lower | 0.000 | | 0.000 | | 0.016 | | 0.000 | |
| rmsea_upper | 0.007 | | 0.052 | | 0.100 | | 0.058 | |
| srmr | 0.007 | Good fit | 0.001 | Good fit | 0.004 | Good fit | 0.001 | Good fit |
| nnfi | 1.023 | Unacceptable fit | 1.011 | Unacceptable fit | 0.896 | Unacceptable fit | 1.015 | Unacceptable fit |
| cfi | 1.000 | Good fit | 1.000 | Good fit | 0.996 | Good fit | 1.000 | Good fit |

312: indirect effect of social isolation on hippocampal volume via chronic stress.
412a: indirect effect of social isolation on executive functions via hippocampal volume.
412b: indirect effect of social isolation on memory via hippocampal volume.
412c: indirect effect of social isolation on processing speed via hippocampal volume.
chisq = chi squared. df = degrees of freedom

**Appendix 1—table 25.** Simulated Bayes factors above the threshold of 3.
BFA0, sided Bayes factor in favour of the alternative hypothesis; FWER, familywise error rate if the threshold would be set just below BFA0. In the simulation with randomly simulated values for our predictors of interest, 14 BFs exceeded the standard threshold of three. Given a family size of 12 tests, a threshold of 10.75 would maintain the FWER below 5%.

| BFA0 | FWER in % | n |
|---|---|---|
| 15.744 | 1.18 | 1 |
| 13.634 | 2.36 | 2 |
| 13.139 | 3.51 | 3 |
| 10.926 | 4.66 | 4 |
| 10.632 | 5.79 | 5 |
| 9.196 | 6.91 | 6 |
| 8.728 | 8.02 | 7 |
| 8.510 | 9.12 | 8 |
| 7.749 | 10.20 | 9 |
| 7.191 | 11.28 | 10 |
| 6.081 | 12.34 | 11 |
| 4.746 | 13.39 | 12 |
| 4.044 | 14.42 | 13 |
| 4.003 | 15.45 | 14 |

**Appendix 1—table 26.** Results of power simulation of Bayes factors.
BFA0b, sided Bayes factor in favour of the alternative hypothesis of baseline social isolation; BFA0c, sided Bayes factor in favour of the alternative hypothesis of change in social isolation; n, number of simulations in the category; model 1, model with reduced number of control variables; model

2, model with full number of control variables; effect, effect size per point in the Lubben Social Network Scale in years of baseline age.

Percentages of Bayes factors giving moderate or stronger evidence in favour of the alternative hypothesis (>3), giving anecdotal evidence (3 ≥ BF ≥ 1/3) and giving moderate or stronger evidence in favour of the null hypothesis (< 1/3).

| Category | BFA0b > 3 in % | 3 ≥ BFA0b ≥ 1/3 in % | BFA0b<1/3 in % | BFA0c>3 in % | 3 ≥ BFA0c ≥ 1/3 in % | BFA0c<1/3 in % | n |
|---|---|---|---|---|---|---|---|
| Overall | 44.23 | 31.41 | 24.36 | 28.85 | 30.45 | 40.71 | 312 |
| Model 1 | 45.51 | 30.13 | 24.36 | 30.13 | 30.13 | 39.74 | 156 |
| Model 2 | 42.95 | 32.69 | 24.36 | 27.56 | 30.77 | 41.67 | 156 |
| Effect = 0.1 | 9.62 | 38.46 | 51.92 | 5.77 | 24.04 | 70.19 | 104 |
| Effect = 0.2 | 37.50 | 44.23 | 18.27 | 21.15 | 39.42 | 39.42 | 104 |
| Effect = 0.5 | 85.58 | 11.54 | 2.88 | 59.62 | 27.88 | 12.50 | 104 |

**Appendix 1—table 27.** Results of power simulation of Bayes factors with adjusted thresholds for a family of 12 tests.

BFA0b, sided Bayes factor in favour of the alternative hypothesis of baseline social isolation; BFA0c, sided Bayes factor in favour of the alternative hypothesis of change in social isolation; n, number of simulations in the category; model 1, model with reduced number of control variables; model 2, model with full number of control variables; effect, effect size per point in the Lubben Social Network Scale in years of baseline age. Percentages of Bayes factors giving moderate or stronger evidence in favour of the alternative hypothesis (>10.75), giving anecdotal evidence (10.75 ≥ BF ≥ 1/3) and giving moderate or stronger evidence in favour of the null hypothesis (<1/3).

| Category | BFA0b>10.75 in % | 10.75 ≥ BFA0 b ≥ 1/3 in % | BFA0b<1/3 in % | BFA0c>10.75 in % | 10.75 ≥ BFA0 c ≥ 1/3 in % | BFA0c<1/3 in % | n |
|---|---|---|---|---|---|---|---|
| Overall | 37.18 | 38.46 | 24.36 | 20.83 | 38.46 | 40.71 | 312 |
| Model 1 | 38.46 | 37.18 | 24.36 | 21.79 | 38.46 | 39.74 | 156 |
| Model 2 | 35.90 | 39.74 | 24.36 | 19.87 | 38.46 | 41.67 | 156 |
| Effect = 0.1 | 5.77 | 42.31 | 51.92 | 0.96 | 28.85 | 70.19 | 104 |
| Effect = 0.2 | 24.04 | 57.69 | 18.27 | 14.42 | 46.15 | 39.42 | 104 |
| Effect = 0.5 | 81.73 | 15.38 | 2.88 | 47.12 | 40.38 | 12.50 | 104 |

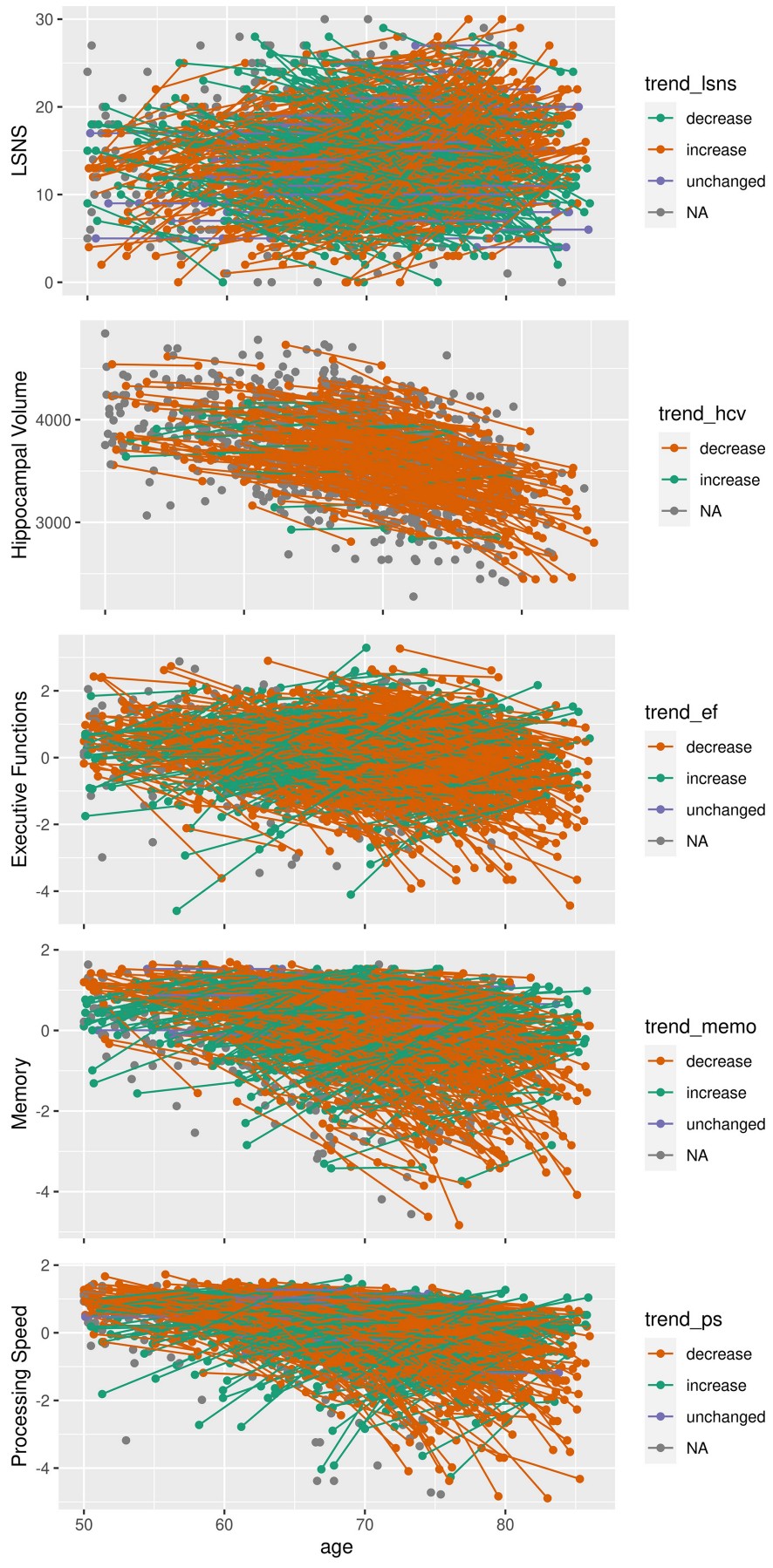

**Appendix 1—figure 1.** Spaghetti plots of individuals' developments in social isolation, hippocampal volume and cognitive functions over time. Outliers in memory and processing speed with scores smaller than 5 are not depicted. LSNS, Lubben Social Network Scale; trend_, individual trend in respective ordinate axis variable development. Unit for hippocampal volume is mm$^3$, unit for LSNS is points on the questionnaire and units for cognitive scores are deviations from the mean in standard deviations.

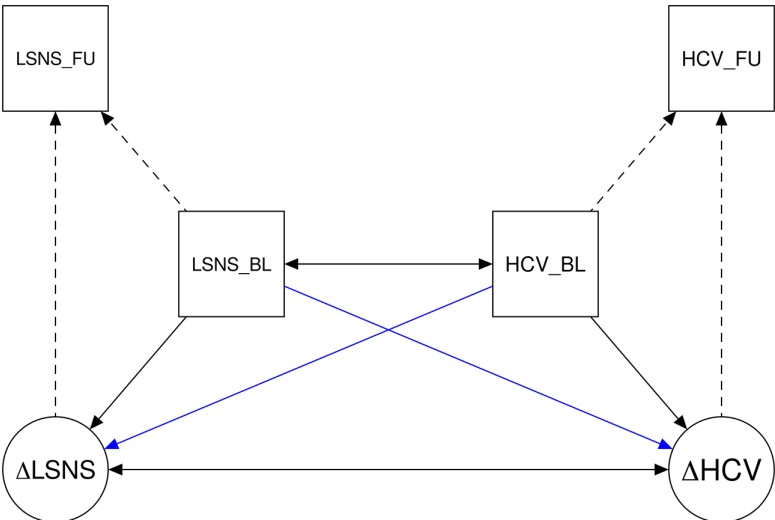

**Appendix 1—figure 2.** Simplified plot of the bivariate latent change score models. LSNS, Lubben Social Network Scale; HCV, hippocampal volume; BL, baseline; FU, follow up; Δ, change in. The blue arrows show our paths of interest.

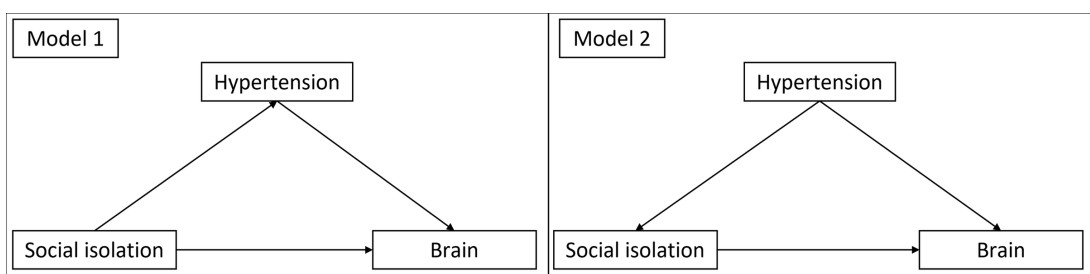

**Appendix 1—figure 3.** Directed acyclic graphs demonstrating the theoretical underpinnings of model 1 and 2. In model 1 the additional risk factors are assumed to be mediators and do not have to be controlled for. In model 2 they are assumed to be confounders. Therefore, they have to be controlled for.

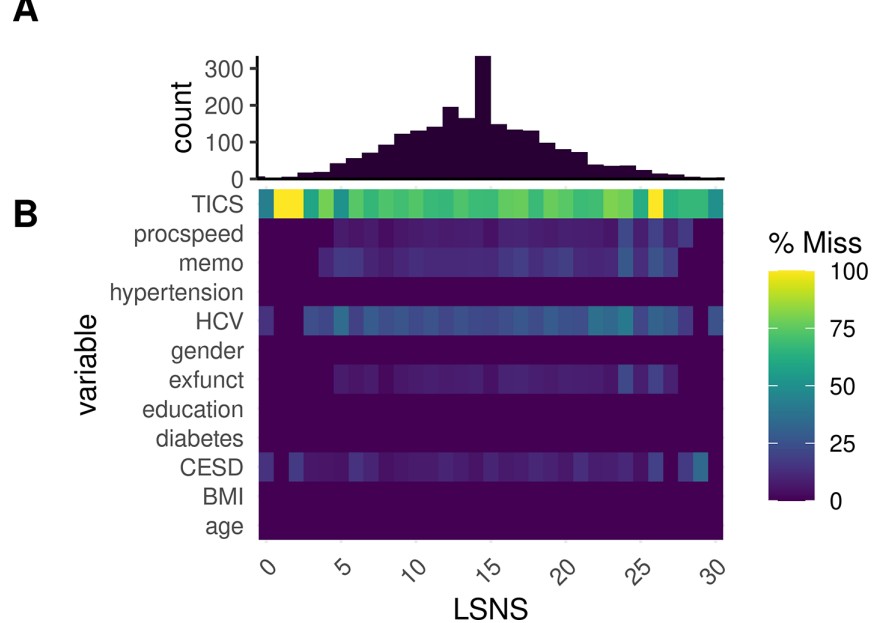

**Appendix 1—figure 4.** Proportional missingness relative to social isolation.
(**A**) Histogram of Lubben Social Network Scale (LSNS) scores by individual observation. (**B**) Heatmap of proportional missingness of variables for different LSNS scores.

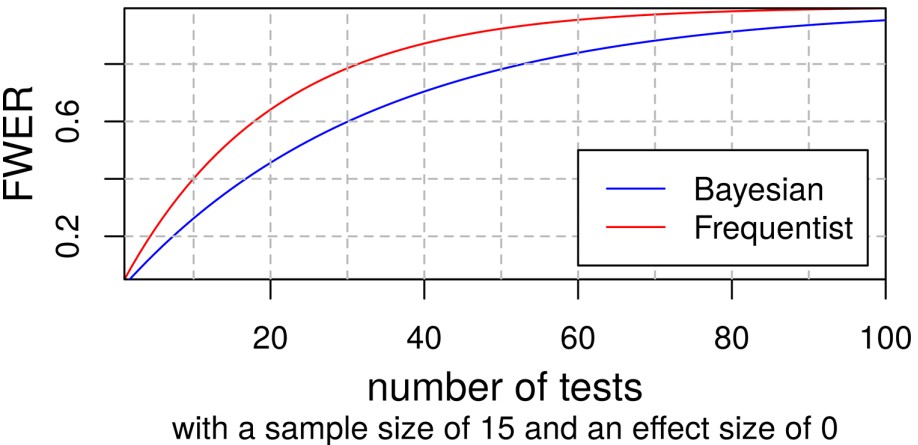

**Appendix 1—figure 5.** Familywise error rates of frequentist and bayesian t-tests.

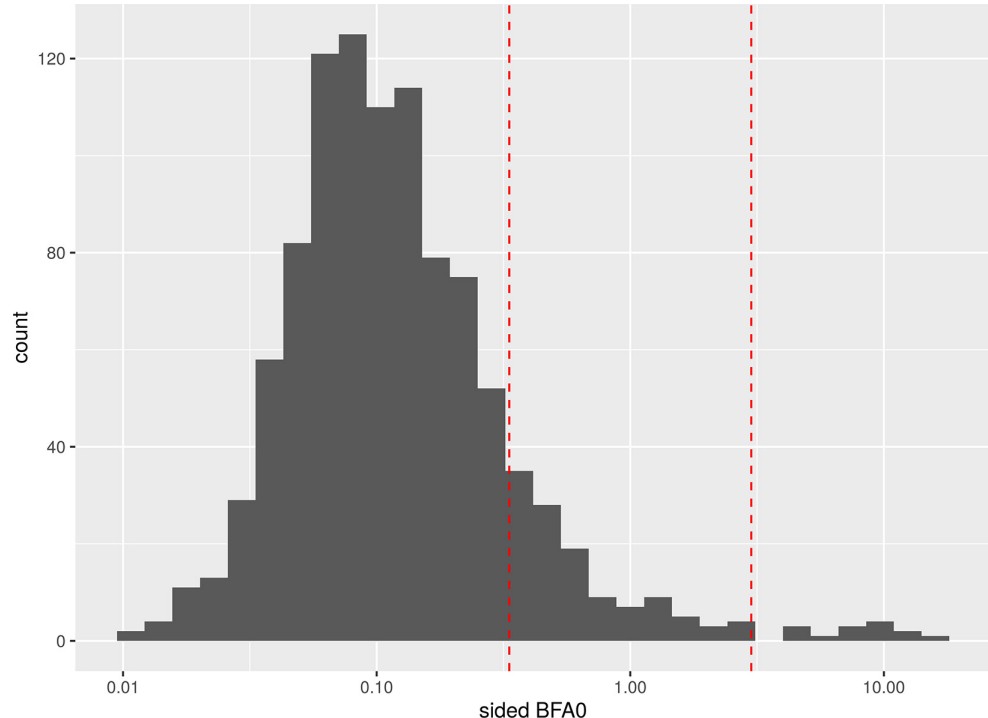

**Appendix 1—figure 6.** Histogram of Bayes factors (BFs) with randomly simulated values for our predictors of interest. The red lines show the traditional thresholds at 1/3 and 3.

