## [Editor Report]

This is a large-scale study on how social isolation influences brain structure and cognition. Baseline social isolation and change in social isolation were associated with smaller hippocampus volumes, reduced cortical thickness, and poorer cognitive function. The level of evidence is strong and converging cross-sectionally and longitudinally.

---

## [Decision Letter]

**Decision letter after peer review:**

Thank you for submitting your article "Impact of social isolation on grey matter structure and cognitive functions: a population-based longitudinal neuroimaging study" for consideration by *eLife*. Your article has been reviewed by 2 peer reviewers, and the evaluation has been overseen by a Reviewing Editor and Jonathan Roiser as the Senior Editor. The reviewers have opted to remain anonymous.

Essential revisions:

1) Revise the analytic methods to further validate the main findings (LME with random slopes and intercepts, confounders, etc).

2) Provide more data on sleep and physical activity and test/discuss its potential influence, if possible.

3) Discuss potential selection bias (education, SES) and its possible impact on the results.

*Reviewer #1 (Recommendations for the authors):*

The authors could try adding a random slope term for age change in their models and plot the extracted slopes as a separate figure in supplementary. Such random slope terms could also help interpret the interaction effect between baseline LSNS and age change.

As for the interaction effect, did the authors test if the threshold of 12 on LSNS (baseline) influences the slope between age change and hippocampal volume?

The authors should consider an ancillary LME (random slope and intercept) with age change predicting LSNS change, and presenting individual slopes in LSNS across time (in supplementary at least), to visualize how many older adults showed increased loneliness.

While going through the R code, I noticed that cognitive scores may have been standardized to the grand mean. It is more appropriate in longitudinal studies to standardize to the baseline mean. The authors could consider re-centering the cognitive composite scores to better estimate changes from the baseline.

The authors should change the y-axis in Figure 1 to "Hippocampal volume change" or "change in hippocampal volume".

*Reviewer #2 (Recommendations for the authors):*

I enjoyed reading the manuscript. I have no significant concerns but some suggestions that may help better understand why social isolation at baseline and over time was associated with indicators of impaired brain health and cognitive function.

1) A recent health survey among 6652 older adults showed that moderate and high physical activity was associated with 15%-30% lower likelihoods of loneliness and social isolation and with 27% to 150% higher protective factors. In addition, physical activity was associated with the mitigation of increased healthcare expenditures related to loneliness, social isolation, and low levels of protective factors (PMID 35905635). Physical activity has been linked to better structural and functional brain health (e.g., 27986412). Do you have data on participants' physical activity status?

2) A previous study demonstrated that increased loneliness was associated with worsening sleep quality among women, whereas the overall sleep quality decreased when perceived social isolation increased in men (36058046). Moreover, daytime sleepiness strongly predicts voluntary decreases in social contact (32817530). Since sleep has been linked to brain health and cognitive function (e.g., 27885006; 23589831), I wonder whether the observed association between loneliness and brain structure and cognitive function was mediated by sleep (e.g., duration or quality) or wake measures (daytime sleepiness).

3) Studies on circadian rhythmicity have shown that cognitive performance in various domains fluctuates across a day in synchrony with endogenous circadian rhythms (18066734). Thus, please provide more details regarding during which time of the day cognitive functions were assessed.

4) Factors that may determine whether a person feels socially isolated include marital/relationship status and employment. The latter does not matter for your finding, but it would be nice for the reader to understand better why some participants felt lonelier than others in your cohort.

5) Did you test whether people had suffered from mild traumatic brain injury before baseline or during baseline and follow-up? For example, I found an animal study suggesting that "the lateral septum and hippocampus are particularly vulnerable regions in repeated mild traumatic brain injury, possibly contributing to memory and emotional impairments associated with repeated concussions." (28078102)

---

## [Author Response]

Essential revisions:Reviewer #1 (Recommendations for the authors):The authors could try adding a random slope term for age change in their models and plot the extracted slopes as a separate figure in supplementary. Such random slope terms could also help interpret the interaction effect between baseline LSNS and age change.

We completely agree with the reviewer’s remark that adding random slopes to our models would further enhance their quality(*1*). Although the LIFE-Adult study is remarkably large, unfortunately our modelling choices are constrained by the study’s two timepoint design. Adding random slopes for each participant without dropping the random intercept term would result in structural unidentifiability as it would demand the models to estimate a greater number of predictors than the number of observations in the dataset(*2*). Random intercepts for each participant are indispensable for accounting for the interdependence of the data from observations from the same participant. Thus, we consider random intercepts to be of more central importance than random slopes and hence refrain from dropping them. To foster the adaptation of random slopes in future studies, we added a comment on their importance to our discussion of future study designs that should plan more numerous timepoints of data acquisition: <milestone-start />“<milestone-end />Studies with more numerous timepoints will be of importance to this end and will furthermore allow us to model important aspects like slopes for individual participants (van Doorn et al., 2021).”

Concerning the link between change in age and LSNS we indeed found a statistically significant effect of age change on higher social isolation in an ancillary LME. However, as the reviewer noticed, the per year effect is very small, meaning that it would need getting more than 20 years older to score one point higher on the LSNS sum score (see new Appendix table 2, see also answer below to questions 4a and 3b). We therefore tend to agree that in our sample, higher age does not affect social isolation substantially.

Concerning the link between change in age and LSNS we indeed found a statistically significant effect of age change on higher social isolation in an ancillary LME. However, as the reviewer noticed, the per year effect is very small, meaning that it would need getting more than 20 years older to score one point higher on the LSNS sum score (see new Appendix table 2, see also answer below to questions 4a and 3b). We therefore tend to agree that in our sample, higher age does not affect social isolation substantially. Nonetheless, the lack of any significant link between chronic stress and social isolation (see Appendix table 2) is hard to align with the stress-buffering hypothesis in spite of the missingness in the TICS.”.

We are grateful for your advice to visualise individual developments in social isolation and outcome measures over time in spaghetti plots and have done so to give our readers insight into these developments (see new Appendix figure 1). As you had assumed, there is no unequivocal pattern of increasing social isolation over time (see also answer to 3a). In addition, we decided to stick with presenting results of the statistical modeling of linear mixed effect using scatterplots in Figure 1, as this is regarded the most appropriate visualization of the tested effectors.

As for the interaction effect, did the authors test if the threshold of 12 on LSNS (baseline) influences the slope between age change and hippocampal volume?The authors should consider an ancillary LME (random slope and intercept) with age change predicting LSNS change, and presenting individual slopes in LSNS across time (in supplementary at least), to visualize how many older adults showed increased loneliness.

Thank you very much for pointing us towards the importance of shedding light on the link of age on social isolation (see also answers above, to questions 3a and 4a). Unfortunately, as discussed above (see answer to 1b), we could not implement the suggested models exactly due to the restraints of the LIFE-study’s two timepoint design. Nevertheless, we investigated the association of age and social isolation using an LME and found that both baseline age and change in age were linked significantly to higher social isolation (see new Appendix table 2). Interestingly, the effect size was noticeably larger for baseline age which is in line with the impression one gets from the spaghetti plot. While this might be partially attributed to certain biases (attrition, etc.) it could also point towards the existence of generational differences, i.e. less old-age social isolation in later generations.

While going through the R code, I noticed that cognitive scores may have been standardized to the grand mean. It is more appropriate in longitudinal studies to standardize to the baseline mean. The authors could consider re-centering the cognitive composite scores to better estimate changes from the baseline.

Thank you very much for your careful reading of our code and this helpful comment. We recalculated our analyses having standardized the data according to your recommendation and found the results to be almost unaffected (see new Appendix table 15). Therefore, we decided to stick to our original standardization procedures for the analyses in the main text, to adhere as best as possible to our preregistration, and referred to baseline-standardised results in Appendix table 15. We will take your advice to heart and follow it in future preregistrations and studies.

The authors should change the y-axis in Figure 1 to "Hippocampal volume change" or "change in hippocampal volume".

Indeed, in our models, we distinguished between from within subject effects of some of our predictors but not for the dependent variables, in this case 'hippocampal volume'. This is the common procedure for linear mixed models. Therefore, the y axis does not give change or differences, but absolute values of hippocampal volumes, this is why labels are correct in the current form and scatterplots are an appropriate way to visualise our results. Please see also response to question 4a, this reviewer, above, and associated Figures.

Reviewer #2 (Recommendations for the authors):I enjoyed reading the manuscript. I have no significant concerns but some suggestions that may help better understand why social isolation at baseline and over time was associated with indicators of impaired brain health and cognitive function.1) A recent health survey among 6652 older adults showed that moderate and high physical activity was associated with 15%-30% lower likelihoods of loneliness and social isolation and with 27% to 150% higher protective factors. In addition, physical activity was associated with the mitigation of increased healthcare expenditures related to loneliness, social isolation, and low levels of protective factors (PMID 35905635). Physical activity has been linked to better structural and functional brain health (e.g., 27986412). Do you have data on participants' physical activity status?

Thank you very much for pointing out the importance of physical activity for social isolation and its effects on health. The LIFE-Adult study assessed participants’ physical activity with the International Physical Activity Questionnaire (IPAQ). Based on the IPAQ developers’ guidelines we calculated weekly total metabolic equivalent of task (MET)s and ran a sensitivity analysis adding this variable to the model to control for potential protective effects of physical activity on social isolation-related brain damage. The results are presented in Appendix table 13, where no association of higher activity on outcome measures is seen and hardly any difference to the models not including this control variable is observable. Unfortunately, the missingness particularly at follow-up was extensive (see Appendix table 1,) and particularly results for change scores thus must be treated carefully. We have added this to the Discussion section, where it now reads „ While we could not observe significant contributions of physical activity or sleep quality measured using questionnaires in a smaller subsample on brain and cognitive outcomes, previous studies suggested that physical activity (Musich et al., 2022) and sleepiness (Holding et al., 2020) interact with social isolation and could protect against negative health effects of social isolation, and should therefore be explored in future studies that incorporated these outcomes more systematically.” Also see answer below to question (2), this reviewer.

2) A previous study demonstrated that increased loneliness was associated with worsening sleep quality among women, whereas the overall sleep quality decreased when perceived social isolation increased in men (36058046). Moreover, daytime sleepiness strongly predicts voluntary decreases in social contact (32817530). Since sleep has been linked to brain health and cognitive function (e.g., 27885006; 23589831), I wonder whether the observed association between loneliness and brain structure and cognitive function was mediated by sleep (e.g., duration or quality) or wake measures (daytime sleepiness).

We are grateful for your advice highlighting the potential role of sleep quality. The Pittsburgh Sleep Quality Index (PSQI) was used to measure sleep quality in the LIFE-Adult study. Following the guidelines, we calculated global PSQI scores (interpreted as a measure of sleep quality). We conducted mediation analyses analogous to our preregistered mediation analyses with PSQI score as mediator. A table of the results is provided below. As in the other mediation analyses with LSNS scores, we could not find a significant result (all p > 0.5) and estimates for the indirect effects are very small.

Additionally, we ran a sensitivity analysis with this additional control variable. No association with sleep quality and no clear pattern of difference between models with and without this further covariate seems to emerge (Appendix table 14). For some models, effect sizes for social isolation are marginally reduced while in other models the effect sizes increase. This is most likely attributable to the missingness in the data particularly at the second timepoint, i.e. more than 1100 participants without data entry (see Appendix table 1). We have commented on these additional results and the importance of sleep in the Discussion section, where it now reads: „While we could not observe significant contributions of physical activity or sleep quality measured using questionnaires in a smaller subsample on brain and cognitive outcomes, previous studies suggested that physical activity (Musich et al., 2022) and sleepiness (Holding et al., 2020) interact with social isolation and could protect against negative health effects of social isolation, and should therefore be explored in future studies that incorporated these outcomes more systematically.” Unfortunately, the data completeness of the LIFE-Adult study for physical activity and sleep quality is not on a par with the quality of the rest of the study.

We additionally explored the associations of sleep quality and physical activity with hippocampal volume and cognitive functions when only controlling for baseline age, change in age and gender. This showed a clear trend of negative effects of poor sleep quality and low physical activity on hippocampal volume and cognitive functions but p-values were only borderline significant. Please see Author response table 1 for details of these models.

**Author response table 1. sa2table1:** 

Adjusted regression coefficients and measures of significance of models for sleep quality and physical activity						
dv	Predictor	Estimate	95% CI	p-value	total n	n individuals
Hippocampal Volume	psqi	-5.764	-23.6, 12.144	0.2631	1,081	968
	age_base	-25.040	-28.409, -21.672		1,081	968
	age_change	-28.033	-32.052, -24.022		1,081	968
	gender	-39.143	-82.377, 4.077		1,081	968
Hippocampal Volume	ipaq	-10.785	-25.642, 3.968	0.0761	1,327	1,210
	age_base	-25.567	-28.549, -22.585		1,327	1,210
	age_change	-27.248	-31.086, -23.459		1,327	1,210
	gender	-37.001	-75.962, 1.956		1,327	1,210
Executive Functions	psqi	-0.040	-0.095, 0.015	0.076	1,179	1,046
	age_base	-0.008	-0.017, 0.001		1,179	1,046
	age_change	-0.053	-0.071, -0.034		1,179	1,046
	gender	-0.197	-0.312, -0.082		1,179	1,046
Executive Functions	ipaq	-0.031	-0.078, 0.017	0.1054	1,478	1,325
	age_base	-0.017	-0.025, -0.01		1,478	1,325
	age_change	-0.052	-0.07, -0.033		1,478	1,325
	gender	-0.126	-0.227, -0.024		1,478	1,325
Memory	psqi	-0.030	-0.078, 0.018	0.1087	1,134	1,000
	age_base	-0.032	-0.04, -0.024		1,134	1,000
	age_change	-0.032	-0.048, -0.017		1,134	1,000
	gender	-0.470	-0.571, -0.369		1,134	1,000
Memory	ipaq	-0.015	-0.058, 0.028	0.246	1,288
	age_base	-0.035	-0.042, -0.028		1,429	1,288
	age_change	0.002	-0.014, 0.017		1,429	1,288
	gender	-0.378	-0.469, -0.286		1,429	1,288
Processing Speed	psqi	-0.056	-0.099, -0.012	0.0059**	1,180	1,044
	age_base	-0.032	-0.039, -0.025		1,180	1,044
	age_change	-0.027	-0.042, -0.011		1,180	1,044
	gender	-0.160	-0.25, -0.07		1,180	1,044
Processing Speed	ipaq	-0.031	-0.071, 0.009	0.0623	1,478	1,328
	age_base	-0.036	-0.042, -0.03		1,478	1,328
	age_change	-0.038	-0.055, -0.022		1,478	1,328
	gender	-0.120	-0.202, -0.037		1,478	1,328

* p<0.05; ** p<0.01; *** p<0.001; **** p<0.0001

dv, dependent variable; CI, confidence interval; total n, total number of observations; n individuals, number of participants; psqi, global Pittsburgh Sleep Quality Index score; LSNS_base, ipaq, total weekly metabolic equivalents according to International Physical Activity Questionnaire

full model: dv~ipaq *or* psqi +age_base+age_change+gender

3) Studies on circadian rhythmicity have shown that cognitive performance in various domains fluctuates across a day in synchrony with endogenous circadian rhythms (18066734). Thus, please provide more details regarding during which time of the day cognitive functions were assessed.

Thank you very much for highlighting the importance of circadian rhythmicity for cognitive test performance. Over 90% of cognitive tests (CERAD and TMT) start times were between 8 and 14 o’clock with over 70% between 9 and 13 o’clock. We added a remark concerning test times to our methods section. We have added this to the limitations part and as a limitation that we did not control for circadian rhythmicity, where it now reads "In addition, time of day during testing might have affected cognitive performance, yet we did not control for this." Furthermore, we added to the methods section: “Most participants were cognitively tested between 9 a.m. and 1 p.m.” In Author response table 2 we provide table and a visualization of cognitive test (CERAD and TMT combined) start times (Author response image 1).

**Author response table 2. sa2table2:** 

start time	absolute n	proportion of participants
before 8	256	0.043163
08-09	537	0.090541
09-10	929	0.156635
10-11	1063	0.179228
11-12	1204	0.203001
12-13	1000	0.168606
13-14	609	0.102681
after 14	333	0.056146

**Author response image 1. sa2fig1:** 

4) Factors that may determine whether a person feels socially isolated include marital/relationship status and employment. The latter does not matter for your finding, but it would be nice for the reader to understand better why some participants felt lonelier than others in your cohort.

We agree with the reviewer and implemented these analyses, please see answer to 2a this reviewer on results and additions to the Discussion section.

5) Did you test whether people had suffered from mild traumatic brain injury before baseline or during baseline and follow-up? For example, I found an animal study suggesting that "the lateral septum and hippocampus are particularly vulnerable regions in repeated mild traumatic brain injury, possibly contributing to memory and emotional impairments associated with repeated concussions." (28078102)

Traumatic brain injury is indeed an important factor for cognitive decline and brain damage. Unfortunately, the LIFE-Adult study does not provide data that could be used to appropriately investigate its role in our study. We will encourage its assessments in upcoming initiatives for neuroimaging cohorts such as another wave of the LIFE-Adult study.

References

1. J. van Doorn, F. Aust, J. M. Haaf, A. M. Stefan, E. J. Wagenmakers, Bayes Factors for Mixed Models. *Comput. Brain Behav.* (2021).

2. J. H. A. Guillaume, J. D. Jakeman, S. Marsili-Libelli, M. Asher, P. Brunner, B. Croke, M. C. Hill, A. J. Jakeman, K. J. Keesman, S. Razavi, J. D. Stigter, Introductory overview of identifiability analysis: A guide to evaluating whether you have the right type of data for your modeling purpose. *Environ. Model. Softw.*
**119** (2019), pp. 418–432.